# Tightening Regret Lower and Upper Bounds in Restless Rising Bandits

**Cristiano Migali**
Politecnico di Milano, Milan, Italy
cristiano.migali@mail.polimi.it

**Marco Mussi**
Politecnico di Milano, Milan, Italy
marco.mussi@polimi.it

**Gianmarco Genalti**
Politecnico di Milano, Milan, Italy
gianmarco.genalti@polimi.it

**Alberto Maria Metelli**
Politecnico di Milano, Milan, Italy
albertomaria.metelli@polimi.it

## Abstract

*Restless* Multi-Armed Bandits (MABs) are a general framework designed to handle real-world decision-making problems where the expected rewards evolve over time, such as in recommender systems and dynamic pricing. In this work, we investigate from a theoretical standpoint two well-known structured subclasses of restless MABs: the *rising* and the *rising concave* settings, where the expected reward of each arm evolves over time following an unknown *non-decreasing* and a *non-decreasing concave* function, respectively. By providing a novel methodology of independent interest for general restless bandits, we establish new lower bounds on the expected cumulative regret for both settings. In the rising case, we prove a lower bound of order $\Omega(T^{2/3})$, matching known upper bounds for restless bandits; whereas, in the rising concave case, we derive a lower bound of order $\Omega(T^{3/5})$, proving for the first time that this setting is provably more challenging than stationary MABs. Then, we introduce `Rising Concave Budgeted Exploration` (`RC-BE`($\alpha$)), a new regret minimization algorithm designed for the rising concave MABs. By devising a novel proof technique, we show that the expected cumulative regret of `RC-BE`($\alpha$) is in the order of $\widetilde{\mathcal{O}}(T^{7/11})$. These results collectively make a step towards closing the gap in rising concave MABs, positioning them between stationary and general restless bandit settings in terms of statistical complexity.

## 1 Introduction

Multi-Armed Bandits (MABs, Lattimore and Szepesvári, 2020) are a well-known framework to model decision-making problems, where, for each round, an agent chooses (*pulls*) an action (*arm*) among a set of available actions and observes a *reward*, i.e., numerical feedback which represents the goodness of the choice. In this setting, the goal of the learner is to minimize the expected cumulative regret accumulated during the interaction, i.e., the sum over time of the difference between the expected reward of the optimal arm and that of the chosen one. The standard MAB setting considers stationary reward distributions. However, in many real-world decision-making problems, the expected rewards of available actions can vary over time due to changes in the surrounding environment, such as shifting in consumer preferences for online marketplaces (Wu et al., 2018) or evolving health status of patients in treatment selection during clinical trials (Aziz et al., 2021). To address such dynamics, the *restless* MABs framework (Tekin and Liu, 2012) has been introduced. This model generalizes the classical MAB setting by explicitly incorporating the *non-stationarity* of the arms.[1]

---

[1]With a slight abuse of terminology, we will use the words *non-stationary* and *restless* interchangeably.

39th Conference on Neural Information Processing Systems (NeurIPS 2025).

| | | Holds for | | | Result |
|---|---|---|---|---|---|
| | | Restless | Restless Rising | Restless Rising Concave | |
| **Lower Bounds** | Lattimore and Szepesvári 2020 (Thm. 15.2) | ✓ | ✓ | ✓ | $\Omega(T^{1/2})$ |
| | Besbes et al. 2014 (Thm. 1) | ✓ | ✗ | ✗ | $\Omega(T^{2/3})$ |
| | **This work** (Thm. 3.2) | ✓ | ✓ | ✗ | $\Omega(T^{1/2}) \rightarrow \Omega(T^{2/3})$ |
| | **This work** (Thm. 3.3) | ✓ | ✓ | ✓ | $\Omega(T^{1/2}) \rightarrow \Omega(T^{3/5})$ |
| **Upper Bounds** | Besbes et al. 2014 (Thm. 2) | ✓ | ✓ | ✓ | $\mathcal{O}(T^{2/3})$ |
| | Metelli et al. 2022 (Thm. 5.3) | ✗ | ✗ | ✓ | $\tilde{\mathcal{O}}(T^{2/3})$ |
| | **This work** (Thm. 4.4) | ✗ | ✗ | ✓ | $\mathcal{O}(T^{2/3}) \rightarrow \tilde{\mathcal{O}}(T^{7/11})$ |

Table 1: Existing and new bounds for the *restless*, *restless rising* and *restless rising concave* settings. The arrow → points from the previous best result to the improved one presented in this paper.

Non-stationarity in bandit problems has been addressed through a variety of models and methods, such as restless bandits with *abrupt changes* in the reward distribution (e.g., Garivier and Moulines, 2011), *smoothly* evolving expected rewards (e.g., Trovò et al., 2020), and settings where the *total variation* of expected rewards is bounded over time (e.g., Besbes et al., 2014). These frameworks allow the expected rewards to fluctuate in complex ways, such as increasing and then decreasing, without constraints on their direction of change. In contrast, there are important classes of bandit models that enforce *monotonicity* on the expected rewards. These include *rising bandits* (Heidari et al., 2016; Metelli et al., 2022), where expected rewards are non-decreasing, and *rotting bandits* (Levine et al., 2017; Seznec et al., 2019, 2020), where they are non-increasing. Such models are well-suited for capturing structured real-world dynamics, including online model selection (Metelli et al., 2022), hyperparameter optimization (Mussi et al., 2024), and recommendation systems (Levine et al., 2017).

**Motivation.** In this work, we focus on the restless *rising* bandits and restless *rising concave* bandits and we aim to characterize them from a theoretical standpoint since several fundamental questions remain unresolved. In the general restless bandit setting, where the expected rewards may vary over time with bounded variation over $T$ rounds, the minimax regret is known to be lower bounded by $\Omega(T^{2/3})$ (Besbes et al., 2014).[2] However, no regret lower bound has been derived for the specific class of non-decreasing (rising) or non-decreasing concave (rising concave) restless bandits yet, making the classical lower bound for stationary bandits, $\Omega(T^{1/2})$ (Lattimore and Szepesvári, 2020, Thm. 15.2), the best available reference, and leaving the following question open.

> **Question 1**: Is it possible to conceive regret **lower bounds** for restless rising and restless rising concave bandits that are *strictly larger* than the $\Omega(T^{1/2})$ bound for stationary bandits?

The currently available algorithms for restless rising bandits are those designed for general restless bandits with bounded variation, which achieve a regret upper bound of order $\mathcal{O}(T^{2/3})$ (Besbes et al., 2014). When incorporating concavity, more specific algorithms have been proposed (Metelli et al., 2022), but unfortunately, they fail to improve the regret order. This generates the following question.

> **Question 2**: Is it possible to devise algorithms for restless rising and rising concave bandits whose regret **upper bounds** are *strictly smaller* than the $\mathcal{O}(T^{2/3})$ bound for general restless bandits?

**Original Contribution.** In this paper, we aim to provide an answer to the research questions presented above, making a step towards the complete statistical characterization of restless rising and restless rising concave bandits. The contribution is summarized as follows:

- In Section 3, we provide a *general recipe* for deriving regret lower bounds for restless bandits, which generalizes the construction of Besbes et al. (2014) and is of potential independent interest (Lemma 3.1). We then *specialize* this construction to the cases of rising and rising concave bandits. First, we derive a lower bound of order $\Omega(T^{2/3})$ for rising bandits, showing that this setting shares the same statistical complexity as general restless bandits (Theorem 3.2) and answering negatively

---

[2]We use $\Omega(\cdot)$ and $\mathcal{O}(\cdot)$ to highlight the dependence on $T$ in the lower and upper bounds, respectively, omitting constant factors. For upper bounds, we also use $\tilde{\mathcal{O}}(\cdot)$ to suppress logarithmic dependencies on $T$ too.

to **Question 2** for rising bandits. Second, for restless rising concave bandits, we show that the regret is at least of order $\Omega(T^{3/5})$, showing that this setting is more challenging than stationary MABs (Theorem 3.3). These results provide a positive answer to **Question 1** for both settings.

- In Section 4, we present `Rising Concave Budgeted Exploration (RC-BE(α))`, a novel regret minimization algorithm for restless rising concave MABs, which extends `Budgeted Exploration` (Jia et al., 2023). By devising a novel analysis, we provide an upper bound on its regret of order $\widetilde{\mathcal{O}}(T^{7/11})$ (Theorem 4.4) with no requested knowledge of the learning horizon or of the total variation. This result improves upon the current best upper bound of order $\mathcal{O}(T^{2/3})$ and provides a positive answer to **Question 2** for rising concave bandits.

Numerical simulations are provided in Section 5. Related works are discussed in Appendix A. Omitted proofs are provided in Appendices B and C for lower and upper bounds, respectively. A summary of known and new results presented in this paper is provided in Table 1.

## 2 Setting

A *restless $K$-armed MAB* (Tekin and Liu, 2012; Lattimore and Szepesvári, 2020) is defined as a vector of probability distributions $\boldsymbol{\nu} = (\nu_i)_{i \in [\![K]\!]}$, where $\nu_i : \mathbb{N}_{\geqslant 1} \to \Delta(\mathbb{R})$.[3] Let $T \in \mathbb{N}_{\geqslant 1}$ be the learning horizon, at each round $t \in [\![T]\!]$, the agent selects an arm $I_t \in [\![K]\!]$ and observes a reward $R_t = X_{I_t, t}$ where $X_{i,t} \sim \nu_i(t)$ for all $i \in [\![K]\!], t \in \mathbb{N}_{\geqslant 1}$. We denote the random table with all possible rewards as $\boldsymbol{X} = (X_{i,t})_{i \in [\![K]\!], t \in \mathbb{N}_{\geqslant 1}}$. For every arm $i \in [\![K]\!]$, we define its expected reward $\mu_i : \mathbb{N}_{\geqslant 1} \to \mathbb{R}$ as the expectation of the reward obtained by pulling such arm, i.e., $\mu_i(t) = \mathbb{E}_{X \sim \nu_i(t)}[X]$ and denote the vector of expected reward functions as $\boldsymbol{\mu} = (\mu_i)_{i \in [\![K]\!]}$. We assume that the expected rewards are bounded in $[0, 1]$, and that the realizations are $\sigma$-subgaussian.[4]

**Rising Bandits.** We revise the *rising* bandits notion, i.e., MABs with *non-decreasing* expected rewards (Heidari et al., 2016). Such a property is captured by the following assumption.

**Assumption 2.1** (Non-Decreasing expected reward). *Let $\boldsymbol{\nu}$ be a restless MAB. For every arm $i \in [\![K]\!]$ and round $t \in \mathbb{N}_{\geqslant 1}$, the function $\mu_i(t)$ is non-decreasing in $t$. In particular, defining the increments:*

$$\gamma_i(t) := \mu_i(t+1) - \mu_i(t) \geqslant 0.$$

We introduce a further assumption on the concavity of the expected rewards (Heidari et al., 2016).

**Assumption 2.2** (Concave expected reward). *Let $\boldsymbol{\nu}$ be a restless MAB. For every arm $i \in [\![K]\!]$ and round $t \in \mathbb{N}_{\geqslant 1}$, the function $\mu_i(t)$ is concave in $t$, i.e.:*

$$\gamma_i(t+1) - \gamma_i(t) \leqslant 0.$$

Formally, we call *restless rising* a restless MAB in which Assumption 2.1 holds, and *restless rising concave* a restless MAB in which both Assumptions 2.1 and 2.2 hold. From now on, we omit the adjective *restless* for the sake of conciseness.

**Learning Problem.** Let $t \in \mathbb{N}_{\geqslant 1}$ be a round, we denote with $\mathcal{H}_t = (I_l, R_l)_{l=1}^t$ the *history* of observations up to $t$. A (non-stationary deterministic) *policy* is a function $\pi : \mathcal{H}_{t-1} \mapsto I_t$ mapping a history to an arm, that is abbreviated as $\pi(t) := \pi(\mathcal{H}_{t-1})$. We define the performance of a policy $\pi$ in a restless MAB $\boldsymbol{\nu}$ as the *expected cumulative reward* collected over the $T$ rounds, formally:

$$J_{\boldsymbol{\nu}}(\pi, T) := \mathbb{E}_{\boldsymbol{X} \sim \boldsymbol{\nu}} \left[ \sum_{t=1}^{T} \mu_{I_t}(t) \right].$$

A policy $\pi_{\boldsymbol{\nu}}^*$ is *optimal* if it maximizes the expected cumulative reward: $\pi_{\boldsymbol{\nu}}^* \in \arg\max_{\pi} \{J_{\boldsymbol{\nu}}(\pi, T)\}$. In restless MABs, the optimal policy does not explicitly depend on $T$ and consists of pulling in each round the arm with the highest expected reward: $\pi_{\boldsymbol{\nu}}^*(t) \in \arg\max_{i \in [\![K]\!]} \mu_i(t)$ for every $t \in \mathbb{N}_{\geqslant 1}$. Denoting with $J_{\boldsymbol{\nu}}^*(T) := J_{\boldsymbol{\nu}}(\pi_{\boldsymbol{\nu}}^*, T)$ the expected cumulative reward of an optimal policy, the suboptimal policies $\pi$ are evaluated via the *expected cumulative regret*:

$$R_{\boldsymbol{\nu}}(\pi, T) := J_{\boldsymbol{\nu}}^*(T) - J_{\boldsymbol{\nu}}(\pi, T). \tag{1}$$

---

[3]Let $a, b \in \mathbb{N}_{\geqslant 1}, b \geqslant a$, we denote with $[\![a, b]\!] := \{a \ldots, b\}$, with $[\![a]\!] := [\![1, a]\!]$, and with $\Delta(\mathcal{X})$ the set of probability measures over the measurable set $\mathcal{X}$.

[4]A random variable $X$ is $\sigma$-subgaussian if $\mathbb{E}[e^{\lambda(X - \mathbb{E}[X])}] \leqslant e^{\frac{\sigma^2 \lambda^2}{2}}$, for every $\lambda \in \mathbb{R}$.

**Instances Characterization.** To characterize an instance $\boldsymbol{\nu}$, we introduce the following quantity, namely the *cumulative increment*, defined for every $t_1, t_2 \in \mathbb{N}_{\geqslant 1}$ with $t_1 \leqslant t_2$ as:

$$\Upsilon_{\boldsymbol{\nu}}(t_1, t_2) := \sum_{l=t_1}^{t_2-1} \max_{i \in [\![K]\!]} \gamma_i(l).$$

The cumulative increment extends to an arbitrary interval with $t_1$ and $t_2$ as extremes the analogous notion $\Upsilon_{\boldsymbol{\mu}}(T, q)$ employed in (Metelli et al., 2022), restricting to $q = 1$. It is immediate to show that $\Upsilon_{\boldsymbol{\nu}}(t_1, t_2) \in [0, K]$ since $\Upsilon_{\boldsymbol{\nu}}(t_1, t_2) \leqslant \sum_{l=t_1}^{t_2-1} \sum_{i \in [\![K]\!]} \gamma_i(l) \leqslant \sum_{i \in [\![K]\!]} 1 = K$. Analogously to what is done in (Besbes et al., 2014), we consider the class of instances whose cumulative increment over the learning horizon $T$ is bounded by a *variation budget* $V_T \in (0, K]$, which we assume known, formally $\Upsilon_{\boldsymbol{\nu}}(1, T) \leqslant V_T$. Then, we call, respectively, $\mathcal{E}_{\mathrm{r}}^{\sigma}(T, V_T)$ and $\mathcal{E}_{\mathrm{c}}^{\sigma}(T, V_T)$ the set of rising MABs and rising concave MABs instances, with $\sigma$-subgaussian rewards, whose $\Upsilon_{\boldsymbol{\nu}}(1, T)$ satisfies the previous inequality.

# 3 Lower Bounds

In this section, we analyze the statistical complexity of the learning problem in both the rising and rising concave settings. To this end, we provide a regret lower bound suffered by any deterministic policy $\pi$ on a class of instances which are rising and rising concave, respectively.[5] In particular, we show that rising MABs are not easier than restless MABs with bounded variation (Besbes et al., 2014, Thm. 1) and that rising concave MABs are harder than stationary MABs (Lattimore and Szepesvári, 2020, Thm. 15.2). The analysis is carried out as follows. We develop a *general recipe* for regret lower bound construction on a richer class of restless MABs, described in Section 3.1. Then we specialize it to both the settings of interest (Sections 3.2 and 3.3).

## 3.1 General Recipe for the Lower Bound

We consider a class of restless MABs with the following structure. The set of rounds $\mathbb{N}_{\geqslant 1}$ is split into windows. Let $(\Delta_w)_{w \in \mathbb{N}_{\geqslant 1}}$ where $\Delta_w \in \mathbb{N}_{\geqslant 1}$ be a sequence of *window widths*. A window consists of a set of rounds $[\![s_w, e_w]\!] \subset \mathbb{N}_{\geqslant 1}$ where $s_w := \sum_{l=1}^{w-1} \Delta_l + 1$ and $e_w := \sum_{l=1}^{w} \Delta_l$, for $w \in \mathbb{N}_{\geqslant 1}$. For each window index $w \in \mathbb{N}_{\geqslant 1}$, we define two functions $\overline{\mu}_w, \widetilde{\mu}_w : [\![\Delta_w]\!] \to [0, 1]$, which we call *base* and *modified trend* respectively, that describe how the expected rewards of the arms evolve in $[\![s_w, e_w]\!]$. In particular, in each window, *at most* one arm among the $K$ has expected reward that follows the modified trend, while all the others have expected rewards that follow the base trend. The arm whose expected reward follows the modified trend can change between windows. We further enforce $\overline{\mu}_w(t) \leqslant \widetilde{\mu}_w(t)$ for all $w \in \mathbb{N}_{\geqslant 1}$, $t \in [\![\Delta_w]\!]$,[6] so that the arm whose expected reward follows the modified trend is the optimal one. More formally, let $w(t) := \min\{w \in \mathbb{N}_{\geqslant 1} \text{ s.t. } e_w \geqslant t\}$ be the index of the window which contains the round $t \in \mathbb{N}_{\geqslant 1}$. For each sequence $\boldsymbol{o} = (o_w)_{w \in \mathbb{N}_{\geqslant 1}}$ with $o_w \in [\![0, K]\!]$ in each window of index $w$ and for each subgaussian parameter $\sigma \geqslant 1$, we define an instance $\boldsymbol{\nu}_{\boldsymbol{o}}^{\sigma} = (\nu_{\boldsymbol{o},i}^{\sigma})_{i \in [\![K]\!]}$ as follows:

$$\nu_{\boldsymbol{o},i}^{\sigma}(t) := \begin{cases} \psi(\overline{\mu}_{w(t)}(t - s_{w(t)} + 1), \sigma) & \text{if } i \neq o_{w(t)} \\ \psi(\widetilde{\mu}_{w(t)}(t - s_{w(t)} + 1), \sigma) & \text{if } i = o_{w(t)} \end{cases}, \tag{2}$$

where $\psi(\mu, \sigma)$ is a probability distribution with parameters $\mu \in [0, 1]$, $\sigma \geqslant 1$ such that if $X \sim \psi(\mu, \sigma)$, then:

$$X = \begin{cases} \frac{3}{2}\sigma & \text{w.p. } \frac{1}{4} + \frac{\mu}{2\sigma} \\ -\frac{1}{2}\sigma & \text{w.p. } \frac{3}{4} - \frac{\mu}{2\sigma} \end{cases}.$$

First of all observe that $\mu \in [0, 1]$, $\sigma \geqslant 1$ imply $\mu/(2\sigma) \in [0, 1/2]$, so that the distribution is well-defined. Furthermore, if $X \sim \psi(\mu, \sigma)$, then, in virtue of Hoeffding's lemma, $X$ is $\sigma$-subgaussian, and, by direct calculation, it has expected value equal to $\mu$. Notice that, if $o_w = 0$, all the arms follow the

---

[5]Since we are considering stochastic bandits, our lower bounds can be generalized to stochastic policies, yielding analogous results, at the cost of additional notational complexity.

[6]We consider $\overline{\mu}_w(t)$ and $\widetilde{\mu}_w(t)$ both in the domain $t \in [\![\Delta_w]\!]$ instead of in the domain $[\![s_w, e_w]\!]$, for the sake of simplicity in the notation, as every window is defined independently from the others.

base trend, otherwise, $o_w$ corresponds to the only arm following the modified trend. We denote with $\overline{\boldsymbol{\mu}} = (\overline{\mu}_w)_{w \in \mathbb{N}_{\geqslant 1}}$ and $\widetilde{\boldsymbol{\mu}} = (\widetilde{\mu}_w)_{w \in \mathbb{N}_{\geqslant 1}}$ the sequences of base and modified trends respectively, and with $\mathcal{E}^{\sigma}_{\overline{\boldsymbol{\mu}}, \widetilde{\boldsymbol{\mu}}} = \{\boldsymbol{\nu}^{\sigma}_o \text{ s.t. } \boldsymbol{o} \in [\![0, K]\!]^{\mathbb{N}_{\geqslant 1}}\}$ the class of instances that they induce by varying the sequence $\boldsymbol{o}$ of optimal arms in each window. The following result, whose proof is deferred to Appendix B, holds.

**Lemma 3.1** (General Lower Bound). *Under the assumption that $\overline{\mu}_w(t) \leqslant \widetilde{\mu}_w(t)$ for all $w \in \mathbb{N}_{\geqslant 1}$, $t \in [\![\Delta_w]\!]$, for any deterministic policy $\pi$, subgaussian parameter $\sigma \geqslant 1$, and learning horizon $T \in \mathbb{N}_{\geqslant 1}$, it holds that:*

$$\sup_{\boldsymbol{\nu} \in \mathcal{E}^{\sigma}_{\overline{\boldsymbol{\mu}}, \widetilde{\boldsymbol{\mu}}}} R_{\boldsymbol{\nu}}(\pi, T) \geqslant \sum_{w=1}^{w(T)} \left( 1 - \frac{1}{K} - \sqrt{\frac{\ln(2) D_w^{\overline{\boldsymbol{\mu}}, \widetilde{\boldsymbol{\mu}}, T, \sigma}}{2K}} \right) A_w^{\overline{\boldsymbol{\mu}}, \widetilde{\boldsymbol{\mu}}, T}, \tag{3}$$

*where:*

$$D_w^{\overline{\boldsymbol{\mu}}, \widetilde{\boldsymbol{\mu}}, T, \sigma} := \sum_{t=s_w}^{\min\{e_w, T\}} \mathrm{D}_{\mathrm{KL}}(\psi(\overline{\mu}_w(t - s_w + 1), \sigma) \| \psi(\widetilde{\mu}_w(t - s_w + 1), \sigma)),$$

$$A_w^{\overline{\boldsymbol{\mu}}, \widetilde{\boldsymbol{\mu}}, T} := \sum_{t=s_w}^{\min\{e_w, T\}} (\widetilde{\mu}_w(t - s_w + 1) - \overline{\mu}_w(t - s_w + 1)),$$

*for all $w \in [\![w(T)]\!]$, with $\mathrm{D}_{\mathrm{KL}}(\cdot \| \cdot)$ being the Kullback-Leibler divergence of the two distributions (formally defined in Appendix B).*

This result highlights the trade-off in designing a "challenging" restless instance. On the one hand, we do not want to make the base and modified trends too far apart, otherwise it would be easy for the agent to discern one from the other. This is reflected in Equation (3), as the term $D_w^{\overline{\boldsymbol{\mu}}, \widetilde{\boldsymbol{\mu}}, T, \sigma}$ increases when the two trends diverge and contributes to reducing the regret lower bound since $A_w^{\overline{\boldsymbol{\mu}}, \widetilde{\boldsymbol{\mu}}, T}$ is non-negative by construction. On the other hand, we want to maximize the area $A_w^{\overline{\boldsymbol{\mu}}, \widetilde{\boldsymbol{\mu}}, T}$ between the two trends. In this way, under the assumption that $D_w^{\overline{\boldsymbol{\mu}}, \widetilde{\boldsymbol{\mu}}, T, \sigma}$ is small enough so that the factor that multiplies $A_w^{\overline{\boldsymbol{\mu}}, \widetilde{\boldsymbol{\mu}}, T}$ is non-negative, we increase the regret lower bound.

## 3.2 Specializing the Lower Bound for the Rising Setting

In this part, we apply Lemma 3.1 to provide a regret lower bound for the class $\mathcal{E}^{\sigma}_{\mathrm{r}}(T, V_T)$, holding for any deterministic policy $\pi$. To this end, we construct sequences of window widths $(\Delta_{\mathrm{r}, w})_{w \in \mathbb{N}_{\geqslant 1}}$ and of base and modified trends $\overline{\boldsymbol{\mu}}_{\mathrm{r}}, \widetilde{\boldsymbol{\mu}}_{\mathrm{r}}$ such that $\mathcal{E}^{\sigma}_{\overline{\boldsymbol{\mu}}_{\mathrm{r}}, \widetilde{\boldsymbol{\mu}}_{\mathrm{r}}} \subseteq \mathcal{E}^{\sigma}_{\mathrm{r}}(T, V_T)$. A representation of the structure of the instances is depicted in Figure 1. We choose windows of the same width. In each window, the base and modified trend are both constant, the latter is greater than the former by a quantity $\varepsilon_{\mathrm{r}} > 0$ and the value of the modified trend in a window corresponds to the value of the base trend in the next window. In this way, we guarantee that the instances are rising no matter which arm follows the modified trend. In Appendix B, we formalize the instances and we prove that the following holds.

**Theorem 3.2** (Lower Bound for the Rising Setting). *For any deterministic policy $\pi$, subgaussian parameter $\sigma \geqslant 1$, and learning horizon $T \in \mathbb{N}_{\geqslant 1}$, $T \geqslant \sigma^2 K \min\{1, V_T\}^{-2}$, it holds that:*

$$\sup_{\boldsymbol{\nu} \in \mathcal{E}^{\sigma}_{\mathrm{r}}(T, V_T)} R_{\boldsymbol{\nu}}(\pi, T) \geqslant \frac{1}{64} \sigma^{\frac{2}{3}} T^{\frac{2}{3}} K^{\frac{1}{3}} \min\{1, V_T\}^{\frac{1}{3}}.$$

The orders of growth for $T$, $K$, and $V_T$ in this result match the upper bound for the general restless case with bounded variation (Besbes et al., 2014, Thm. 2) when $V_T \leqslant 1$.[7] This implies that rising MABs are not easier than general restless MABs with bounded variation despite the additional assumption. Thus, the characterization of the statistical complexity of this setting is completed.

## 3.3 Specializing the Lower Bound for the Rising Concave Setting

In this part, we provide a regret lower bound for the class $\mathcal{E}^{\sigma}_{\mathrm{c}}(T, V_T)$ holding for any deterministic policy $\pi$. In analogy to Section 3.2 for the rising setting, we construct sequences of window widths

---

[7]We believe this is an artifact of the analysis since, in our the lower bound construction, we have $\Upsilon(1, T) \leqslant 1$.

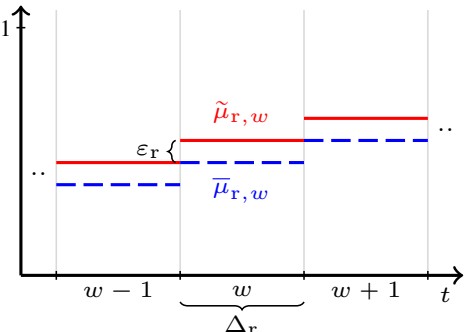

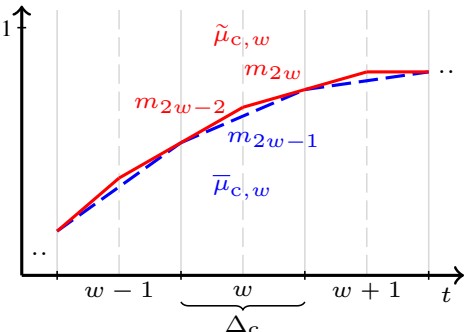

Figure 1: Base (dashed) and modified (solid) trends of the lower bound instances for the *rising* setting.

Figure 2: Base (dashed) and modified (solid) trends of the lower bound instances for the *rising concave* setting.

$(\Delta_{c,w})_{w\in\mathbb{N}_{\geqslant 1}}$ and of base and modified trends $\overline{\boldsymbol{\mu}}_c, \widetilde{\boldsymbol{\mu}}_c$ such that $\mathcal{E}^{\sigma}_{\overline{\boldsymbol{\mu}}_c, \widetilde{\boldsymbol{\mu}}_c} \subseteq \mathcal{E}^{\sigma}_c(T, V_T)$. A representation of the instances is depicted in Figure 2. We choose again windows of the same width. In each window, the base and modified trends share the same starting and ending values. Furthermore, the end value of expected rewards in a window matches the start value of expected rewards in the next window. The end value is greater than the start value to guarantee that the instances are rising. The base trend joins the two endpoints of the expected rewards of each window with a single segment, while the modified trend uses two segments. At the beginning, it rises with a slope greater than that of the base trend until half the window. At this point, the distance between the base and the modified trend in the window is maximum. Then, the modified trend keeps rising, but with a slope that is smaller than that of the base trend, until the two trends meet at the end of the window. The pattern repeats and the slopes are chosen in such a way that the slope of the second part of the modified trend in a window (which is the smallest slope in a window) corresponds to the slope of the first part of the modified trend in the next window (which is the greatest slope of an expected reward in a window). In this way, we guarantee that the instances are rising and concave, no matter the choice of which arm follows the modified trend. In Appendix B, we formally present the instances and we prove the following result.

**Theorem 3.3** (Lower Bound for the Rising Concave Setting). *For any deterministic policy $\pi$, subgaussian parameter $\sigma \geqslant 1$, and learning horizon $T \in \mathbb{N}_{\geqslant 1}$, $T \geqslant 2^{10}\sigma^2 K \min\{1, V_T\}^{-2}$, it holds that:*

$$\sup_{\boldsymbol{\nu}\in\mathcal{E}^{\sigma}_c(T,V_T)} R_{\boldsymbol{\nu}}(\pi, T) \geqslant 2^{-14}\sigma^{4/5}T^{\frac{3}{5}}K^{\frac{2}{5}}\min\{1, V_T\}^{\frac{1}{5}}.$$

This result proves that regret minimization in rising concave MABs represents a harder learning problem w.r.t. stationary MABs which are characterized by the usual $\Omega(T^{1/2})$ lower bound.

## 4 Upper Bound for the Rising Concave Setting

In this section, we present a novel regret minimization algorithm, `Rising Concave Budgeted Exploration` (RC-BE($\alpha$)), designed for rising concave MABs (Algorithm 1), and analyze its performance by providing an upper bound of the expected cumulative regret suffered on a generic instance $\boldsymbol{\nu} \in \mathcal{E}^{\sigma}_c(T, V_T)$. We show that this upper bound attains a strictly smaller rate w.r.t. the lower bound on the expected cumulative regret on a generic restless MAB with bounded variation (Besbes et al., 2014), and thus that rising concave MABs are indeed an easier setting w.r.t. them.

**Algorithm.** RC-BE($\alpha$) is an improvement of the `Budgeted Exploration` (BE) algorithm (Jia et al., 2023), originally designed for 2-armed general restless bandits.[8] The original BE algorithm works as follows. The learning horizon $T$ is split in windows of $\Delta \in \mathbb{N}_{\geqslant 1}$ rounds each. In each window, the algorithm restarts. At the beginning of each window, the agent carries out an exploration phase which consists of several *round-robin* cycles. In particular, the agent keeps track of the arms alive in

---

[8]The extension of BE to $K$-armed bandits is proposed in the unpublished preprint (Jia et al., 2024) for the case of *smooth* MABs. However, we have found soundness issues in the analysis proposed there (see Appendix F). For this reason, we will develop an independent analysis which overcomes these issues.

**Algorithm 1** RC-BE($\alpha$).

---

1: **Input**: $\alpha \geqslant 1$, $K \in \mathbb{N}_{\geqslant 2}$
2: Initialize $w \leftarrow 1$, $d \leftarrow 1$, $\mathcal{A} \leftarrow [\![K]\!]$, $\mathcal{B} \leftarrow \mathcal{A}$, $\hat{S}_i \leftarrow 0$, $\forall i \in [\![K]\!]$
3: **for** $t \in [\![T]\!]$ **do**
4:     **if** $d = \Delta_w^{(\alpha)} + 1$ **then**
5:         Increment $w \leftarrow w + 1$
6:         Reset $d \leftarrow 1$, $\mathcal{A} \leftarrow [\![K]\!]$, $\mathcal{B} \leftarrow \mathcal{A}$, $\hat{S}_i \leftarrow 0$, $\forall i \in [\![K]\!]$
7:     **end if**
8:     Pull $I_t \in \mathcal{B}$
9:     Remove $\mathcal{B} \leftarrow \mathcal{B} \setminus \{I_t\}$
10:     Observe $R_t = X_{I_t, t}$
11:     Update $\hat{S}_{I_t} \leftarrow \hat{S}_{I_t} + R_t$
12:     **if** $\mathcal{B} = \{\}$ **then**
13:         Compute $\hat{S}^* \leftarrow \max_{i \in \mathcal{A}} \hat{S}_i$
14:         **for** $i \in [\![K]\!]$ **do**
15:             **if** $i \in \mathcal{A}$ and $\hat{S}_i + B_w^{(\alpha)} < \hat{S}^*$ **then**
16:                 Remove $\mathcal{A} \leftarrow \mathcal{A} \setminus \{i\}$
17:             **end if**
18:         **end for**
19:         Reset $\mathcal{B} \leftarrow \mathcal{A}$
20:     **end if**
21:     Increment $d \leftarrow d + 1$
22: **end for**

---

the current window in a set $\mathcal{A} \subseteq [\![K]\!]$, initialized to $[\![K]\!]$ at the beginning of each window, and, in each round-robin cycle, pulls each of these arms once. The agent cumulates the observed rewards for each arm in the variables $\hat{S}_i$ with $i \in [\![K]\!]$. At the end of each round-robin cycle, the agent compares the cumulative reward of each alive arm with the maximum cumulative reward among alive arms $\hat{S}^* := \max_{i \in \mathcal{A}} \hat{S}_i$. If for $i \in \mathcal{A}$ we have $\hat{S}_i + B < \hat{S}^*$, where $B > 0$ is a parameter of the algorithm, we say that arm $i$ has run out of budget and the agent removes it from the set of alive arms. It can happen that, after several round-robin cycles, the set of alive arms becomes a singleton: $\mathcal{A} = \{\hat{i}^*\}$. In this case, no more eliminations can happen and the agent will commit to the remaining arm $\hat{i}^*$.

RC-BE($\alpha$) extends the original algorithm as follows. It exploits the concavity of the instance through increasing window widths $\Delta_w^{(\alpha)} := \lceil w^\alpha \rceil$ and corresponding budgets $B_w^{(\alpha)} := 2(1 + 2\sigma(\Delta_w^{(\alpha)} \ln(2K\Delta_w^{(\alpha)}))^{1/2})$. The rationale is the following. The algorithm suffers a high regret in windows during which the optimal arm changes. Indeed, in windows where no change happens, the algorithm is likely to commit to the best arm, suffering no regret after the initial exploration phase. Conversely, in windows where the optimal arm changes, the algorithm could commit to an arm that then becomes suboptimal, or it could fail in estimating the optimal arm. In this case, the regret increases with the distance of the expected rewards of $\hat{i}^*$ and the actual optimal arm in round $t$: $i_t^* \in \arg\max_{i \in [\![K]\!]} \mu_i(t)$. Thanks to the concavity, the maximum increment $\max_{i \in [\![K]\!]} \gamma_i(t)$ decreases as $t$ increases. Thus, as time passes, if the optimal arm changes, it takes longer for the expected rewards of $\hat{i}^*$ and $i_t^*$ to diverge significantly. Hence, we can restart the algorithm with a lower frequency, which is equivalent to having windows with increasing width.

**Regret Analysis.** RC-BE($\alpha$) partitions the set of rounds $\mathbb{N}_{\geqslant 1}$ in windows $[\![s_w^{(\alpha)}, e_w^{(\alpha)}]\!]$ with $s_w^{(\alpha)} := \sum_{l=1}^{w-1} \Delta_l^{(\alpha)} + 1$ and $e_w^{(\alpha)} := \sum_{l=1}^{w} \Delta_l^{(\alpha)}$, for $w \in \mathbb{N}_{\geqslant 1}$. Let $w^{(\alpha)}(t) = \min\{w \in \mathbb{N}_{\geqslant 1}$ s.t. $e_w^{(\alpha)} \geqslant t\}$ be the index of the window that contains the round $t \in \mathbb{N}_{\geqslant 1}$. Thus, the learning horizon $T$ is split in $w^{(\alpha)}(T)$ windows. In what follows, we bound the regret suffered by RC-BE($\alpha$) on the set of windows $\mathcal{W}$ which enjoy certain properties that we introduce later. To this end, we denote the regret suffered by a policy $\pi$ *on a set of windows* $\mathcal{W} \subset \mathbb{N}_{\geqslant 1}$, $|\mathcal{W}| < \infty$ as:

$$R_{\boldsymbol{\nu}}(\pi, \mathcal{W}) := \sum_{w \in \mathcal{W}} \sum_{t = s_w^{(\alpha)}}^{e_w^{(\alpha)}} \mathop{\mathbb{E}}_{\boldsymbol{X} \sim \boldsymbol{\nu}} \left[ \mu_{i_t^*}(t) - \mu_{I_t}(t) \right].$$

Now, we present the properties which induce the classes of windows of interest for the analysis. In particular, we need to formally characterize the fact that, in a window, the optimal arm can change. To this end, we introduce the following definitions, in analogy to what is done in (Jia et al., 2024).

**Definition 4.1** (Overtaking). *An arm $i \in [\![K]\!]$ overtakes an arm $j \in [\![K]\!]$ at time $t \in \mathbb{N}_{\geqslant 2}$ if $\mu_i(t-1) \leqslant \mu_j(t-1)$ and $\mu_i(t) \geqslant \mu_j(t)$. Formally, we write $i \uparrow_t j$ (note that $i \uparrow_t i$).*

**Definition 4.2** (Crossing). *Two arms $i, j \in [\![K]\!]$ cross at time $t \in \mathbb{N}_{\geqslant 2}$, if $i \uparrow_t j$ or $j \uparrow_t i$. Formally, we write $i \times_t j$ (note that $i \times_t i$).*

We introduce a binary relation for arms that cross in the $w$-th window. For $w \in \mathbb{N}_{\geqslant 1}$, $i, j \in [\![K]\!]$:

$$i \; {}_w\!\times j \quad \text{iff} \quad i \times_t j \text{ for some } t \in [\![s_w^{(\alpha)} + 1, e_w^{(\alpha)}]\!].$$

Let ${}_w\!\times^+$ be the transitive closure of ${}_w\!\times$. ${}_w\!\times^+$ is an equivalence relation since ${}_w\!\times$ is reflexive and symmetric. For an arm $i \in [\![K]\!]$, we denote with $[i]_{{}_w\!\times^+}$ its equivalence class w.r.t. ${}_w\!\times^+$. Let:

$$\mathcal{I}_w^* := \left\{ i \in [\![K]\!] \text{ s.t. there exists } t \in [\![s_w^{(\alpha)}, e_w^{(\alpha)}]\!] \text{ with } i \in \arg\max_{j \in [\![K]\!]} \mu_j(t) \right\},$$

be the set of *optimal arms in window* $w$. Furthermore, we define $\mathcal{I}_w^\times := [i_w^*]_{{}_w\!\times^+}$ for some $i_w^* \in \mathcal{I}_w^*$. Observe that the definition is well posed since, in virtue of Lemma C.2, it does not depend on the choice of $i_w^*$. For $w \in \mathbb{N}_{\geqslant 1}$, $i \in [\![K]\!]$, we define the *diameter* of its equivalence class w.r.t. ${}_w\!\times^+$ as

$$d_w(i) := \max_{j,k \in [i]_{{}_w\!\times^+}, \, t \in [\![s_w^{(\alpha)}, e_w^{(\alpha)}]\!]} |\mu_j(t) - \mu_k(t)| .$$

We use the shorthand $d_w^*$ for $d_w(i_w^*)$ where $i_w^* \in \mathcal{I}_w^*$. The following lemma decomposes the regret suffered by RC-BE$(\alpha)$ during the $w$-th window as the sum of the regret due to the exploration phase plus the regret due to the commitment phase.

**Lemma 4.1.** *For all restless rising concave MABs $\boldsymbol{\nu}$, $\alpha \geqslant 1$, $w \in \mathbb{N}_{\geqslant 1}$ we have that:*

$$R_{\boldsymbol{\nu}}(\text{RC-BE}(\alpha), \{w\}) \leqslant \underbrace{3KB_w^{(\alpha)}}_{\text{Exploration}} + \underbrace{\Delta_w^{(\alpha)} d_w^*}_{\text{Commitment}} .$$

Thus, the regret due to exploration is proportional to the budget $B_w^{(\alpha)}$, while the regret suffered during the commitment phase depends on the width of the window $\Delta_w^{(\alpha)}$ and on the diameter $d_w^*$ of $\mathcal{I}_w^\times$. In windows where the optimal arm does not change, $\mathcal{I}_w^\times$ is a singleton and, thus, its diameter is 0. This reflects the fact that, in such windows, the algorithm suffers only the regret due to the exploration.

We now provide an upper bound for $d_w(i)$ with $w \in \mathbb{N}_{\geqslant 1}$, $i \in [\![K]\!]$ which exploits concavity.

**Lemma 4.2.** *For all restless rising concave MABs $\boldsymbol{\nu}$, $\alpha \geqslant 1$, $w \in \mathbb{N}_{\geqslant 1}$, $i \in [\![K]\!]$, we have that:*

$$d_w(i) \leqslant 8(1 + \alpha) \left( \left| [i]_{{}_w\!\times^+} \right| - 1 \right) w^{-1} \Upsilon_{\boldsymbol{\nu}}(1, e_w^{(\alpha)}) \leqslant 16\alpha K w^{-1} \Upsilon_{\boldsymbol{\nu}}(1, e_w^{(\alpha)}).$$

Recall that $\Upsilon_{\boldsymbol{\nu}}(1, e_w^{(\alpha)})$ is upper bounded by $K$. Thus, as expected, eventually the upper bound of the diameter decreases as $w$ increases. This reflects what we informally stated before. As time goes, due to the concavity, it takes more time for the expected rewards of arms which have crossed to diverge significantly. Thus, it makes sense to increase the width of the windows over time.

We now discriminate between two kinds of windows: those in which the expected rewards of arms which cross (and thus of the arms which belong to $\mathcal{I}_w^\times$) do not diverge significantly and those in which, instead, the converse happens. More formally, let $d \in (0, K]$:

$$\mathcal{W}_{\leqslant d}(T) := \left\{ w \in [\![w^{(\alpha)}(T)]\!] \text{ s.t. } d_w(i) \leqslant d \text{ for all } i \in [\![K]\!] \right\},$$

$$\mathcal{W}_{>d}(T) := \left\{ w \in [\![w^{(\alpha)}(T)]\!] \text{ s.t. } d_w(i) > d \text{ for some } i \in [\![K]\!] \right\}.$$

In the second class of windows, we have no upper bound to the diameter $d_w^*$ other than that of Lemma 4.2, which considers a worst-case scenario in which the divergence of the expected rewards of the arms which cross is the maximum possible. We now show that this scenario, in the rising concave setting, can happen only a limited number of times. In particular, this is translated into an upper bound to the number of windows in $\mathcal{W}_{>d}(T)$, which is captured by the following lemma.

**Lemma 4.3.** *For all restless rising concave MABs $\boldsymbol{\nu}$, $\alpha \geqslant 1$, $T \in \mathbb{N}_{\geqslant 1}$, $d \in (0, K]$, we have that:*

$$|\mathcal{W}_{>d}(T)| \leqslant 9 \ln \left( 3 e_{w^{(\alpha)}(T)}^{(\alpha)} K/d \right) K^{\frac{5}{2}} d^{-\frac{1}{2}}.$$

Informally, this lemma states that, in the rising concave setting, it cannot happen in too many windows that the expected rewards of arms which cross diverge significantly (i.e., more than $d$).

We use this fact to conclude the analysis. In particular, observe that we can always upper bound the regret suffered on a set of windows $\mathcal{W}$ as $R_{\boldsymbol{\nu}}(\pi, \mathcal{W}) \leqslant |\mathcal{W}| \max_{w \in \mathcal{W}} R_{\boldsymbol{\nu}}(\pi, \{w\})$. We use this to upper bound the regret suffered on both $\mathcal{W}_{\leqslant d}(T)$ and $\mathcal{W}_{>d}(T)$. In the first case, we observe that $|\mathcal{W}_{\leqslant d}(T)| \leqslant w^{(\alpha)}(T)$ and use the definition of $\mathcal{W}_{\leqslant d}(T)$ together with Lemma 4.1 to bound $\max_{w \in \mathcal{W}_{\leqslant d}(T)} R_{\boldsymbol{\nu}}(\texttt{RC-BE}(\alpha), \{w\})$. In the second case, we use Lemma 4.3 to upper bound $|\mathcal{W}_{>d}(T)|$ and Lemma 4.1 together with Lemma 4.2 to deal with $\max_{w \in \mathcal{W}_{>d}(T)} R_{\boldsymbol{\nu}}(\texttt{RC-BE}(\alpha), \{w\})$. These observations lead to the following result which is formally proven in Appendix C.

**Theorem 4.4** (Upper Bound for the Rising Concave Setting). *For all restless rising concave MABs $\boldsymbol{\nu}$, $\alpha \geqslant 1$, $T \in \mathbb{N}_{\geqslant 24}$, we have that:*

$$R_{\boldsymbol{\nu}}(\texttt{RC-BE}(\alpha), T) \leqslant 2^{15} \alpha^3 \left( \ln \left( \alpha K T^3 \right) \right)^{\frac{3}{2}} \left( (1+\sigma) K^3 T^{\frac{3/4\alpha}{1+\alpha}} + K^3 T^{\frac{5/4\alpha-1}{1+\alpha}} \Upsilon_{\boldsymbol{\nu}}(1, T) \right.$$

$$\left. + (1+\sigma) K T^{\frac{1+\alpha/2}{1+\alpha}} \right).$$

*In particular, for $\alpha' = 8/3$, we get:*

$$R_{\boldsymbol{\nu}}(\texttt{RC-BE}(\alpha'), T) = \widetilde{\mathcal{O}} \left( \sigma K^3 T^{\frac{6}{11}} + K^3 T^{\frac{7}{11}} \Upsilon_{\boldsymbol{\nu}}(1, T) + \sigma K T^{\frac{7}{11}} \right).$$

*Furthermore, for*

$$\alpha'' = \frac{8 - 4 \log_T \left( \frac{K^2 V_T}{1+\sigma} \right)}{3 + 4 \log_T \left( \frac{K^2 V_T}{1+\sigma} \right)},$$

*under the additional assumptions $\boldsymbol{\nu} \in \mathcal{E}_c^{\sigma}(T, V_T)$,*

$$T \geqslant \max \begin{cases} (1+\sigma)^{4/3} K^{-8/3} V_T^{-4/3} + 1 \\ (1+\sigma)^{-8/5} K^{16/5} V_T^{8/5} \end{cases},$$

*we get:*

$$R_{\boldsymbol{\nu}}(\texttt{RC-BE}(\alpha''), T) = \widetilde{\mathcal{O}} \left( \sigma^{\frac{14}{11}} K^{\frac{27}{11}} T^{\frac{6}{11}} V_T^{-\frac{3}{11}} + \sigma^{\frac{9}{11}} K^{\frac{15}{11}} T^{\frac{7}{11}} V_T^{\frac{2}{11}} \right).$$

By looking at the algorithm and at Theorem 4.4, we observe how by selecting $\alpha = 8/3$, we achieve a regret of order $\widetilde{\mathcal{O}}(T^{7/11})$ without the knowledge of either the total variation $V_T$ or the learning horizon $T$, making it an anytime algorithm, at the price of a worse dependence on $K$ and $V_T$. This result shows that the regret minimization problem in rising concave MABs is indeed easier w.r.t. general restless MABs with bounded variation (Besbes et al., 2014) and rising MABs. Indeed, the regret $\widetilde{\mathcal{O}}(T^{7/11})$ in our upper bound is smaller than that of the lower bound for restless MABs with bounded variation (Besbes et al., 2014, Theorem 1) and rising MABs (Theorem 3.2), i.e., $\Omega(T^{2/3})$.

## 5 Numerical Simulations

In this section, we present the results of numerical simulation of $\texttt{RC-BE}(\alpha)$ compared to state-of-the-art algorithms for restless, restless rising concave, and stationary MABs.[9]

**Baselines.** We consider the baseline algorithms: $\texttt{Rexp3}$ (Besbes et al., 2014), an algorithm for restless MABs based on a variation budget; $\texttt{R-less-UCB}$ (Metelli et al., 2022), an algorithm for restless rising concave MABs; and $\texttt{UCB1}$ (Auer et al., 2002a; Bubeck, 2010), one of the most effective

---

[9]Additional simulations are reported in Appendix E. The code to reproduce the results is available at https://github.com/m1gwings/rcbealpha-experiments.

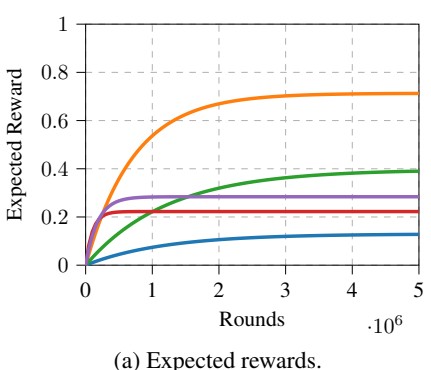

(a) Expected rewards.

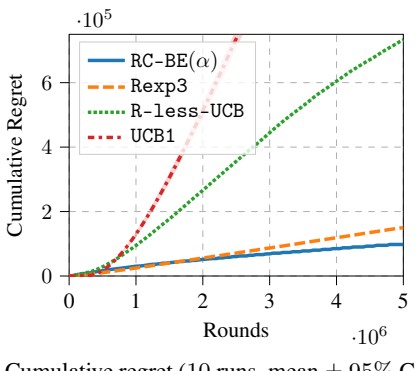

(b) Cumulative regret (10 runs, mean $\pm$ 95% C.I.).

Figure 3: Instance and results of the experimental validation.

algorithms for stationary MABs. The choices of the parameters of the algorithms are reported in Appendix E.

**Setting.** The algorithms are evaluated for $T = 5 \cdot 10^6$ rounds on synthetic instances with $K = 5$ arms. The stochasticity is realized by adding Gaussian noise with standard deviation $\sigma = 0.1$. The curves of the expected rewards have the functional form $f(t) = c(1 - \exp(-sat/T))$ for $t \in [\![T]\!]$ where $a, c \in (0, 1]$, $s = 50$, and are reported in Figure 3a. We compare the algorithms in terms of empirical cumulative regret $\widehat{R}_{\boldsymbol{\nu}}(\pi, t)$ which is the empirical counterpart of the expected cumulative regret $R_{\boldsymbol{\nu}}(\pi, t)$ at round $t$ averaged over multiple independent runs. In each simulation, the parameter $\alpha$ of RC-BE($\alpha$) is set to $\alpha = 8/3$, as suggested by Theorem 4.4.

**Results.** The empirical cumulative regret suffered by the algorithms is shown in Figure 3b. We observe that RC-BE($\alpha$) is the algorithm that achieves the lowest regret at the horizon. UCB1 has the lowest regret in the first rounds, afterwards its regret starts increasing when the optimal arm changes. This is consistent with the fact that we are violating the stationarity assumption on which the algorithm relies. Rexp3 is an algorithm which restarts at a fixed frequency. In particular, the number of restarts has order $T^{1/3}$. Thus, in this simulation, there are $\approx 10^2$ restarts, and, by looking at the figure, it is not possible to appreciate the behavior of the algorithm between one restart and the next. For this reason, Rexp3 shows a cumulative regret which increases linearly. This is consistent with the fact that the algorithm is not anytime. R-less-UCB, consistently with its theoretical guarantees, shows a sublinear growth of the cumulative regret. Its estimator relies on a rested model of the evolution of the expected rewards of the arms, penalizing the empirical performance.

## 6 Discussion and Conclusions

In this paper, we studied the restless rising and rising concave MABs, where the expected rewards of the arms are non-decreasing and non-decreasing concave in the number of played rounds, respectively. We derived lower bounds to the expected cumulative regret in both settings. The lower bound in the rising setting has order $\Omega(T^{2/3})$ and implies that the non-decreasing expected reward assumption does not simplify the learning problem w.r.t. the general restless setting with bounded variation, and so that all the algorithms which are optimal for the general setting are optimal also in this special subclass, closing in this way the gap present in the literature. Thus, for the rising setting, we provided a positive answer to our **Question 1** and a negative answer to our **Question 2**. The lower bound in the rising concave setting has order $\Omega(T^{3/5})$ and implies that rising concave MABs represent a statistically harder problem w.r.t. stationary MABs. After having presented two statistical barriers for these settings, we developed a learning algorithm with the goal of exploiting the more structured model of rising concave MABs. To this end, we designed RC-BE($\alpha$), and we derived an upper bound to its expected regret of order $\widetilde{\mathcal{O}}(T^{7/11})$. This result implies that the non-decreasing expected reward assumption, together with the concave expected reward assumption, simplifies the learning problem w.r.t. the same setting without concavity. Thus, for the rising concave setting, we provided a positive answer for both **Question 1** and **Question 2**. The natural future research direction includes closing the gap in rising concave MABs which is now only $7/11 - 3/5 = 2/55$ in the exponent of $T$.

## Acknowledgments

Funded by the European Union – Next Generation EU within the project NRPP M4C2, Investment 1.3 DD. 341 – 15 March 2022 – FAIR – Future Artificial Intelligence Research – Spoke 4 – PE00000013 – D53C22002380006.

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

# A    Related Works

**Restless Bandits.** In the original *restless* MAB setting, introduced by Tekin and Liu (2012), the evolution of the expected reward of each arm was described by a Markov chain. Several algorithms have been proposed to deal with this new framework, e.g., `Restless-UCB` (Wang et al., 2020), which relies on the optimistic estimation of the transition kernel of the underlying chain. Over time, the term restless acquired a broader meaning, encompassing all bandits in which the expected reward changes as time passes. Such arbitrary evolution can be described by a function that maps each round to the expected reward of a given arm. This is the type of restless bandit we target in this work. There are two families of methods to tackle restless MABs: *passive* (e.g., Garivier and Moulines, 2011; Besbes et al., 2014; Auer et al., 2019; Trovò et al., 2020) and *active* (e.g., Liu et al., 2018; Besson et al., 2022; Cao et al., 2019). Passive methods base their estimates on the recent feedback, forgetting obsolete observations. Active methods try to detect the changes in arms' expected rewards and use only the observations gathered after the last change. Among the most common passive approaches we find methods based on discounted rewards, e.g., `D-UCB` (Garivier and Moulines, 2011), or adaptive sliding window, e.g., `SW-UCB` (Garivier and Moulines, 2011). Both algorithms suffer a $\widetilde{\mathcal{O}}(T^{1/2})$ regret in the setting in which expected rewards change abruptly a fixed number of times over the time horizon, and such number is known. Auer et al. (2019) obtained a similar result in the same setting, without knowing the number of changes, by resorting on the doubling trick (Besson and Kaufmann, 2018). Another common setting is the one that allows the expected rewards to evolve arbitrarily, with the only constraint that the maximum absolute difference between the expected rewards of an arm in one round and the next, summed over the time horizon, is smaller than or equal to a variation budget $V_T$ (Besbes et al., 2014). The `Rexp3` algorithm (Besbes et al., 2014), a modification of the `Exp3` (Auer et al., 2002b) policy, originally designed for adversarial MABs, shows a regret bound of $\mathcal{O}(T^{2/3})$ under the knowledge of the variation budget $V_T$. The need for the knowledge of such quantity has been removed by Chen et al. (2019) by means of the doubling trick. In (Trovò et al., 2020), an approach which combines a Thompson-Sampling-like algorithm with a sliding window, shows theoretical guarantees in both the abruptly and smoothly changing settings.

**Rising Bandits.** *Rising concave* MABs have been introduced in the deterministic setting by Heidari et al. (2016) and Li et al. (2020), where the rewards observed by the agent in each round are not affected by noise. In their formulation of the problem, the rewards of an arm are non-decreasing in the number of times such an arm has been pulled and satisfy the *decreasing marginal return* assumption, i.e., the increment in the reward observed between one pull and the next of the same arm is non-increasing in the number of pulls. The online algorithm designed by Heidari et al. (2016) to minimize the regret relies on an optimistic estimate of the cumulative reward that can be obtained by pulling a given arm. Indeed, in this setting, Heidari et al. (2016) show that the optimal policy consists of repeatedly pulling the arm with the highest cumulative reward over the horizon. Li et al. (2020) use the rising concave MAB framework to model the problem of parameter optimization in machine learning and design an algorithm based on iterative elimination of unpromising arms that has good empirical performance. Cella et al. (2021) consider a setting in which the reward is increasing in expectation and the observations are affected by noise. However, in their framework, the expected rewards are constrained to follow a specific parametric form known to the agent. The authors analyze the setting under both the regret minimization and best arm identification frameworks. Anyway, the given parametric form makes this setting not applicable to an arbitrary expected reward evolution that satisfies the non-decreasing assumption. Recently, a surge of approaches has been designed for addressing other learning problems in stochastic rising concave MABs, including regret minimization (e.g., Metelli et al., 2022) and best arm identification (e.g., Takemori et al., 2024; Mussi et al., 2024). Finally, Genalti et al. (2024a,b) proposes a novel framework that interpolates between rested and restless MABs, still assuming the rising concave condition.

# B    Lower Bounds

In this appendix, we provide the proofs of the results presented in Section 3 in the main paper.

## B.1    General Recipe for the Lower Bound

The goal of this section is to prove Lemma 3.1. Remember that we work with rewards which follow the distribution $\psi(\mu, \sigma)$ defined in Section 3.1. The result is obtained through techniques from the

adversarial literature in which the instance is also affected by randomness. Thus, we define two probability distributions over $[\![0, K]\!]^{\mathbb{N}_{\geqslant 1}}$, which induce probability distributions over the instances in $\mathcal{E}^\sigma_{\overline{\boldsymbol{\mu}}, \widetilde{\boldsymbol{\mu}}}$. In particular, let $\overline{\xi}, \widetilde{\xi} \in \Delta([\![0, K]\!])$ defined as:

$$\overline{\xi}(\{o\}) := \begin{cases} 0 & \text{if } o \in [\![K]\!], \\ 1 & \text{if } o = 0 \end{cases}$$

$$\widetilde{\xi}(\{o\}) := \begin{cases} \dfrac{1}{K} & \text{if } o \in [\![K]\!], \\ 0 & \text{if } o = 0 \end{cases}$$

for $o \in [\![0, K]\!]$. We can extend $\overline{\xi}$ and $\widetilde{\xi}$ to probability distributions over $[\![0, K]\!]^{\mathbb{N}_{\geqslant 1}}$ via infinite product (see Example 1.63 of Klenke 2020):

$$\overline{\tau}_w := \left( \otimes_{l=1}^{w-1} \widetilde{\xi} \right) \otimes \left( \otimes_{l=w}^{+\infty} \overline{\xi} \right) \text{ for all } w \in \mathbb{N}_{\geqslant 1},$$

$$\widetilde{\tau} := \otimes_{w \in \mathbb{N}_{\geqslant 1}} \widetilde{\xi}.$$

$\widetilde{\tau}$ models a random instance in which, in each window, we choose independently and uniformly one arm whose expected reward follows the modified trend, while the expected rewards of all the other arms follow the base trend. $\overline{\tau}_w$ instead models a random instance which behaves like $\widetilde{\tau}$ up to window $w \in \mathbb{N}_{\geqslant 1}$ (excluded); from window $w$ onward all arms follow the base trend. For technical reasons which will be clear in what follows, we need to build a probability space in which the randomness over the instance and the randomness over the rewards are unlinked. Observe that with the current construction this is not the case. Indeed, $\boldsymbol{X}$ is sampled from $\nu_{\boldsymbol{o}}^\sigma$, but $\boldsymbol{o}$ is also a random element. To this end, let $\boldsymbol{s} = (s_{i,t})_{i \in [\![K]\!], t \in \mathbb{N}_{\geqslant 1}} \sim \lambda := \otimes_{i \in [\![K]\!], t \in \mathbb{N}_{\geqslant 1}} \mathrm{Unif}(0, 1)$ where $\mathrm{Unif}(0, 1)$ is the uniform distribution with support $[0, 1]$. Then, we can redefine:

$$X_{i,t}(\boldsymbol{o}, \boldsymbol{s}) = 2\sigma \cdot \mathbf{1} \left[ s_{i,t} \leqslant \frac{1}{4} + \frac{\mu_{\boldsymbol{o}, i}(t)}{2\sigma} \right] - \frac{1}{2}\sigma,$$

where $\mu_{\boldsymbol{o}, i}(t)$ is defined in analogy to Equation (2). In this way, we moved the dependency from $\boldsymbol{o}$ inside the definition of the random variables, preserving their distributions. Indeed, once $\boldsymbol{o}$ is fixed, we have $X_{i,t} \sim \psi(\mu_{\boldsymbol{o}, i}(t), \sigma)$. For consistency with the notation, we introduce the random variables $\boldsymbol{O} = (O_w)_{w \in \mathbb{N}_{\geqslant 1}}$ where $O_w(\boldsymbol{o}) = o_w$. The probability distributions that we just defined, induce probability density functions over finite reward sequences taking into account the randomness both in the instance and in the rewards. In particular, let

$$\overline{p}_w(r_1, \ldots, r_T) := \mathop{\mathbb{P}}_{\substack{\boldsymbol{o} \sim \overline{\tau}_w \\ \boldsymbol{s} \sim \lambda}} [R_1 = r_1, \ldots, R_T = r_T],$$

$$\widetilde{p}_{w,i}(r_1, \ldots, r_T) := \mathop{\mathbb{P}}_{\substack{\boldsymbol{o} \sim \widetilde{\tau} \\ \boldsymbol{s} \sim \lambda}} [R_1 = r_1, \ldots, R_T = r_T \mid O_w = i],$$

for $w \in \mathbb{N}_{\geqslant 1}$, $i \in [\![K]\!]$, $r_1, \ldots, r_T \in \{-1/2\sigma, 3/2\sigma\}$. We use $\overline{p}_w$ and $\widetilde{p}_{w,i}$ to denote also all the conditional and marginal distributions; disambiguation happens through the arguments, e.g., $\overline{p}_w(r_{s_w} \mid r_1, \ldots, r_{s_w - 1})$.

To obtain the result, we use the following tools from information theory (Cover and Thomas, 2006).

**Definition B.1** ($L^1$ Distance of Two Discrete Probability Density Functions). *Let $p, q$ be two discrete probability density functions defined over the finite set $\mathcal{X}$, we define their $L^1$ distance as:*

$$\|p - q\|_1 := \sum_{x \in \mathcal{X}} |p(x) - q(x)|.$$

**Definition B.2** (Kullback-Leibler Divergence of Two Discrete Probability Density Functions). *Let $p, q$ be two discrete probability density functions defined over the finite set $\mathcal{X}$, we define their Kullback-Leibler divergence as:*

$$\mathrm{D}_{\mathrm{KL}}(p\|q) := \sum_{x \in \mathcal{X}} p(x) \log_2 \left( \frac{p(x)}{q(x)} \right).$$

*By extension, we define:*

$$\mathrm{D}_{\mathrm{KL}}(\nu\|\xi) := \mathrm{D}_{\mathrm{KL}}(p_\nu\|p_\xi)$$

*where $\nu, \xi$ are probability distributions with discrete support and $p_\nu, p_\xi$ are their corresponding discrete probability density functions.*

We now state and prove a generalization of Lemma A.1 in (Auer et al., 2002b) which we then use to derive Lemma 3.1.

**Lemma B.1.** *Let $w \in [\![w(T)]\!]$, $i \in [\![K]\!]$, $f : \{-1/2\sigma, 3/2\sigma\}^{\min\{e_w, T\}} \to [0, M]$ with $M \geqslant 0$. Then:*

$$
\mathop{\mathbb{E}}_{\substack{o \sim \widetilde{\tau} \\ s \sim \lambda}}[f(R_1, \ldots, R_{\min\{e_w, T\}}) \mid O_w = i] - \mathop{\mathbb{E}}_{\substack{o \sim \overline{\tau}_w \\ s \sim \lambda}}[f(R_1, \ldots, R_{\min\{e_w, T\}})]
$$

$$
\leqslant \frac{M}{\sqrt{2}}\sqrt{\ln(2)\sum_{t=s_w}^{\min\{e_w, T\}} D_{KL}(\psi(\overline{\mu}_w(t - s_w + 1), \sigma)\|\psi(\widetilde{\mu}_w(t - s_w + 1), \sigma))\mathop{\mathbb{P}}_{\substack{o \sim \overline{\tau}_w \\ s \sim \lambda}}[I_t = i]}. \quad (4)
$$

*Proof.* To simplify the notation, let $t_1 := s_w, t_2 := \min\{e_w, T\}$. The lhs of Equation (4) can be written as:

$$
\sum_{r_1, \ldots, r_{t_2} \in \{-1/2\sigma, 3/2\sigma\}} f(r_1, \ldots, r_{t_2})\,(\widetilde{p}_{w,i}(r_1, \ldots, r_{t_2}) - \overline{p}_w(r_1, \ldots, r_{t_2}))
$$

$$
\leqslant M \sum_{\substack{r_1, \ldots, r_{t_2} \in \{-1/2\sigma, 3/2\sigma\} \\ \text{s.t. } \widetilde{p}_{w,i}(r_1, \ldots, r_{t_2}) \geqslant \overline{p}_w(r_1, \ldots, r_{t_2})}} (\widetilde{p}_{w,i}(r_1, \ldots, r_{t_2}) - \overline{p}_w(r_1, \ldots, r_{t_2}))
$$

$$
= \frac{M}{2}\|\overline{p}_w(r_1, \ldots, r_{t_2}) - \widetilde{p}_{w,i}(r_1, \ldots, r_{t_2})\|_1, \quad (5)
$$

where line (5) can be found in (Chapter 11, Cover and Thomas, 2006). Again, from (Lemma 11.6.1 Cover and Thomas, 2006), we have that:

$$
\|\overline{p}_w(r_1, \ldots, r_{t_2}) - \widetilde{p}_{w,i}(r_1, \ldots, r_{t_2})\|_1^2 \leqslant 2\ln(2)D_{KL}(\overline{p}_w(r_1, \ldots, r_{t_2})\|\widetilde{p}_{w,i}(r_1, \ldots, r_{t_2})).
$$

From the chain rule of entropy:

$$
D_{KL}(\overline{p}_w(r_1, \ldots, r_{t_2})\|\widetilde{p}_{w,i}(r_1, \ldots, r_{t_2})) = \underbrace{\sum_{t=t_1}^{t_2} D_{KL}(\overline{p}_w(r_t \mid r_1, \ldots, r_{t-1})\|\widetilde{p}_{w,i}(r_t \mid r_1, \ldots, r_{t-1}))}_{(a)}
$$

$$
+ \underbrace{D_{KL}(\overline{p}_w(r_1, \ldots, r_{t_1-1})\|\widetilde{p}_{w,i}(r_1, \ldots, r_{t_1-1}))}_{(b)}.
$$

Because of how $\overline{\tau}_w$ and $\widetilde{\tau}$ are defined, we have that:

$$
\widetilde{p}_{w,i}(r_1, \ldots, r_{t_1-1}) = \overline{p}_w(r_1, \ldots, r_{t_1-1}) \quad \text{for all} \quad r_1, \ldots, r_{t_1-1} \in \{-1/2\sigma, 3/2\sigma\}
$$

and thus term (b) is 0 because of the properties of $D_{KL}(\cdot\|\cdot)$. To deal with term (a) we need to work on the expressions of $\widetilde{p}_{w,i}(r_t \mid r_1, \ldots, r_{t-1})$ and $\overline{p}_w(r_t \mid r_1, \ldots, r_{t-1})$ for $t \in [\![t_1, t_2]\!]$. First of all observe that the arm that the agent pulls at round $t$ is fully determined by the past sequence of observed rewards $r_1, \ldots, r_{t-1}$ since the policy $\pi$ is deterministic. As remarked in Section 2, we denote it through $\pi(t)$, omitting the dependence on $r_1, \ldots, r_{t-1}$. Now:[10]

$$
\widetilde{p}_{w,i}(r_1, \ldots, r_t) = \mathop{\mathbb{P}}_{\substack{o \sim \widetilde{\tau} \\ s \sim \lambda}}[R_1 = r_1, \ldots, R_t = r_t \mid O_w = i]
$$

$$
= \mathop{\mathbb{P}}_{\substack{o \sim \widetilde{\tau} \\ s \sim \lambda}}[X_{\pi(1),1} = r_1, \ldots, X_{\pi(t),t} = r_t \mid O_w = i]
$$

$$
= \mathop{\mathbb{P}}_{\substack{o \sim \widetilde{\tau} \\ s \sim \lambda}}[X_{\pi(1),1} = r_1, \ldots, X_{\pi(t-1),t-1} = r_{t-1} \mid O_w = i]
$$

$$
\cdot (\mathbf{1}[\pi(t) = i]\psi(r_t \mid \widetilde{\mu}_w(t - s_w + 1), \sigma)
$$

$$
+ \mathbf{1}[\pi(t) \neq i]\psi(r_t \mid \overline{\mu}_w(t - s_w + 1), \sigma)) \quad (6)
$$

$$
= \widetilde{p}_{w,i}(r_1, \ldots, r_{t-1})(\mathbf{1}[\pi(t) = i]\psi(r_t \mid \widetilde{\mu}_w(t - s_w + 1), \sigma)
$$

$$
+ \mathbf{1}[\pi(t) \neq i]\psi(r_t \mid \overline{\mu}_w(t - s_w + 1), \sigma)),
$$

---

[10] With slight abuse of notation, we will use the symbol $\psi(x \mid \mu, \sigma)$ to denote the p.d.f. associated to the distribution $\psi(\mu, \sigma)$.

where line (6) follows from the fact that, under the event $O_w = i$, $X_{\pi(t),t}$ is independent from $X_{\pi(1),1}, \ldots, X_{\pi(t-1),t-1}$ and follows distribution $\psi(\tilde{\mu}_w(t - s_w + 1), \sigma)$ if $\pi(t) = i$, $\psi(\overline{\mu}_w(t - s_w + 1), \sigma)$ otherwise. Thus, we conclude:

$$\tilde{p}_{w,i}(r_t \mid r_1, \ldots, r_{t-1}) = \mathbf{1}[\pi(t) = i]\psi(r_t \mid \tilde{\mu}_w(t - s_w + 1), \sigma) + \mathbf{1}[\pi(t) \neq i]\psi(r_t \mid \overline{\mu}_w(t - s_w + 1), \sigma).$$

From analogous calculations, it is possible to derive:

$$\overline{p}_w(r_t \mid r_1, \ldots, r_{t-1}) = \psi(r_t \mid \overline{\mu}_w(t - s_w + 1), \sigma).$$

Thanks to the last results and the definition of $\mathrm{D}_{\mathrm{KL}}(\cdot\|\cdot)$:

$$\mathrm{D}_{\mathrm{KL}}(\overline{p}_w(r_1, \ldots, r_{t_2})\|\tilde{p}_{w,i}(r_1, \ldots, r_{t_2})) = \sum_{t=t_1}^{t_2} \sum_{r_1, \ldots, r_t \in \{-1/2\sigma, 3/2\sigma\}} \overline{p}_w(r_1, \ldots, r_t)$$

$$\cdot \log_2 \left( \frac{\psi(r_t \mid \overline{\mu}_w(t - s_w + 1), \sigma)}{\mathbf{1}[\pi(t) = i]\psi(r_t \mid \tilde{\mu}_w(t - s_w + 1), \sigma) + \mathbf{1}[\pi(t) \neq i]\psi(r_t \mid \overline{\mu}_w(t - s_w + 1), \sigma)} \right)$$

$$= \sum_{t=t_1}^{t_2} \sum_{r_1, \ldots, r_{t-1} \in \{-1/2\sigma, 3/2\sigma\}} \overline{p}_w(r_1, \ldots, r_{t-1})\mathbf{1}[\pi(t) = i] \sum_{r_t \in \{-1/2\sigma, 3/2\sigma\}} \psi(r_t \mid \overline{\mu}_w(t - s_w + 1), \sigma)$$

$$\cdot \log_2 \left( \frac{\psi(r_t \mid \overline{\mu}_w(t - s_w + 1), \sigma)}{\psi(r_t \mid \tilde{\mu}_w(t - s_w + 1), \sigma)} \right)$$

$$= \sum_{t=t_1}^{t_2} \mathrm{D}_{\mathrm{KL}}(\psi(\overline{\mu}_w(t - s_w + 1), \sigma)\|\psi(\tilde{\mu}_w(t - s_w + 1), \sigma))$$

$$\cdot \sum_{r_1, \ldots, r_{t-1} \in \{-1/2\sigma, 3/2\sigma\}} \overline{p}_w(r_1, \ldots, r_{t-1})\mathbf{1}[\pi(t) = i]$$

$$= \sum_{t=s_w}^{\min\{e_w, T\}} \mathrm{D}_{\mathrm{KL}}(\psi(\overline{\mu}_w(t - s_w + 1), \sigma)\|\psi(\tilde{\mu}_w(t - s_w + 1), \sigma)) \mathop{\mathbb{P}}_{\substack{o \sim \overline{\tau}_w \\ s \sim \lambda}}[I_t = i].$$

The lemma follows by chaining the results. $\qquad\square$

We are ready to prove Lemma 3.1.

**Lemma 3.1** (General Lower Bound). *Under the assumption that $\overline{\mu}_w(t) \leqslant \tilde{\mu}_w(t)$ for all $w \in \mathbb{N}_{\geqslant 1}$, $t \in [\![\Delta_w]\!]$, for any deterministic policy $\pi$, subgaussian parameter $\sigma \geqslant 1$, and learning horizon $T \in \mathbb{N}_{\geqslant 1}$, it holds that:*

$$\sup_{\boldsymbol{\nu} \in \mathcal{E}_{\overline{\boldsymbol{\mu}}, \tilde{\boldsymbol{\mu}}}^{\sigma}} R_{\boldsymbol{\nu}}(\pi, T) \geqslant \sum_{w=1}^{w(T)} \left( 1 - \frac{1}{K} - \sqrt{\frac{\ln(2)D_w^{\overline{\boldsymbol{\mu}}, \tilde{\boldsymbol{\mu}}, T, \sigma}}{2K}} \right) A_w^{\overline{\boldsymbol{\mu}}, \tilde{\boldsymbol{\mu}}, T}, \tag{3}$$

*where:*

$$D_w^{\overline{\boldsymbol{\mu}}, \tilde{\boldsymbol{\mu}}, T, \sigma} := \sum_{t=s_w}^{\min\{e_w, T\}} \mathrm{D}_{\mathrm{KL}}(\psi(\overline{\mu}_w(t - s_w + 1), \sigma)\|\psi(\tilde{\mu}_w(t - s_w + 1), \sigma)),$$

$$A_w^{\overline{\boldsymbol{\mu}}, \tilde{\boldsymbol{\mu}}, T} := \sum_{t=s_w}^{\min\{e_w, T\}} (\tilde{\mu}_w(t - s_w + 1) - \overline{\mu}_w(t - s_w + 1)),$$

*for all $w \in [\![w(T)]\!]$, with $\mathrm{D}_{\mathrm{KL}}(\cdot\|\cdot)$ being the Kullback-Leibler divergence of the two distributions (formally defined in Appendix B).*

*Proof.* For $\boldsymbol{o} \in [\![0, K]\!]^{\mathbb{N}_{\geqslant 1}}$, $t \in [\![T]\!]$, let $i_{\boldsymbol{o},t}^* \in \arg\max_{i \in [\![K]\!]} \mu_{\boldsymbol{o},i}(t)$. Then:

$$\sup_{\boldsymbol{\nu} \in \mathcal{E}_{\overline{\boldsymbol{\mu}}, \tilde{\boldsymbol{\mu}}}^{\sigma}} R_{\boldsymbol{\nu}}(\pi, T) = \sup_{\boldsymbol{o} \in [\![0, K]\!]^{\mathbb{N}_{\geqslant 1}}} \mathop{\mathbb{E}}_{\boldsymbol{s} \sim \lambda} \left[ \sum_{t=1}^{T} \left( \mu_{\boldsymbol{o}, i_{\boldsymbol{o},t}^*}(t) - \mu_{\boldsymbol{o}, I_t}(t) \right) \right]$$

$$\geqslant \mathbb{E}_{\substack{\boldsymbol{o}\sim\widetilde{\tau} \\ \boldsymbol{s}\sim\lambda}}\left[\sum_{t=1}^{T}\left(\mu_{\boldsymbol{o},i^{*}_{\boldsymbol{o},t}}(t)-\mu_{\boldsymbol{o},I_{t}}(t)\right)\right].$$

Under the assumption $\widetilde{\mu}_{w}(t)\geqslant\overline{\mu}_{w}(t)$ for all $w\in\mathbb{N}_{\geqslant 1}$, $t\in[\![\Delta_{w}]\!]$, we have:

$$\mu_{\boldsymbol{o},i^{*}_{\boldsymbol{o},t}}(t)-\mu_{\boldsymbol{o},I_{t}}(t)=\mathbf{1}[O_{w(t)}\neq 0,O_{w(t)}\neq I_{t}](\widetilde{\mu}_{w(t)}(t-s_{w(t)}+1)-\overline{\mu}_{w(t)}(t-s_{w(t)}+1)).$$

Then, observing that $O_{w}=0$ has probability $0$ under $\widetilde{\tau}$:

$$\sup_{\boldsymbol{\nu}\in\mathcal{E}^{\sigma}_{\overline{\mu},\widetilde{\mu}}}R_{\boldsymbol{\nu}}(\pi,T)\geqslant\sum_{w=1}^{w(T)}\sum_{t=s_{w}}^{\min\{e_{w},T\}}(\widetilde{\mu}_{w}(t-s_{w}+1)-\overline{\mu}_{w}(t-s_{w}+1))\mathbb{E}_{\substack{\boldsymbol{o}\sim\widetilde{\tau} \\ \boldsymbol{s}\sim\lambda}}[\mathbf{1}[O_{w}\neq I_{t}]]$$

$$=\sum_{w=1}^{w(T)}\sum_{t=s_{w}}^{\min\{e_{w},T\}}(\widetilde{\mu}_{w}(t-s_{w}+1)-\overline{\mu}_{w}(t-s_{w}+1))\sum_{i\in[\![K]\!]}\mathbb{E}_{\substack{\boldsymbol{o}\sim\widetilde{\tau} \\ \boldsymbol{s}\sim\lambda}}[\mathbf{1}[I_{t}\neq i,O_{w}=i]]$$

$$=\sum_{w=1}^{w(T)}\sum_{t=s_{w}}^{\min\{e_{w},T\}}(\widetilde{\mu}_{w}(t-s_{w}+1)-\overline{\mu}_{w}(t-s_{w}+1))\sum_{i\in[\![K]\!]}\mathbb{P}_{\substack{\boldsymbol{o}\sim\widetilde{\tau} \\ \boldsymbol{s}\sim\lambda}}[O_{w}=i]\frac{\mathbb{E}_{\substack{\boldsymbol{o}\sim\widetilde{\tau} \\ \boldsymbol{s}\sim\lambda}}[\mathbf{1}[I_{t}\neq i,O_{w}=i]]}{\mathbb{P}_{\substack{\boldsymbol{o}\sim\widetilde{\tau} \\ \boldsymbol{s}\sim\lambda}}[O_{w}=i]}$$

$$=\sum_{w=1}^{w(T)}\sum_{t=s_{w}}^{\min\{e_{w},T\}}(\widetilde{\mu}_{w}(t-s_{w}+1)-\overline{\mu}_{w}(t-s_{w}+1))\frac{1}{K}\sum_{i\in[\![K]\!]}\mathbb{E}_{\substack{\boldsymbol{o}\sim\widetilde{\tau} \\ \boldsymbol{s}\sim\lambda}}[\mathbf{1}[I_{t}\neq i\,]\,|\,O_{w}=i]$$

$$=\sum_{w=1}^{w(T)}\sum_{t=s_{w}}^{\min\{e_{w},T\}}(\widetilde{\mu}_{w}(t-s_{w}+1)-\overline{\mu}_{w}(t-s_{w}+1))\frac{1}{K}\sum_{i\in[\![K]\!]}\left(1-\mathbb{E}_{\substack{\boldsymbol{o}\sim\widetilde{\tau} \\ \boldsymbol{s}\sim\lambda}}[\mathbf{1}[I_{t}=i\,]\,|\,O_{w}=i]\right)$$

$$\geqslant\sum_{w=1}^{w(T)}\sum_{t=s_{w}}^{\min\{e_{w},T\}}(\widetilde{\mu}_{w}(t-s_{w}+1)-\overline{\mu}_{w}(t-s_{w}+1))\frac{1}{K}\sum_{i\in[\![K]\!]}\left(1-\mathbb{E}_{\substack{\boldsymbol{o}\sim\overline{\tau}_{w} \\ \boldsymbol{s}\sim\lambda}}[\mathbf{1}[I_{t}=i]]\right.$$

$$\left.-\frac{1}{\sqrt{2}}\sqrt{\ln(2)\sum_{t'=s_{w}}^{\min\{e_{w},T\}}\mathrm{D}_{\mathrm{KL}}(\psi(\overline{\mu}_{w}(t'-s_{w}+1),\sigma)\|\psi(\widetilde{\mu}_{w}(t'-s_{w}+1),\sigma))\mathop{\mathbb{P}}_{\substack{\boldsymbol{o}\sim\overline{\tau}_{w} \\ \boldsymbol{s}\sim\lambda}}[I_{t'}=i]}\right)$$

$$\tag{7}$$

$$\geqslant\sum_{w=1}^{w(T)}\sum_{t=s_{w}}^{\min\{e_{w},T\}}(\widetilde{\mu}_{w}(t-s_{w}+1)-\overline{\mu}_{w}(t-s_{w}+1))\left(1-\frac{1}{K}\right.$$

$$\left.-\frac{1}{K}\cdot\frac{\sqrt{K}}{\sqrt{2}}\sqrt{\ln(2)\sum_{t'=s_{w}}^{\min\{e_{w},T\}}\mathrm{D}_{\mathrm{KL}}(\psi(\overline{\mu}_{w}(t'-s_{w}+1),\sigma)\|\psi(\widetilde{\mu}_{w}(t'-s_{w}+1),\sigma))}\right),\tag{8}$$

where line (7) follows from Lemma B.1 with $f$ corresponding to the function from the observed rewards to the arm $I_{t}$ pulled in round $t$, which is well defined for deterministic policies, and line (8) follows from Cauchy-Schwarz inequality applied to a vector of $K$ ones and the vector of the terms under square root. The result follows from the definitions of $D^{\overline{\mu},\widetilde{\mu},T,\sigma}_{w}$ and $A^{\overline{\mu},\widetilde{\mu},T}_{w}$. □

## B.2 Specializing the Lower Bound for the Rising Setting

The goal of this section is to prove Theorem 3.2.

**Theorem 3.2** (Lower Bound for the Rising Setting). *For any deterministic policy $\pi$, subgaussian parameter $\sigma\geqslant 1$, and learning horizon $T\in\mathbb{N}_{\geqslant 1}$, $T\geqslant\sigma^{2}K\min\{1,V_{T}\}^{-2}$, it holds that:*

$$\sup_{\boldsymbol{\nu}\in\mathcal{E}^{\sigma}_{\mathrm{r}}(T,V_{T})}R_{\boldsymbol{\nu}}(\pi,T)\geqslant\frac{1}{64}\sigma^{\frac{2}{3}}T^{\frac{2}{3}}K^{\frac{1}{3}}\min\{1,V_{T}\}^{\frac{1}{3}}.$$

*Proof.* First of all, we need to formally define the sequences of window widths, base, and modified trends. Let $\Delta_{\mathrm{r},w} = \Delta_{\mathrm{r}} := \lfloor \sigma^{2/3} T^{2/3} K^{1/3} \min\{1, V_T\}^{-2/3} \rfloor$ and:

$$\overline{\mu}_{\mathrm{r},w}(t) := \begin{cases} \varepsilon_{\mathrm{r}}(w-1) & \text{if } w \leqslant w(T) \\ \varepsilon_{\mathrm{r}} w(T) & \text{if } w > w(T) \end{cases},$$

$$\widetilde{\mu}_{\mathrm{r},w}(t) := \begin{cases} \varepsilon_{\mathrm{r}} w & \text{if } w \leqslant w(T) \\ \varepsilon_{\mathrm{r}} w(T) & \text{if } w > w(T) \end{cases},$$

for all $w \in \mathbb{N}_{\geqslant 1}$ where $\varepsilon_{\mathrm{r}} := \frac{1}{4} \min\{1, V_T\}/w(T) > 0$. Observe that $\widetilde{\mu}_{\mathrm{r},w}(\Delta_{\mathrm{r}}) \leqslant \overline{\mu}_{\mathrm{r},w+1}(1)$ for all $w \in \mathbb{N}_{\geqslant 1}$, hence, for any choice of $\boldsymbol{o} \in [\![0, K]\!]^{\mathbb{N}_{\geqslant 1}}$, $\boldsymbol{\nu}_{\mathrm{r},\boldsymbol{o}}^{\sigma}$ satisfies Assumption 2.1. Furthermore, for all $\boldsymbol{o} \in [\![0, K]\!]^{\mathbb{N}_{\geqslant 1}}$, the expected rewards of the arms change at most between one window and the next, i.e., $w(T) - 1$ times in the learning horizon, and the magnitude of the increment is at most $2\varepsilon_{\mathrm{r}}$, thus:

$$\Upsilon_{\boldsymbol{\nu}_{\mathrm{r},\boldsymbol{o}}^{\sigma}}(1, T) \leqslant 2(w(T) - 1)\varepsilon_{\mathrm{r}} \leqslant V_T.$$

Hence $\mathcal{E}_{\overline{\boldsymbol{\mu}}_{\mathrm{r}}, \widetilde{\boldsymbol{\mu}}_{\mathrm{r}}}^{\sigma} \subseteq \mathcal{E}_{\mathrm{r}}^{\sigma}(T, V_T)$ indeed holds. Finally, it is easy to verify that $0 \leqslant \overline{\mu}_{\mathrm{r},w}(t) \leqslant \widetilde{\mu}_{\mathrm{r},w}(t) \leqslant 1$ for all $w \in \mathbb{N}_{\geqslant 1}$, $t \in [\![\Delta_{\mathrm{r}}]\!]$, so that the assumptions of Lemma 3.1 are satisfied. From Lemma D.1, we have that:

$$D_w^{\overline{\boldsymbol{\mu}}_{\mathrm{r}}, \widetilde{\boldsymbol{\mu}}_{\mathrm{r}}, T, \sigma} \leqslant \frac{2\varepsilon_{\mathrm{r}}^2}{\ln(2)\sigma^2} \Delta_{\mathrm{r}}.$$

The choice of $\Delta_{\mathrm{r},w} = \Delta_{\mathrm{r}}$ implies $\varepsilon_{\mathrm{r}} \leqslant \frac{1}{4} \sigma^{2/3} T^{-1/3} K^{1/3} \min\{1, V_T\}^{1/3}$ once we observe that $w(T) = \lceil T/\Delta_{\mathrm{r}} \rceil$. Then:

$$D_w^{\overline{\boldsymbol{\mu}}_{\mathrm{r}}, \widetilde{\boldsymbol{\mu}}_{\mathrm{r}}, T, \sigma} \leqslant \frac{1}{8} \frac{K}{\ln(2)} \text{ for all } w \in [\![w(T)]\!].$$

Thus, observing that $K \geqslant 2$, we have

$$1 - \frac{1}{K} - \sqrt{\frac{\ln(2) D_w^{\overline{\boldsymbol{\mu}}_{\mathrm{r}}, \widetilde{\boldsymbol{\mu}}_{\mathrm{r}}, T, \sigma}}{2K}} \geqslant \frac{1}{4}.$$

Since $A_w^{\overline{\boldsymbol{\mu}}_{\mathrm{r}}, \widetilde{\boldsymbol{\mu}}_{\mathrm{r}}, T} = \varepsilon_{\mathrm{r}}(\min\{e_w, T\} - s_w + 1)$, by plugging the previous results in Lemma 3.1, assuming that $T \geqslant \sigma^2 K \min\{1, V_T\}^{-2}$ which guarantees $T \geqslant \Delta_{\mathrm{r}}$, we have:

$$\sup_{\boldsymbol{\nu} \in \mathcal{E}_{\mathrm{r}}^{\sigma}(T, V_T)} R_{\boldsymbol{\nu}}(\pi, T) \geqslant \frac{1}{4} \varepsilon_{\mathrm{r}} T \geqslant \frac{1}{64} \sigma^{\frac{2}{3}} T^{\frac{2}{3}} K^{\frac{1}{3}} \min\{1, V_T\}^{\frac{1}{3}},$$

where the last step follows from the definition of $\varepsilon_{\mathrm{r}}$ and the fact that $\lfloor x \rfloor \geqslant x/2$ and $\lceil x \rceil \leqslant 2x$ for $x \geqslant 1$. $\qquad\square$

### B.3 Specializing the Lower Bound for the Rising Concave Setting

The goal of this section is to prove Theorem 3.3.

**Theorem 3.3** (Lower Bound for the Rising Concave Setting). *For any deterministic policy $\pi$, subgaussian parameter $\sigma \geqslant 1$, and learning horizon $T \in \mathbb{N}_{\geqslant 1}$, $T \geqslant 2^{10} \sigma^2 K \min\{1, V_T\}^{-2}$, it holds that:*

$$\sup_{\boldsymbol{\nu} \in \mathcal{E}_{\mathrm{c}}^{\sigma}(T, V_T)} R_{\boldsymbol{\nu}}(\pi, T) \geqslant 2^{-14} \sigma^{4/5} T^{\frac{3}{5}} K^{\frac{2}{5}} \min\{1, V_T\}^{\frac{1}{5}}.$$

*Proof.* First of all, we need to formally define the sequences of window widths, base, and modified trends. Let $N_{\mathrm{c}} := \lceil \sigma^{-2/5} T^{1/5} K^{-1/5} \min\{1, V_T\}^{2/5} \rceil$, $\Delta_{\mathrm{c},w} = \Delta_{\mathrm{c}} := \lceil T/N_{\mathrm{c}} \rceil$ for all $w \in \mathbb{N}_{\geqslant 1}$. Observe that $\Delta_{\mathrm{c}}$ is defined in such a way that $w(T) = \lceil T/\Delta_{\mathrm{c}} \rceil \leqslant \lceil T/(T/N_{\mathrm{c}}) \rceil = N_{\mathrm{c}}$. Furthermore, being $\sigma, K \geqslant 1$, we have:

$$N_{\mathrm{c}} \leqslant \left\lceil T^{\frac{1}{5}} \right\rceil \leqslant \lceil T \rceil = T,$$

so that $T/N_{\mathrm{c}} \geqslant 1$ and $\Delta_{\mathrm{c}} \leqslant 2T/N_{\mathrm{c}}$ since $\lceil x \rceil \leqslant 2x$ for $x \geqslant 1$. Let $m_0 := \frac{1}{4} \min\{1, V_T\}/T \in (0, 1)$, $m_w := (2N_{\mathrm{c}} - w)m_0/(2N_{\mathrm{c}})$ for $w \in [\![2N_{\mathrm{c}}]\!]$. $(m_w)_{w=0}^{2N_{\mathrm{c}}}$ are the slopes of the segments which

constitute the trends. Observe that $m_0 > m_1 > \cdots > m_{2N_c-1} > m_{2N_c} = 0$. We are ready to define the trends:

$$
\overline{\mu}_{c,w}(t) := \begin{cases} \Delta_c \displaystyle\sum_{l=1}^{w-1} m_{2l-1} + t m_{2w-1} & \text{if } w \leqslant w(T) \\ \Delta_c \displaystyle\sum_{l=1}^{w(T)} m_{2l-1} & \text{if } w > w(T) \end{cases},
$$

$$
\widetilde{\mu}_{c,w}(t) := \begin{cases} \Delta_c \displaystyle\sum_{l=1}^{w-1} m_{2l-1} + t m_{2w-2} + \left(t - \dfrac{\Delta_c}{2}\right)^+ (m_{2w} - m_{2w-2}) & \text{if } w \leqslant w(T) \\ \Delta_c \displaystyle\sum_{l=1}^{w(T)} m_{2l-1} & \text{if } w > w(T) \end{cases},
$$

for all $w \in \mathbb{N}_{\geqslant 1}$. In what follows, with a slight abuse of notation, we will regard $\overline{\mu}_{c,w}$ and $\widetilde{\mu}_{c,w}$ as defined on $[0, \Delta_c]$. Observe that, as we informally stated before, $\overline{\mu}_{c,w}(0) = \widetilde{\mu}_{c,w}(0)$, and $\overline{\mu}_{c,w}(\Delta_c) = \widetilde{\mu}_{c,w}(\Delta_c) = \overline{\mu}_{c,w+1}(0)$ for all $w \in \mathbb{N}_{\geqslant 1}$. Furthermore, it is easy to check that the slope of the second segment of the modified trend in a window is equal to the slope of the first segment of the modified trend in the next window. Thus, because of what we remarked when we informally introduced the construction, for any choice of $o \in [\![0, K]\!]^{\mathbb{N}_{\geqslant 1}}$, $\nu_{c,o}^{\sigma}$ satisfies Assumptions 2.1 and 2.2. Furthermore, in each window with index $w \in [\![w(T)]\!]$, the maximum increment of the expected reward of an arm, corresponds to the slope of the first half of the modified trend $m_{2w-2}$. Thus:

$$
\Upsilon_{\nu_{c,o}^{\sigma}}(1, T) \leqslant \Delta_c \sum_{w=1}^{w(T)} m_{2w-2} \leqslant \Delta_c \sum_{w=1}^{N_c} m_{2w-2} = \frac{\Delta_c m_0}{2}(N_c + 1)
$$

$$
\leqslant \Delta_c m_0 N_c \leqslant 2\frac{T}{N_c} \cdot \frac{1}{4} \frac{\min\{1, V_T\}}{T} \cdot N_c = \frac{1}{2}\min\{1, V_T\} \leqslant V_T.
$$

Hence $\mathcal{E}_{\overline{\mu}_c, \widetilde{\mu}_c}^{\sigma} \subseteq \mathcal{E}_c^{\sigma}(T, V_T)$ indeed holds. Finally, by calculations analogous to what we did above to bound the cumulative increment, one can verify that:

$$
\Delta_c \sum_{w=1}^{w(T)} m_{2w-1} \leqslant \frac{1}{4}\min\{1, V_T\} \leqslant 1,
$$

which, together with the previous remarks, implies $0 \leqslant \overline{\mu}_{c,w}(t) \leqslant \widetilde{\mu}_{c,w}(t) \leqslant 1$ for all $w \in \mathbb{N}_{\geqslant 1}$, $t \in [\![\Delta_c]\!]$, so that the assumptions of Lemma 3.1 are satisfied. The maximum distance between the two trends in a window is attained for $t = \frac{\Delta_c}{2}$ and has value:

$$
\varepsilon_c := \widetilde{\mu}_{c,w}\left(\frac{\Delta_c}{2}\right) - \overline{\mu}_{c,w}\left(\frac{\Delta_c}{2}\right) = \frac{\Delta_c m_0}{4N_c} \leqslant \frac{1}{8}\frac{\min\{1, V_T\}}{N_c^2}.
$$

Hence, in virtue of Lemma D.1, we have:

$$
D_w^{\overline{\mu}_c, \widetilde{\mu}_c, T, \sigma} \leqslant \frac{2\varepsilon_c^2}{\ln(2)\sigma^2}\Delta_c = \frac{1}{16}\frac{\min\{1, V_T\}^2 T}{\ln(2)\sigma^2 N_c^5} \leqslant \frac{1}{16}\frac{K}{\ln(2)}
$$

for all $w \in [\![w(T)]\!]$. Thus, remembering that $K \geqslant 2$, we get:

$$
1 - \frac{1}{K} - \sqrt{\frac{\ln(2)D_w^{\overline{\mu}_c, \widetilde{\mu}_c, T, \sigma}}{2K}} \geqslant \frac{1}{4}.
$$

Now, let's lower bound the expression of $A_w^{\overline{\mu}_c, \widetilde{\mu}_c, T}$ for $w \in [\![w(T) - 1]\!]$:

$$
A_w^{\overline{\mu}_c, \widetilde{\mu}_c, T} = \sum_{t=1}^{\lfloor \frac{\Delta_c}{2} \rfloor} (m_{2w-2} - m_{2w-1})t + \sum_{t=\lfloor \frac{\Delta_c}{2} \rfloor + 1}^{\Delta_c} \left[ (m_{2w} - m_{2w-1})t - \frac{\Delta_c}{2}(m_{2w} - m_{2w-2}) \right]
$$

$$
= \frac{m_0}{2N_c}\left( \frac{\lfloor \frac{\Delta_c}{2} \rfloor (\lfloor \frac{\Delta_c}{2} \rfloor + 1)}{2} + \frac{(\Delta_c - \lfloor \frac{\Delta_c}{2} \rfloor - 1)(\Delta_c - \lfloor \frac{\Delta_c}{2} \rfloor)}{2} \right)
$$

$$\geqslant \frac{m_0}{4N_c}\left(\frac{5}{16}\Delta_c^2 - \frac{\Delta_c}{4}\right) \geqslant \frac{m_0}{4N_c}\left(\frac{5}{16}\Delta_c^2 - \frac{4}{16}\Delta_c^2\right) = \frac{m_0\Delta_c^2}{64N_c} \tag{9}$$

$$\geqslant \frac{m_0 T^2}{64N_c^3} = \frac{\min\{1, V_T\}T}{256N_c^3},$$

where line (9) follows from the fact that $\lfloor x \rfloor \geqslant x/2$ for $x \geqslant 1$ together with $\Delta_c/2 \geqslant 1$ being $T > N_c$ when $T \geqslant 4$ (which is guaranteed by the constraint on $T$), and that $x \leqslant x^2$ for $x \geqslant 1$. Finally, $T \geqslant 2^{10}\sigma^2 K \min\{1, V_T\}^{-2}$ guarantees $w(T) - 1 \geqslant N_c/4$, and thus, by Lemma 3.1 in conjunction with the results we just proved, we have:

$$\sup_{\boldsymbol{\nu}\in\mathcal{E}_c^\sigma(T,V_T)} R_{\boldsymbol{\nu}}(\pi, T) \geqslant \frac{1}{4}\sum_{w=1}^{w(T)} A_w^{\overline{\boldsymbol{\mu}}_c,\widetilde{\boldsymbol{\mu}}_c,T} \geqslant \frac{1}{4}\left(w(T)-1\right)\frac{\min\{1, V_T\}T}{256N_c^3}$$

$$\geqslant \frac{\min\{1, V_T\}T}{2^{12}N_c^2} \geqslant 2^{-14}\sigma^{4/5}T^{\frac{3}{5}}K^{\frac{2}{5}}\min\{1, V_T\}^{\frac{1}{5}}, \tag{10}$$

where line (10) follows from our choice of $N_c$ and from the fact that $\lceil x \rceil \leqslant 2x$ for $x \geqslant 1$. $\qquad\square$

## C  Upper Bound for the Rising Concave Setting

In this appendix, we provide the proofs of the results presented in Section 4 in the main paper.

### C.1  Additional notation

We begin by introducing the additional notation required for the analysis. Let:

$$\hat{S}_{i,w,d} := \sum_{t=s_w^{(\alpha)}}^{s_w^{(\alpha)}+d-1} \mathbf{1}[I_t = i]R_t, \qquad \widetilde{S}_{i,w,d} := \sum_{t=s_w^{(\alpha)}}^{s_w^{(\alpha)}+d-1} \mathbf{1}[I_t = i]\mu_i(t),$$

be respectively the *cumulative reward* and *cumulative expected reward* by $\texttt{RC-BE}(\alpha)$ for arm $i \in [\![K]\!]$ in the first $d \in [\![0, \Delta_w^{(\alpha)}]\!]$ rounds of window $w \in \mathbb{N}_{\geqslant 1}$. Let $N_w$ be the number of round-robin cycles of window $w \in \mathbb{N}_{\geqslant 1}$, where we also count the degenerate cycles in which we pull the only remaining alive arm $\hat{i}^*$. Let $t_{w,l}$ be the round in which the $l$-th round-robin cycle (with $l \in [\![N_w]\!]$) is started during window $w \in \mathbb{N}_{\geqslant 1}$. Analogously, let $N_{i,w}$ be the number of times arm $i \in [\![K]\!]$ is pulled in the $w$-th window (with $w \in \mathbb{N}_{\geqslant 1}$) and $t_{i,w,l}$ the round in which arm $i$ is pulled for the $l$-th time (with $l \in [\![N_{i,w}]\!]$) during window $w$. For simplicity in the notation, we define $d_{w,l} = t_{w,l} - s_w^{(\alpha)} + 1$ and $d_{i,w,l} = t_{i,w,l} - s_w^{(\alpha)} + 1$. Finally, we define the good events:

$$\mathcal{G}_{i,w,d,\delta} := \left\{\left|\hat{S}_{i,w,d} - \widetilde{S}_{i,w,d}\right| \leqslant \sigma\sqrt{2\Delta_w^{(\alpha)}\left(\ln\left(2K\Delta_w^{(\alpha)}\right) + \ln\left(\frac{1}{\delta}\right)\right)}\right\},$$

for $i \in [\![K]\!]$, $w \in \mathbb{N}_{\geqslant 1}$, $d \in [\![\Delta_w^{(\alpha)}]\!]$, $\delta \in (0, 1]$, and

$$\mathcal{G}_{w,\delta} = \bigcap_{\substack{i\in[\![K]\!] \\ d\in[\![\Delta_w^{(\alpha)}]\!]}} \mathcal{G}_{i,w,d,\delta}$$

for $i \in [\![K]\!]$, $\delta \in (0, 1]$.

### C.2  Concentration

We start the analysis with a concentration result for $\hat{S}_{i,w,d}$.

**Lemma C.1** (Concentration). *For every $w \in \mathbb{N}_{\geqslant 1}$, $\delta \in (0, 1]$, we have that:*

$$\mathop{\mathbb{P}}_{\boldsymbol{X}\sim\boldsymbol{\nu}}\left[\overline{\mathcal{G}_{w,\delta}}\right] \leqslant \delta.$$

*Proof.* For $i \in [\![K]\!]$, $d \in [\![0, \Delta_w^{(\alpha)}]\!]$, $\lambda \in \mathbb{R}$, let:

$$M_{i,w,d}(\lambda) := \exp\left(\lambda\left(\hat{S}_{i,w,d} - \tilde{S}_{i,w,d}\right)\right),$$

$$\mathcal{F}_{w,d} := \sigma\left(X_{1,1}, \ldots, X_{K,1}, \ldots, X_{1,s_w^{(\alpha)}+d-1}, \ldots, X_{K,s_w^{(\alpha)}+d-1}\right).$$

Let $t' := s_w^{(\alpha)} + d - 1$ to ease the notation. Observe that $I_{t'}$ is $\mathcal{F}_{w,d-1}$-measurable and that $X_{i,t'}$ is independent from $\mathcal{F}_{w,d-1}$. Furthermore, we can rewrite $\hat{S}_{i,w,d}$ as

$$\hat{S}_{i,w,d} = \sum_{t=s_w^{(\alpha)}}^{t'} \mathbf{1}[I_t = i]X_{i,t}.$$

Then:

$$
\begin{aligned}
\mathbb{E}_{\boldsymbol{X}\sim\boldsymbol{\nu}}\left[M_{i,w,d}(\lambda) \mid \mathcal{F}_{w,d-1}\right] &= M_{i,w,d-1}(\lambda)\,\mathbb{E}_{\boldsymbol{X}\sim\boldsymbol{\nu}}\Big[\mathbf{1}[I_{t'} = i]\exp\left(\lambda\left(X_{i,t'} - \mu_i(t')\right)\right) \\
&\quad + 1 - \mathbf{1}[I_{t'} = i] \mid \mathcal{F}_{w,d-1}\Big] \\
&\leqslant M_{i,w,d-1}(\lambda)\exp\left(\mathbf{1}[I_{t'} = i]\frac{\lambda^2\sigma^2}{2}\right) \leqslant M_{i,w,d-1}(\lambda)\exp\left(\frac{\lambda^2\sigma^2}{2}\right),
\end{aligned}
$$

where in the last line we use the properties of conditional expectation (Klenke, 2020) and the sub-gaussianity of $X_{i,t'}$. Thus, by induction:

$$\mathbb{E}_{\boldsymbol{X}\sim\boldsymbol{\nu}}\left[M_{i,w,d}(\lambda)\right] \leqslant \exp\left(d\frac{\lambda^2\sigma^2}{2}\right) \leqslant \exp\left(\Delta_w^{(\alpha)}\frac{\lambda^2\sigma^2}{2}\right).$$

Then, thanks to Markov inequality, for every $\varepsilon \in \mathbb{R}$:

$$
\begin{aligned}
\mathbb{P}_{\boldsymbol{X}\sim\boldsymbol{\nu}}\left[\hat{S}_{i,w,d} - \tilde{S}_{i,w,d} > \varepsilon\right] &= \mathbb{P}_{\boldsymbol{X}\sim\boldsymbol{\nu}}\left[M_{i,w,d}(\lambda) > \exp(\lambda\varepsilon)\right] \\
&\leqslant \mathbb{E}_{\boldsymbol{X}\sim\boldsymbol{\nu}}\left[M_{i,w,d}(\lambda)\right]\exp(-\lambda\varepsilon) \\
&\leqslant \exp\left(\lambda^2\frac{\Delta_w^{(\alpha)}\sigma^2}{2} - \lambda\varepsilon\right).
\end{aligned}
$$

By choosing $\varepsilon = \sigma\sqrt{2\Delta_w^{(\alpha)}\left(\ln\left(2K\Delta_w^{(\alpha)}\right) + \ln\left(\frac{1}{\delta}\right)\right)}$, $\lambda = \frac{\varepsilon}{\Delta_w^{(\alpha)}\sigma^2}$, we get:

$$\mathbb{P}_{\boldsymbol{X}\sim\boldsymbol{\nu}}\left[\hat{S}_{i,w,d} - \tilde{S}_{i,w,d} > \varepsilon\right] \leqslant \frac{\delta}{2K\Delta_w^{(\alpha)}}.$$

An analogous bound holds for

$$\mathbb{P}_{\boldsymbol{X}\sim\boldsymbol{\nu}}\left[\tilde{S}_{i,w,d} - \hat{S}_{i,w,d} > \varepsilon\right].$$

Then, thanks to a union bound,

$$\mathbb{P}_{\boldsymbol{X}\sim\boldsymbol{\nu}}\left[\overline{\mathcal{G}_{i,w,d,\delta}}\right] \leqslant \frac{\delta}{K\Delta_w^{(\alpha)}}.$$

Finally:

$$\mathbb{P}_{\boldsymbol{X}\sim\boldsymbol{\nu}}\left[\overline{\mathcal{G}_{w,\delta}}\right] \leqslant \sum_{i\in[\![K]\!]}\sum_{d\in[\![\Delta_w^{(\alpha)}]\!]}\mathbb{P}_{\boldsymbol{X}\sim\boldsymbol{\nu}}\left[\overline{\mathcal{G}_{i,w,d,\delta}}\right] \leqslant \delta.$$

$\square$

## C.3 Proof of Lemma 4.1

The goal of this section is to prove Lemma 4.1. To this end, we need several intermediate results. We start by proving that $\mathcal{I}_w^\times$ is indeed well-defined.

**Lemma C.2.** *Let $i_w^*, j_w^* \in \mathcal{I}_w^*$, then $i_w^* \,_w\!\times j_w^*$.*

*Proof.* If $\mu_{i_w^*}(t') = \mu_{j_w^*}(t')$ for some $t' \in [\![s_w^{(\alpha)}, e_w^{(\alpha)}]\!]$, then it must be $i_w^* {}_w\times j_w^*$. Thus, assume $\mu_{i_w^*}(t') < \mu_{j_w^*}(t')$ for some $t' \in [\![s_w^{(\alpha)}, e_w^{(\alpha)}]\!]$. If $i_w^*$ and $j_w^*$ do not cross, then

$$\mu_{i_w^*}(t) < \mu_{j_w^*}(t) \text{ for all } t \in [\![s_w^{(\alpha)}, e_w^{(\alpha)}]\!]$$

which is a contradiction with the fact that $i_w^* \in \mathcal{I}_w^*$. $\qquad\square$

We now prove a very useful property of ${}_w\times^+$.

**Lemma C.3.** *Let $i, j, k \in [\![K]\!]$. If $i {}_w\times^+ j$ and there exists $t' \in [\![s_w^{(\alpha)}, e_w^{(\alpha)}]\!]$ such that $\mu_i(t') \leqslant \mu_k(t') \leqslant \mu_j(t')$, then $k \in [i]_{w\times^+}$.*

*Proof.* If $\mu_k(t') = \mu_i(t')$ or $\mu_k(t') = \mu_j(t')$ then the statement is trivial. Consider $\mu_i(t') < \mu_k(t') < \mu_j(t')$. We proceed by contradiction. Assume that it is not true that $k {}_w\times^+ i$. Let $\mathcal{I}_1 = \left\{ l \in [i]_{w\times^+} \text{ s.t. } \mu_l(t') < \mu_k(t') \right\}$ and $\mathcal{I}_2 = \left\{ l \in [i]_{w\times^+} \text{ s.t. } \mu_l(t') > \mu_k(t') \right\}$. Since $\mathcal{I}_1 \cup \mathcal{I}_2 \subseteq [i]_{w\times^+}$, $\mathcal{I}_1 \cap \mathcal{I}_2 = \{\}$, $\mathcal{I}_1, \mathcal{I}_2 \neq \{\}$ there must be $i_1 \in \mathcal{I}_1$, $i_2 \in \mathcal{I}_2$ such that $i_1 {}_w\times i_2$. But, since it is not true that $k {}_w\times^+ i$, it cannot be $k {}_w\times i_1$ nor $k {}_w\times i_2$. Thus it must be

$$\mu_{i_1}(t) < \mu_k(t) < \mu_{i_2}(t) \text{ for all } t \in [\![s_w^{(\alpha)}, e_w^{(\alpha)}]\!].$$

But this is absurd since $i_1 {}_w\times i_2$, concluding the proof. $\qquad\square$

This leads to the following corollary.

**Corollary C.4.** *Let $i \in \mathcal{I}_w^\times$, $j \notin \mathcal{I}_w^\times$, then:*

$$\mu_j(t) < \mu_i(t) \text{ for all } t \in [\![s_w^{(\alpha)}, e_w^{(\alpha)}]\!].$$

*Proof.* By contrapositive, if $\mu_j(t') \geqslant \mu_i(t')$ for some $t' \in [\![s_w^{(\alpha)}, e_w^{(\alpha)}]\!]$, then there exists $k \in \arg\max_{l \in [\![K]\!]} \mu_l(t')$ such that $\mu_i(t') \leqslant \mu_j(t') \leqslant \mu_k(t')$ and thus $j \in \mathcal{I}_w^\times$ by Lemma C.3. $\qquad\square$

We are ready to prove Lemma 4.1.

**Lemma 4.1.** *For all restless rising concave MABs $\boldsymbol{\nu}$, $\alpha \geqslant 1$, $w \in \mathbb{N}_{\geqslant 1}$ we have that:*

$$R_{\boldsymbol{\nu}}(\texttt{RC-BE}(\alpha), \{w\}) \leqslant \underbrace{3KB_w^{(\alpha)}}_{\text{Exploration}} + \underbrace{\Delta_w^{(\alpha)} d_w^*}_{\text{Commitment}}.$$

*Proof.* We start by proving that, under event $\mathcal{G}_{w,(2K\Delta_w^{(\alpha)})-1}$, at least one arm in $\mathcal{I}_w^\times$ is always alive in each round-robin cycle. We need to consider all the eliminations which happen at the end of a round-robin cycle, except for the last, in which eliminations are irrelevant (remember that the window ends at the end of the last round-robin cycle and the algorithm is restarted). To this end, let $n \in [\![N_w - 1]\!]$. For an arm $i \in [\![K]\!]$, to eliminate an arm $j \in [\![K]\!]$ at the end of the $n$-th round-robin cycle, it must be:

$$\hat{S}_{i,w,d_{w,n+1}-1} > \hat{S}_{j,w,d_{w,n+1}-1} + B_w^{(\alpha)}$$

which, under event $\mathcal{G}_{w,(2K\Delta_w^{(\alpha)})-1}$, implies

$$\widetilde{S}_{i,w,d_{w,n+1}-1} + 4\sigma\sqrt{\Delta_w^{(\alpha)} \ln\left(2K\Delta_w^{(\alpha)}\right)} > \widetilde{S}_{j,w,d_{w,n+1}-1} + B_w^{(\alpha)}$$

if and only if

$$\sum_{l=1}^n \left[ \mu_i(t_{w,l}) + \mu_i(t_{i,w,l}) - \mu_i(t_{w,l}) \right] + 4\sigma\sqrt{\Delta_w^{(\alpha)} \ln\left(2K\Delta_w^{(\alpha)}\right)}$$

$$> \sum_{l=1}^n \left[ \mu_j(t_{w,l}) + \mu_j(t_{j,w,l}) - \mu_j(t_{w,l}) \right] + B_w^{(\alpha)}$$

which implies, being the instance rising:

$$\sum_{l=1}^{n} \mu_i(t_{w,l}) + 1 + 4\sigma\sqrt{\Delta_w^{(\alpha)} \ln\left(2K\Delta_w^{(\alpha)}\right)} > \sum_{l=1}^{n} \mu_j(t_{w,l}) + B_w^{(\alpha)}$$

and thus, because of the choice of $B_w^{(\alpha)}$, it must be:

$$\sum_{l=1}^{n} \mu_i(t_{w,l}) > \sum_{l=1}^{n} \mu_j(t_{w,l}).$$

Thus, in virtue of Corollary C.4, it cannot be $i \notin \mathcal{I}_w^\times$, $j \in \mathcal{I}_w^\times$. But, to eliminate all alive arms in $\mathcal{I}_w^\times$, we would need at least one cycle in which an elimination of the kind above happens. Hence there will always be at least an arm in $\mathcal{I}_w^\times$ alive. Let $i_{w,n}^\times$ be such arm during the $n$-th round-robin cycle. Let's bound the regret of a generic arm $j \in [\![K]\!]$ during the $w$-th window, under event $\mathcal{G}_{w,(2K\Delta_w^{(\alpha)})^{-1}}$.

$$
\begin{aligned}
\sum_{l=1}^{N_{j,w}} \left[\mu_{i_{t_{j,w,l}}^*}(t_{j,w,l}) - \mu_j(t_{j,w,l})\right] &\leq \sum_{l=1}^{N_{j,w}-1} \left[\mu_{i_{t_{j,w,l}}^*}(t_{j,w,l}) - \mu_j(t_{j,w,l})\right] + 1 \\
&\leq \sum_{l=1}^{N_{j,w}-1} \left[\mu_{i_{w,N_{j,w}}^\times}(t_{j,w,l}) - \mu_j(t_{j,w,l})\right] + N_{j,w}d_w^* + 1 \\
&= \sum_{l=1}^{N_{j,w}-1} \left[\mu_{i_{w,N_{j,w}}^\times}(t_{i_{w,N_{j,w}}^\times,w,l}) - \mu_j(t_{j,w,l})\right] \\
&\quad + \sum_{l=1}^{N_{j,w}-1} \left[\mu_{i_{w,N_{j,w}}^\times}(t_{j,w,l}) - \mu_{i_{w,N_{j,w}}^\times}(t_{i_{w,N_{j,w}}^\times,w,l})\right] \\
&\quad + N_{j,w}d_w^* + 1 \\
&\leq \widetilde{S}_{i_{w,N_{j,w}}^\times,w,d_{w,N_{j,w}}-1} - \widetilde{S}_{j,w,d_{w,N_{j,w}}-1} + 1 + N_{j,w}d_w^* + 1 \\
&\leq 2 + 4\sigma\sqrt{\Delta_w^{(\alpha)}\ln\left(2K\Delta_w^{(\alpha)}\right)} + \hat{S}_{i_{w,N_{j,w}}^\times,w,d_{w,N_{j,w}}-1} \\
&\quad - \hat{S}_{j,w,d_{w,N_{j,w}}-1} + N_{j,w}d_w^* \\
&= B_w^{(\alpha)} + \hat{S}_{i_{w,N_{j,w}}^\times,w,d_{w,N_{j,w}}-1} - \hat{S}_{j,w,d_{w,N_{j,w}}-1} + N_{j,w}d_w^* \\
&\leq 2B_w^{(\alpha)} + N_{j,w}d_w^*
\end{aligned}
$$

where the last line follows from the fact that we have not eliminated arm $j$ at the end of the $(N_{j,w}-1)$-th round robin cycle. Thus, the regret during the $w$-th window, under event $\mathcal{G}_{w,(2K\Delta_w^{(\alpha)})^{-1}}$, is upper bounded as:

$$\sum_{t=s_w^{(\alpha)}}^{e_w^{(\alpha)}} \left[\mu_{i_t^*}(t) - \mu_{I_t}(t)\right] = \sum_{j \in [\![K]\!]} \sum_{l=1}^{N_{j,w}} \left[\mu_{i_{t_{j,w,l}}^*}(t_{j,w,l}) - \mu_j(t_{j,w,l})\right] \leq 2KB_w^{(\alpha)} + \Delta_w^{(\alpha)}d_w^*.$$

Finally, in virtue of Lemma C.1:

$$
\begin{aligned}
R_\nu(\texttt{RC-BE}(\alpha), \{w\}) &\leq 2KB_w^{(\alpha)} + \Delta_w^{(\alpha)}d_w^* + \Delta_w^{(\alpha)} \mathop{\mathbb{P}}_{X \sim \nu}\left[\overline{\mathcal{G}_{w,(2K\Delta_w^{(\alpha)})^{-1}}}\right] \\
&\leq 2KB_w^{(\alpha)} + \Delta_w^{(\alpha)}d_w^* + \frac{1}{2K} \leq 3KB_w^{(\alpha)} + \Delta_w^{(\alpha)}d_w^*.
\end{aligned}
$$

$\square$

## C.4 Proof of Lemma 4.2

The goal of this section is to prove Lemma 4.2. To this end, we need several intermediate results. We start with a lower bound to $e_w^{(\alpha)}$.

**Lemma C.5.** *For any $\alpha \geqslant 1$, $w \in \mathbb{N}_{\geqslant 1}$ it holds that*

$$e_w^{(\alpha)} \geqslant \frac{w^{1+\alpha}}{2(1+\alpha)}.$$

*Proof.* If $w = 1$, we trivially have

$$e_1^{(\alpha)} = 1 > \frac{1}{2(1+\alpha)}.$$

Now, suppose $w \geqslant 2$, then

$$e_w^{(\alpha)} = \sum_{l=1}^{w} \Delta_l^{(\alpha)} \geqslant \sum_{l=1}^{w} l^\alpha \geqslant \int_1^w x^\alpha dx = \left( \frac{w^{1+\alpha}}{1+\alpha} - \frac{1}{1+\alpha} \right) \geqslant \frac{w^{1+\alpha}}{2(1+\alpha)}.$$

$\square$

Now we introduce the results through which we exploit the concavity of the instance.

**Lemma C.6.** *For any restless rising concave MAB $\nu$, $t_1, t_2 \in \mathbb{N}_{\geqslant 1}$, $t_2 \geqslant t_1 \geqslant 2$, we have:*

$$\Upsilon_{\boldsymbol{\nu}}(t_1, t_2) \leqslant \frac{t_2 - t_1}{t_2 - t_1 + 1} \Upsilon_{\boldsymbol{\nu}}(t_1 - 1, t_2).$$

*Proof.*

$$\begin{aligned}
\Upsilon_{\boldsymbol{\nu}}(t_1, t_2) &= \sum_{l=t_1}^{t_2-1} \max_{i \in [\![K]\!]} \gamma_i(l) \\
&\leqslant \sum_{l=t_1}^{t_2-1} \max_{i \in [\![K]\!]} \gamma_i(l) + \frac{t_2 - t_1}{t_2 - t_1 + 1} \left( \max_{i \in [\![K]\!]} \gamma_i(t_1 - 1) - \frac{1}{t_2 - t_1} \sum_{l=t_1}^{t_2-1} \max_{i \in [\![K]\!]} \gamma_i(l) \right) \\
&= \frac{t_2 - t_1}{t_2 - t_1 + 1} \sum_{l=t_1-1}^{t_2-1} \max_{i \in [\![K]\!]} \gamma_i(l) = \frac{t_2 - t_1}{t_2 - t_1 + 1} \Upsilon_{\boldsymbol{\nu}}(t_1 - 1, t_2).
\end{aligned}$$

$\square$

Before proving Lemma 4.2, we need an intermediate upper bound to $d_w(i)$.

**Lemma C.7.** *For all restless rising concave MABs $\nu$, $\alpha \geqslant 1$, $w \in \mathbb{N}_{\geqslant 1}$, $i \in [\![K]\!]$, we have that:*

$$d_w(i) \leqslant (|[i]_{w \times +}| - 1) \max_{\substack{j,k \in [i]_{w \times +} \; s.t. j \; _w \times k \\ t \in [\![s_w^{(\alpha)}, e_w^{(\alpha)}]\!]}} |\mu_j(t) - \mu_k(t)|.$$

*Proof.* If $j_{\;w} \times^+ k$, there must exist distinct $i_1, \ldots, i_n$ different from $j$ and $k$ ($n \in [\![0, |[i]_{w \times +}| - 2]\!]$) such that $j_{\;w} \times i_1, i_{1\;w} \times i_2, \ldots, i_{n-1\;w} \times i_n, i_{n\;w} \times k$. Then, for $t \in [\![s_w^{(\alpha)}, e_w^{(\alpha)}]\!]$, we have:

$$\begin{aligned}
|\mu_j(t) - \mu_k(t)| &\leqslant |\mu_j(t) - \mu_{i_1}(t)| + |\mu_{i_1}(t) - \mu_{i_2}(t)| + \ldots |\mu_{i_n}(t) - \mu_k(t)| \\
&\leqslant (n+1) \max_{\substack{j',k' \in [i]_{w \times +} \; s.t. j'_{\;w} \times k' \\ t' \in [\![s_w^{(\alpha)}, e_w^{(\alpha)}]\!]}} |\mu_{j'}(t') - \mu_{k'}(t')| \\
&\leqslant (|[i]_{w \times +}| - 1) \max_{\substack{j',k' \in [i]_{w \times +} \; s.t. j'_{\;w} \times k' \\ t' \in [\![s_w^{(\alpha)}, e_w^{(\alpha)}]\!]}} |\mu_{j'}(t') - \mu_{k'}(t')|.
\end{aligned}$$

$\square$

We are ready to prove Lemma 4.2.

**Lemma 4.2.** *For all restless rising concave MABs $\nu$, $\alpha \geqslant 1$, $w \in \mathbb{N}_{\geqslant 1}$, $i \in [\![K]\!]$, we have that:*

$$d_w(i) \leqslant 8(1+\alpha) \left( |[i]_{w \times +}| - 1 \right) w^{-1} \Upsilon_{\boldsymbol{\nu}}(1, e_w^{(\alpha)}) \leqslant 16\alpha K w^{-1} \Upsilon_{\boldsymbol{\nu}}(1, e_w^{(\alpha)}).$$

*Proof.* Let $j \uparrow_{t'} k$ for some $j, k \in [i]_{w \times +}$, $t' \in [\![s_w^{(\alpha)} + 1, e_w^{(\alpha)}]\!]$. Let $t \geqslant t'$, $t \in [\![s_w^{(\alpha)}, e_w^{(\alpha)}]\!]$, then

$$\mu_j(t) - \mu_k(t) \leqslant \mu_j(t) - \mu_k(t'-1) \leqslant \mu_j(t) - \mu_j(t'-1) \leqslant \Upsilon_{\boldsymbol{\nu}}\left(s_w^{(\alpha)}, e_w^{(\alpha)}\right),$$

$$\mu_k(t) - \mu_j(t) \leqslant \mu_k(t) - \mu_j(t') \leqslant \mu_k(t) - \mu_k(t') \leqslant \Upsilon_{\boldsymbol{\nu}}\left(s_w^{(\alpha)}, e_w^{(\alpha)}\right).$$

Analogously, if $t < t'$, we have

$$\mu_j(t) - \mu_k(t) \leqslant \mu_j(t'-1) - \mu_k(t) \leqslant \mu_k(t'-1) - \mu_k(t) \leqslant \Upsilon_{\boldsymbol{\nu}}\left(s_w^{(\alpha)}, e_w^{(\alpha)}\right),$$

$$\mu_k(t) - \mu_j(t) \leqslant \mu_k(t') - \mu_j(t) \leqslant \mu_j(t') - \mu_j(t) \leqslant \Upsilon_{\boldsymbol{\nu}}\left(s_w^{(\alpha)}, e_w^{(\alpha)}\right).$$

We conclude that, if $j_{\ w} \times k$, then $|\mu_j(t) - \mu_k(t)| \leqslant \Upsilon_{\boldsymbol{\nu}}\left(s_w^{(\alpha)}, e_w^{(\alpha)}\right)$ for all $t \in [\![s_w^{(\alpha)}, e_w^{(\alpha)}]\!]$. Thus, in virtue of Lemma C.7, if $j_{\ w} \times^+ k$, then:

$$|\mu_j(t) - \mu_k(t)| \leqslant \left(|[i]_{w \times +}| - 1\right) \Upsilon_{\boldsymbol{\nu}}\left(s_w^{(\alpha)}, e_w^{(\alpha)}\right).$$

For $w \geqslant 2$, by applying iteratively Lemma C.6, we have

$$\Upsilon_{\boldsymbol{\nu}}\left(s_w^{(\alpha)}, e_w^{(\alpha)}\right) \leqslant \frac{e_w^{(\alpha)} - s_w^{(\alpha)}}{e_w^{(\alpha)} - 1} \Upsilon_{\boldsymbol{\nu}}\left(1, e_w^{(\alpha)}\right) \leqslant 2 \frac{\Delta_w^{(\alpha)}}{e_w^{(\alpha)}} \Upsilon_{\boldsymbol{\nu}}\left(1, e_w^{(\alpha)}\right)$$

$$\leqslant 8(1+\alpha) \frac{w^\alpha}{w^{1+\alpha}} \Upsilon_{\boldsymbol{\nu}}\left(1, e_w^{(\alpha)}\right) = 8(1+\alpha) w^{-1} \Upsilon_{\boldsymbol{\nu}}\left(1, e_w^{(\alpha)}\right)$$

where in the last line we used Lemma C.5, the fact that $\lceil x \rceil \leqslant 2x$ for $x \geqslant 1$, and the definition of $\Delta_w^{(\alpha)}$. The same upper bound holds trivially for $w = 1$ since $s_1^{(\alpha)} = e_1^{(\alpha)} = 1$. $\qquad \square$

## C.5 Proof of Lemma 4.3

The goal of this section is to prove Lemma 4.3. To get the result, we start by providing an upper bound to the number of times an arm $i$ overtakes arm $j$ and the expected rewards diverge by a quantity greater than $G > 0$. To this end, we need to prove two auxiliary results.

**Lemma C.8.** *Let* $t^\uparrow, \hat{t}, t^\downarrow \in \mathbb{N}_{\geqslant 1}$, $t^\downarrow > \hat{t} \geqslant t^\uparrow$, $G \in (0, 1]$, $i, j \in [\![K]\!]$ *such that*

$$i \uparrow_{t^\uparrow} j, \mu_i(\hat{t}) \geqslant \mu_j(\hat{t}) + G, j \uparrow_{t^\downarrow} i.$$

*Then:*

$$\gamma_i(t^\uparrow - 1) > \gamma_j(\hat{t}) \geqslant \gamma_i(t^\downarrow), \tag{11}$$

$$\hat{t} - (t^\uparrow - 1) \geqslant G \frac{1}{\gamma_i(t^\uparrow - 1) - \gamma_j(\hat{t})}, \tag{12}$$

$$\mu_i(\hat{t}) - \mu_i(t^\uparrow - 1) \geqslant G \frac{\gamma_j(\hat{t})}{\gamma_i(t^\uparrow - 1) - \gamma_j(\hat{t})}. \tag{13}$$

*Proof.* We start by proving Equation (11). Suppose $\gamma_j(\hat{t}) \geqslant \gamma_i(t^\uparrow - 1)$. Then:

$$\mu_j(\hat{t}) \geqslant \mu_j(t^\uparrow - 1) + (\hat{t} - (t^\uparrow - 1)) \gamma_j(\hat{t})$$
$$\geqslant \mu_i(t^\uparrow - 1) + (\hat{t} - (t^\uparrow - 1)) \gamma_i(t^\uparrow - 1)$$
$$\geqslant \mu_i(\hat{t})$$

which is a contradiction with the definition of $\hat{t}$. Thus it must be $\gamma_j(\hat{t}) < \gamma_i(t^\uparrow - 1)$. Analogously, suppose $\gamma_j(\hat{t}) < \gamma_i(t^\downarrow)$. Then:

$$\mu_j(t^\downarrow) \leqslant \mu_j(\hat{t}) + (t^\downarrow - \hat{t}) \gamma_j(\hat{t})$$
$$< \mu_i(\hat{t}) - G + (t^\downarrow - \hat{t}) \gamma_i(t^\downarrow)$$
$$\leqslant \mu_i(t^\downarrow) - G$$

which is a contradiction with the definition of $t^\downarrow$. Thus it must be $\gamma_j(\hat{t}) \geqslant \gamma_i(t^\downarrow)$. We now prove Equation (12):

$$
\begin{aligned}
G \leqslant \mu_i(\hat{t}) - \mu_j(\hat{t}) &\leqslant \mu_i(t^\uparrow - 1) + (\hat{t} - (t^\uparrow - 1))\gamma_i(t^\uparrow - 1) \\
&\quad - \mu_j(t^\uparrow - 1) - (\hat{t} - (t^\uparrow - 1))\gamma_j(\hat{t}) \\
&\leqslant (\hat{t} - (t^\uparrow - 1))(\gamma_i(t^\uparrow - 1) - \gamma_j(\hat{t}))
\end{aligned}
$$

and thus

$$
\hat{t} - (t^\uparrow - 1) \geqslant G \frac{1}{\gamma_i(t^\uparrow - 1) - \gamma_j(\hat{t})}.
$$

Finally, we prove Equation (13):

$$
\mu_i(\hat{t}) - \mu_i(t^\uparrow - 1) \geqslant \mu_j(\hat{t}) - \mu_j(t^\uparrow - 1) \geqslant (\hat{t} - (t^\uparrow - 1))\gamma_j(\hat{t})
$$

$$
\geqslant G \frac{\gamma_j(\hat{t})}{\gamma_i(t^\uparrow - 1) - \gamma_j(\hat{t})}.
$$

$\square$

**Lemma C.9.** *Let $M \in \mathbb{N}_{\geqslant 1}$, $M \geqslant 2$, $m_1 > m_2 > \cdots > m_M > m_{M+1} > 0$, then:*

$$
\sum_{i=1}^{M} \frac{1}{m_i - m_{i+1}} \geqslant \frac{M^2}{m_1 - m_{M+1}}, \tag{14}
$$

$$
\sum_{i=1}^{M} \frac{m_{i+1}}{m_i - m_{i+1}} \geqslant \frac{M}{\left(\frac{m_1}{m_{M+1}}\right)^{\frac{1}{M}} - 1}. \tag{15}
$$

*Proof.* We regard $m_1 > m_{M+1} > 0$ as fixed constants and study the functions

$$
f(m_2, \ldots, m_M) = \sum_{i=1}^{M} \frac{1}{m_i - m_{i+1}},
$$

$$
g(m_2, \ldots, m_M) = \sum_{i=1}^{M} \frac{m_{i+1}}{m_i - m_{i+1}}
$$

defined for $m_1 > m_2 > \cdots > m_M > m_{M+1}$. Observe that the functions are defined on an open set and their values tend to infinity when the input tends to the border of the domain. We show that they have only one stationary point, which then must be a minimum point. We start by proving Equation (14). Let $k \in [\![2, M]\!]$:

$$
\frac{df}{dm_k}(m_2, \ldots, m_M) = \frac{1}{(m_{k-1} - m_k)^2} - \frac{1}{(m_k - m_{k+1})^2} = 0
$$

if and only if

$$
m_{k+1} = 2m_k - m_{k-1}.
$$

The linear system above is equivalent to:

$$
m_i = (i-1)m_2 - (i-2)m_1 \quad \text{for} \quad i \in [\![3, M+1]\!]. \tag{16}
$$

Thus $m_{M+1} = Mm_2 - (M-1)m_1$, and then

$$
m_2 = \frac{(M-1)m_1 + m_{M+1}}{M}.
$$

By plugging this result into Equation (16), we get the coordinates of the minimum point:

$$
m_i^* := \frac{(M+1-i)m_1 + (i-1)m_{M+1}}{M} \quad \text{for} \quad i \in [\![2, M]\!].
$$

Thus:

$$
f(m_1, \ldots, m_M) \geqslant f(m_1^*, \ldots, m_M^*) = \frac{M^2}{m_1 - m_{M+1}}.
$$

We now prove Equation (15) analogously:

$$\frac{dg}{dm_k}(m_2, \ldots, m_M) = \frac{m_{k-1}}{(m_{k-1} - m_k)^2} - \frac{m_{k+1}}{(m_k - m_{k+1})^2} = 0$$

if and only if

$$m_{k+1} = \frac{m_k^2}{m_{k-1}}$$

if and only if

$$\ln m_{k+1} = 2\ln m_k - \ln m_{k-1}.$$

Observe that we get the same linear system of the previous case, with the difference that the variables are now $\ln m_i$. Thus, the solution is:

$$\ln m_i = \frac{(M + 1 - i)\ln m_1 + (i - 1)\ln m_{M+1}}{M}$$

and then

$$m_i^* := m_1^{\frac{M+1-i}{M}} m_{M+1}^{\frac{i-1}{M}} \quad \text{for} \quad i \in [\![2, M]\!].$$

Finally:

$$g(m_2, \ldots, m_M) \geqslant g(m_2^*, \ldots, m_M^*) = \frac{M}{\left(\frac{m_1}{m_{M+1}}\right)^{\frac{1}{M}} - 1}.$$

$\square$

**Lemma C.10.** *Let $G \in (0, 1]$, $T' \in \mathbb{N}_{\geqslant 1}$, $M \in \mathbb{N}_{\geqslant 1}$, $i, j \in [\![K]\!]$ such that there exist rounds*

$$2 \leqslant t_1^\uparrow \leqslant \hat{t}_1 < t_1^\downarrow \leqslant t_2^\uparrow \leqslant \hat{t}_2 < t_2^\downarrow \leqslant \cdots \leqslant t_M^\uparrow \leqslant \hat{t}_M \leqslant T'$$

*which satisfy*

$$i \uparrow_{t_l^\uparrow} j, \mu_i(\hat{t}_l) \geqslant \mu_j(\hat{t}_l) + G \text{ for all } l \in [\![M]\!],$$
$$j \uparrow_{t_l^\downarrow} i \text{ for all } l \in [\![M-1]\!].$$

*Then:*

$$M \leqslant 4\ln(3T'/G)G^{-\frac{1}{2}}.$$

*Proof.* Observe that, since

$$\mu_i(\hat{t}_M) \geqslant \mu_j(\hat{t}_M) + G \geqslant \mu_j(t_M^\uparrow - 1) + G$$
$$\geqslant \mu_i(t_M^\uparrow - 1) + G,$$

we have

$$T'\gamma_i(t_M^\uparrow - 1) \geqslant (\hat{t}_M - (t_M^\uparrow - 1))\gamma_i(t_M^\uparrow - 1) \geqslant \mu_i(\hat{t}_M) - \mu_i(t_M^\uparrow - 1) \geqslant G$$

and thus

$$\gamma_i(t_M^\uparrow - 1) \geqslant \frac{G}{T'}.$$

Now, assume $M \geqslant 3$. Then:

$$1 \geqslant \mu_i(T') - \mu_i(1) \geqslant \sum_{l=1}^{M-1} (\mu_i(\hat{t}_l) - \mu_i(t_l^\uparrow - 1))$$

$$\geqslant G \sum_{l=1}^{M-1} \frac{\gamma_j(\hat{t}_l)}{\gamma_i(t_l^\uparrow - 1) - \gamma_j(\hat{t}_l)} \tag{17}$$

$$\geqslant G \sum_{l=1}^{M-1} \frac{\gamma_i(t_{l+1}^\uparrow - 1)}{\gamma_i(t_l^\uparrow - 1) - \gamma_i(t_{l+1}^\uparrow - 1)} \tag{18}$$

$$\geqslant G \frac{M - 1}{\left(\frac{\gamma_i(t_1^\uparrow - 1)}{\gamma_i(t_M^\uparrow - 1)}\right)^{\frac{1}{M-1}} - 1} \tag{19}$$

$$\geqslant G \frac{M-1}{\left(\frac{T'}{G}\right)^{\frac{1}{M-1}}-1} = G \frac{M-1}{\exp\left(\frac{\ln(T'/G)}{M-1}\right)-1} \tag{20}$$

where line (17) follows from Lemma C.8, line (18) follows from the fact that $\frac{x}{a-x}$ is non-decreasing for $a \geqslant 0$ and the concavity, line (19) follows from Lemma C.9, and line (20) follows from the fact that $\gamma_i(t_1^\uparrow - 1) \leqslant 1$ and $\gamma_i(t_M^\uparrow - 1) \geqslant \frac{G}{T'}$. Now, if $M \geqslant 1 + \ln(T'/G)$, by Lemma D.2, we have $\exp\left(\frac{\ln(T'/G)}{M-1}\right) - 1 \leqslant 3\frac{\ln(T'/G)}{M-1}$, and thus, by the chain of inequalities above:

$$1 \geqslant G \frac{(M-1)^2}{3\ln(T'/G)} \text{ iff } M \leqslant 1 + \sqrt{3\ln(T'/G)G^{-1}}.$$

Thus, by considering all possible cases, we have:

$$M \leqslant \max\{2, \ln(T'/G), 1 + \sqrt{3\ln(T'/G)G^{-1}}\} \leqslant 4\ln(3T'/G)G^{-\frac{1}{2}}.$$

$\square$

We are now ready to prove Lemma 4.3.

**Lemma 4.3.** *For all restless rising concave MABs $\nu$, $\alpha \geqslant 1$, $T \in \mathbb{N}_{\geqslant 1}$, $d \in (0, K]$, we have that:*

$$|\mathcal{W}_{>d}(T)| \leqslant 9\ln\left(3e^{(\alpha)}_{w^{(\alpha)}(T)}K/d\right)K^{\frac{5}{2}}d^{-\frac{1}{2}}.$$

*Proof.* Let $w \in \mathcal{W}_{>d}(T)$. Then there exists $i \in [\![K]\!]$ such that $d_w(i) > d$. But, in virtue of Lemma C.7, we have:

$$(|[i]_{w \times +}| - 1) \max_{\substack{j,k \in [i]_{w \times +} \text{ s.t. } j_{\,w} \times k \\ t \in [\![s_w^{(\alpha)}, e_w^{(\alpha)}]\!]}} |\mu_j(t) - \mu_k(t)| \geqslant d_w(i) > d.$$

Thus, there must be $j, k \in [i]_{w \times +}$ and $t \in [\![s_w^{(\alpha)}, e_w^{(\alpha)}]\!]$ such that $j_{\,w} \times k$ and

$$|\mu_j(t) - \mu_k(t)| > \frac{d}{|[i]_{w \times +}| - 1} > \frac{d}{K}.$$

Observe that it must be either $i \times_{t'} j$ for $t' \leqslant t$ or $i \times_{t'} j$ for $t' > t$, with $t' \in [\![s_w^{(\alpha)} + 1, e_w^{(\alpha)}]\!]$. W.l.o.g. we assume that $i$ overtakes $j$. In the first case, window $w$ must contain one of the rounds in which $i$ overtakes $j$ and then their expected rewards diverge by at least $d/K$. In the second case, window $w$ must contain either the first round in which $i$ overtakes $j$ and which is right after one of the rounds in which $i$ overtakes $j$ and their expected rewards diverge by at least $d/K$ or the first time in which $i$ overtakes $j$. In virtue of Lemma C.10 with $G = \frac{d}{K}$ and $T' = e^{(\alpha)}_{w^{(\alpha)}(T)}$, the rounds described in the first case are in number no more than $4\ln\left(3e^{(\alpha)}_{w^{(\alpha)}(T)}K/d\right)(d/K)^{-1/2}$, while the rounds described in the second case are in number no more than $4\ln\left(3e^{(\alpha)}_{w^{(\alpha)}(T)}K/d\right)(d/K)^{-1/2} + 1$ for a fixed choice of $i, j \in [\![K]\!]$. Since we have at most $K^2$ such choices, it must be:

$$|\mathcal{W}_{>d}(T)| \leqslant K^2\left(8\ln\left(3e^{(\alpha)}_{w^{(\alpha)}(T)}K/d\right)(d/K)^{-\frac{1}{2}} + 1\right)$$
$$\leqslant 9\ln\left(3e^{(\alpha)}_{w^{(\alpha)}(T)}K/d\right)K^{\frac{5}{2}}d^{-\frac{1}{2}}.$$

$\square$

## C.6  Proof of Theorem 4.4

The goal of this section is to prove Theorem 4.4. We start with an upper bound to $w^{(\alpha)}(T)$, $e^{(\alpha)}_{w^{(\alpha)}(T)}$, and $\Upsilon_\nu\left(1, e^{(\alpha)}_{w^{(\alpha)}(T)}\right)$.

**Lemma C.11.** *For all restless rising concave MABs $\boldsymbol{\nu}$, $\alpha \geqslant 1$, $T \in \mathbb{N}_{\geqslant 2}$, we have:*

$$w^{(\alpha)}(T) \leqslant (2(1+\alpha)T)^{1/(1+\alpha)} \leqslant 4\alpha T^{1/(1+\alpha)}, \tag{21}$$

$$e^{(\alpha)}_{w^{(\alpha)}(T)} \leqslant 4(1+\alpha)T \leqslant 8\alpha T, \tag{22}$$

$$\Upsilon_{\boldsymbol{\nu}}\left(1, e^{(\alpha)}_{w^{(\alpha)}(T)}\right) \leqslant 8(1+\alpha)\Upsilon_{\boldsymbol{\nu}}(1,T) \leqslant 16\alpha\Upsilon_{\boldsymbol{\nu}}(1,T). \tag{23}$$

*Proof.* We start by proving Equation (21). If $w \in \mathbb{N}_{\geqslant 1}$, $w \geqslant (2(1+\alpha)T)^{1/(1+\alpha)}$, then, by Lemma C.5, we have:

$$e^{(\alpha)}_w \geqslant \frac{w^{1+\alpha}}{2(1+\alpha)} \geqslant T.$$

Thus it must be $w^{(\alpha)}(T) \leqslant (2(1+\alpha)T)^{1/(1+\alpha)}$. We now use Equation (21) to prove Equation (22).

$$e^{(\alpha)}_{w^{(\alpha)}(T)} \leqslant w^{(\alpha)}(T)\Delta^{(\alpha)}_{w^{(\alpha)}(T)}$$

$$\leqslant 2(2(1+\alpha)T)^{\frac{1}{1+\alpha}}(2(1+\alpha)T)^{\frac{\alpha}{1+\alpha}} = 4(1+\alpha)T, \tag{24}$$

where in line (24) we use the definition of $\Delta^{(\alpha)}_w$, Equation (21), and the fact that $\lceil x \rceil \leqslant 2x$ for $x \geqslant 1$. Finally, we prove Equation (23).

$$\Upsilon_{\boldsymbol{\nu}}\left(1, e^{(\alpha)}_{w^{(\alpha)}(T)}\right) \leqslant \frac{e^{(\alpha)}_{w^{(\alpha)}(T)} - 1}{T-1}\Upsilon_{\boldsymbol{\nu}}(1,T) \tag{25}$$

$$\leqslant 2\frac{e^{(\alpha)}_{w^{(\alpha)}(T)}}{T}\Upsilon_{\boldsymbol{\nu}}(1,T) \leqslant 8(1+\alpha)\Upsilon_{\boldsymbol{\nu}}(1,T), \tag{26}$$

where line (25) follows by applying iteratively Lemma C.6 and line (26) follows from the fact that $T \geqslant 2$ and by Equation (22). $\qquad\square$

We are ready to prove Theorem 4.4.

**Theorem 4.4** (Upper Bound for the Rising Concave Setting). *For all restless rising concave MABs $\boldsymbol{\nu}$, $\alpha \geqslant 1$, $T \in \mathbb{N}_{\geqslant 24}$, we have that:*

$$R_{\boldsymbol{\nu}}(\texttt{RC-BE}(\alpha), T) \leqslant 2^{15}\alpha^3 \left(\ln\left(\alpha K T^3\right)\right)^{\frac{3}{2}} \left((1+\sigma)K^3 T^{\frac{3/4\alpha}{1+\alpha}} + K^3 T^{\frac{5/4\alpha-1}{1+\alpha}}\Upsilon_{\boldsymbol{\nu}}(1,T)\right.$$

$$\left. + (1+\sigma)KT^{\frac{1+\alpha/2}{1+\alpha}}\right).$$

*In particular, for $\alpha' = 8/3$, we get:*

$$R_{\boldsymbol{\nu}}(\texttt{RC-BE}(\alpha'), T) = \tilde{\mathcal{O}}\left(\sigma K^3 T^{\frac{6}{11}} + K^3 T^{\frac{7}{11}}\Upsilon_{\boldsymbol{\nu}}(1,T) + \sigma KT^{\frac{7}{11}}\right).$$

*Furthermore, for*

$$\alpha'' = \frac{8 - 4\log_T\left(\frac{K^2 V_T}{1+\sigma}\right)}{3 + 4\log_T\left(\frac{K^2 V_T}{1+\sigma}\right)},$$

*under the additional assumptions $\boldsymbol{\nu} \in \mathcal{E}_c^\sigma(T, V_T)$,*

$$T \geqslant \max \begin{cases} (1+\sigma)^{4/3}K^{-8/3}V_T^{-4/3} + 1 \\ (1+\sigma)^{-8/5}K^{16/5}V_T^{8/5} \end{cases},$$

*we get:*

$$R_{\boldsymbol{\nu}}(\texttt{RC-BE}(\alpha''), T) = \tilde{\mathcal{O}}\left(\sigma^{\frac{14}{11}}K^{\frac{27}{11}}T^{\frac{6}{11}}V_T^{-\frac{3}{11}} + \sigma^{\frac{9}{11}}K^{\frac{15}{11}}T^{\frac{7}{11}}V_T^{\frac{2}{11}}\right).$$

*Proof.* Let $d' := KT^{-(\alpha/2)/(1+\alpha)} \in (0, K]$. Then:

$$R_{\boldsymbol{\nu}}(\text{RC-BE}(\alpha), \mathcal{W}_{>d'}(T)) \leqslant |\mathcal{W}_{>d'}(T)| \max_{w \in \mathcal{W}_{>d'}(T)} \{3KB_w^{(\alpha)} + \Delta_w^{(\alpha)} d_w^*\} \tag{27}$$

$$\leqslant 9\ln\left(3e_{w^{(\alpha)}(T)}^{(\alpha)} T^{\frac{\alpha/2}{1+\alpha}}\right) K^2 T^{\frac{\alpha/4}{1+\alpha}} \tag{28}$$

$$\cdot \max_{w \in \mathcal{W}_{>d'}(T)} \{3KB_w^{(\alpha)} + 16\alpha K \Delta_w^{(\alpha)} w^{-1} \Upsilon_{\boldsymbol{\nu}}(1, e_w^{(\alpha)})\}$$

$$\leqslant 9\ln(24\alpha T^2) K^2 T^{\frac{\alpha/4}{1+\alpha}} \tag{29}$$

$$\cdot \max_{w \in \mathcal{W}_{>d'}(T)} \left\{ 6K\left(1 + 2\sigma\sqrt{\Delta_w^{(\alpha)} \ln(2K\Delta_w^{(\alpha)})}\right) \right.$$

$$\left. + 32\alpha K w^{\alpha-1} \Upsilon_{\boldsymbol{\nu}}(1, e_w^{(\alpha)}) \right\}$$

$$\leqslant 9\ln(24\alpha T^2) K^2 T^{\frac{\alpha/4}{1+\alpha}} \tag{30}$$

$$\cdot \left( 6K\left(1 + 2\sigma\sqrt{8\alpha T^{\frac{\alpha}{1+\alpha}} \ln(16\alpha KT)}\right) \right.$$

$$\left. + 2^{11}\alpha^3 KT^{\frac{\alpha-1}{1+\alpha}} \Upsilon_{\boldsymbol{\nu}}(1, T) \right)$$

$$\leqslant 2^4 \ln(\alpha KT^3) K^2 T^{\frac{\alpha/4}{1+\alpha}} \tag{31}$$

$$\cdot 2^{11}\alpha^3 (\ln(\alpha KT^3))^{\frac{1}{2}} K\left((1+\sigma)T^{\frac{\alpha/2}{1+\alpha}} + T^{\frac{\alpha-1}{1+\alpha}} \Upsilon_{\boldsymbol{\nu}}(1, T)\right)$$

$$= 2^{15}\alpha^3(\ln(\alpha KT^3))^{\frac{3}{2}} K^3\left((1+\sigma)T^{\frac{3/4\alpha}{1+\alpha}} + T^{\frac{5/4\alpha-1}{1+\alpha}} \Upsilon_{\boldsymbol{\nu}}(1, T)\right)$$

where line (27) follows from Lemma 4.1, line (28) follows from Lemma 4.3 and Lemma 4.2, line (29) follows from Lemma C.11, the definition of $\Delta_w^{(\alpha)}$, the fact that $\lceil x \rceil \leqslant 2x$ for $x \geqslant 1$, and the definition of $B_w^{(\alpha)}$, line (30) follows from the fact that the expression inside $\max$ is increasing in $w$, Lemma C.11, and the fact that $\lceil x \rceil \leqslant 2x$ for $x \geqslant 1$, and line (31) follows from $T \geqslant 24$. Furthermore:

$$R_{\boldsymbol{\nu}}(\text{RC-BE}(\alpha), \mathcal{W}_{\leqslant d'}(T)) \leqslant |\mathcal{W}_{\leqslant d'}(T)| \max_{w \in \mathcal{W}_{\leqslant d'}(T)} \{3KB_w^{(\alpha)} + \Delta_w^{(\alpha)} d_w^*\} \tag{32}$$

$$\leqslant w^{(\alpha)}(T)\left( 6K\left(1 + 2\sigma\sqrt{\Delta_{w^{(\alpha)}(T)}^{(\alpha)} \ln(2K\Delta_{w^{(\alpha)}(T)}^{(\alpha)})}\right) + \Delta_{w^{(\alpha)}(T)}^{(\alpha)} d'\right) \tag{33}$$

$$\leqslant 4\alpha T^{\frac{1}{1+\alpha}}(1+\sigma)(\ln(\alpha KT^3))^{\frac{1}{2}}(12K\sqrt{8\alpha T^{\frac{\alpha}{1+\alpha}}} \tag{34}$$

$$+ 8\alpha KT^{\frac{\alpha}{1+\alpha}} T^{-\frac{\alpha/2}{1+\alpha}})$$

$$\leqslant 2^9 \alpha^2(1+\sigma)(\ln(\alpha KT^3))^{\frac{3}{2}} KT^{\frac{1+\alpha/2}{1+\alpha}}$$

$$\leqslant 2^{15}\alpha^3(1+\sigma)(\ln(\alpha KT^3))^{\frac{3}{2}} KT^{\frac{1+\alpha/2}{1+\alpha}}$$

where line (32) follows from Lemma 4.1, line (33) follows from the definitions of $B_w^{(\alpha)}$ and $\mathcal{W}_{\leqslant d'}(T)$, and line (34) follows from Lemma C.11, $T \geqslant 24$, $\lceil x \rceil \leqslant 2x$ for $x \geqslant 1$, and the definition of $d'$. By summing the previous results:

$$R_{\boldsymbol{\nu}}(\text{RC-BE}(\alpha), T) \leqslant R_{\boldsymbol{\nu}}(\text{RC-BE}(\alpha), \mathcal{W}_{>d'}(T)) + R_{\boldsymbol{\nu}}(\text{RC-BE}(\alpha), \mathcal{W}_{\leqslant d'}(T))$$

$$\leqslant 2^{15}\alpha^3(\ln(\alpha KT^3))^{\frac{3}{2}}\left((1+\sigma)K^3 T^{\frac{3/4\alpha}{1+\alpha}} + K^3 T^{\frac{5/4\alpha-1}{1+\alpha}} \Upsilon_{\boldsymbol{\nu}}(1, T)\right.$$

$$\left. + (1+\sigma)KT^{\frac{1+\alpha/2}{1+\alpha}}\right).$$

Finally, observe that, under the additional assumption $\boldsymbol{\nu} \in \mathcal{E}_c^\sigma(T, V_T)$, we have $\Upsilon_{\boldsymbol{\nu}}(1, T) \leqslant V_T$, and the additional constraint on $T$ guarantees $\alpha'' \geqslant 1$. $\qquad\square$

# D  Technical Lemmas

**Lemma D.1.** *Let $\mu_1, \mu_2 \in [0, 1]$ with $\mu_1 \leqslant \mu_2$ and $\sigma \geqslant 1$. Then:*

$$\mathrm{D}_{\mathrm{KL}}(\psi(\mu_1, \sigma) \| \psi(\mu_2, \sigma)) \leqslant \frac{2(\mu_2 - \mu_1)^2}{\ln(2)\sigma^2}$$

*where $\mathrm{D}_{\mathrm{KL}}(\cdot \| \cdot)$ is the Kullback-Leibler divergence defined in Appendix B, and $\psi(\mu, \sigma)$ is the distribution defined in Section 3.1.*

*Proof.* Let $p(\mu, \sigma) := \frac{1}{4} + \frac{\mu}{2\sigma}$. Consider the function:

$$f(x) := \mathrm{D}_{\mathrm{KL}}(\psi(\mu_1, \sigma) \| \psi(\mu_1 + x, \sigma))$$

$$= p(\mu_1, \sigma) \log_2 \left( \frac{p(\mu_1, \sigma)}{p(\mu_1 + x, \sigma)} \right) + (1 - p(\mu_1, \sigma)) \log_2 \left( \frac{1 - p(\mu_1, \sigma)}{1 - p(\mu_1 + x, \sigma)} \right)$$

for $x \in [0, \mu_2 - \mu_1]$. Then:

$$f'(x) = \frac{1}{\ln(2)} \frac{\partial p}{\partial \mu}(\mu + x, \sigma) \left( \frac{1 - p(\mu_1, \sigma)}{1 - p(\mu_1 + x, \sigma)} - \frac{p(\mu_1, \sigma)}{p(\mu_1 + x, \sigma)} \right),$$

$$f''(x) = \frac{1}{\ln(2)} \left( \frac{\partial p}{\partial \mu}(\mu + x, \sigma) \right)^2 \left( \frac{1 - p(\mu_1, \sigma)}{(1 - p(\mu_1 + x, \sigma))^2} + \frac{p(\mu_1, \sigma)}{p^2(\mu_1 + x, \sigma)} \right).$$

By direct evaluation, we have $f(0) = f'(0) = 0$. Furthermore, since $\mu \in [0, 1]$, $\sigma \geqslant 1$, imply $p(\mu, \sigma) \in [1/4, 3/4]$, then:

$$f''(x) \leqslant \frac{1}{\ln(2)} \left( \frac{\partial p}{\partial \mu}(\mu + x, \sigma) \right)^2 \left( \frac{1 - p(\mu_1, \sigma)}{(1/4)^2} + \frac{p(\mu_1, \sigma)}{(1/4)^2} \right)$$

$$= \frac{16}{\ln(2)} \left( \frac{\partial p}{\partial \mu}(\mu + x, \sigma) \right)^2 = \frac{4}{\ln(2)\sigma^2}.$$

Finally:

$$f(x) = f(0) + \int_0^x \left( f'(0) + \int_0^{x_1} f''(x_2) dx_2 \right) dx_1 \leqslant \frac{2x^2}{\ln(2)\sigma^2}.$$

The result follows from the fact that $\mathrm{D}_{\mathrm{KL}}(\psi(\mu_1, \sigma) \| \psi(\mu_2, \sigma)) = f(\mu_2 - \mu_1)$. $\qquad\square$

**Lemma D.2.**

$$e^x - 1 \leqslant 3x \quad for \quad x \in [0, 1].$$

*Proof.* Let $f(x) = e^x - 1$. Then: $f'(x) = e^x = f''(x)$. Thus, by Taylor's theorem, if $x \in [0, 1]$, there exists $\xi \in (0, 1)$ such that

$$f(x) = f(0) + f'(0)x + \frac{f''(\xi)}{2}x^2 = x \left( 1 + \frac{e^\xi}{2}x \right) \leqslant x \left( 1 + \frac{e}{2} \right) \leqslant 3x.$$

$\qquad\square$

# E  Numerical Simulations

In this appendix, we present additional numerical simulations which compare `RC-BE`$(\alpha)$ with the baseline algorithms reported in Section 5. Furthermore, we provide information regarding the compute resources used to run the simulations.

**Baselines.** We consider the following baseline algorithms:

- `Rexp3` (Besbes et al., 2014), an algorithm for restless MABs based on a variation budget for the expected rewards of the arms over the learning horizon.
- `R-less-UCB` (Metelli et al., 2022), an algorithm for restless rising concave MABs which relies on the optimism principle and exploits the structure of the setting through a specifically crafted estimator.

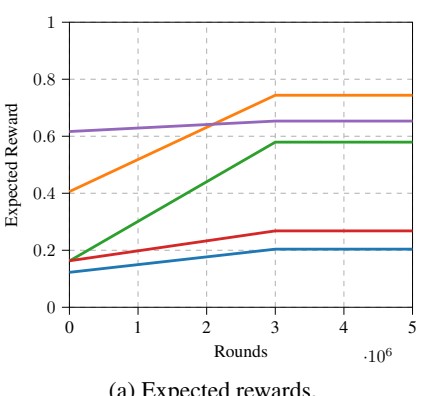

(a) Expected rewards.

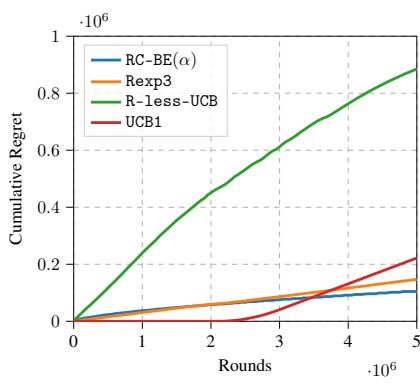

(b) Cumulative regret (10 runs, mean $\pm$ 95% C.I.).

Figure 4: Piecewise linear instance.

- UCB1 (Auer et al., 2002a; Bubeck, 2010), one of the most effective algorithms for stationary MABs.

The choices of the parameters of the algorithms that we compared are the following:

- Rexp3: $V_T = K$ since, as remarked in Section 2, in the rising setting the cumulative increment is always smaller than or equal to $K$; $\Delta_T = \lceil (K \ln(K))^{1/3} (T/V_T)^{2/3} \rceil$; $\gamma = \min \left\{ 1, \sqrt{K \ln(K)/(\Delta_T(e-1))} \right\}$ as recommended in (Besbes et al., 2014).
- R-less-UCB: $h_{i,t} = \lfloor \epsilon N_{i,t-1} \rfloor$ where $N_{i,t-1}$ is the number of times arm $i$ has been pulled by the agent in the first $t-1$ rounds, with $\epsilon \in (0, 1/2)$; $\alpha > 2$ as prescribed in (Metelli et al., 2022). In particular, we choose $\epsilon = 0.25$; $\alpha = 2.1$.
- UCB1: the upper confidence bound interval for arm $i$ at round $t$ is $\sigma \sqrt{4 \ln(t)/N_{i,t-1}}$.

### E.1 Additional Instances

**Piecewise Linear Instance.** The piecewise linear curves that describe the evolution of the expected rewards in the simulation have the following functional form:

$$f(t) = \begin{cases} \frac{T-t}{T-1}\mu_i + \frac{t-1}{T-1}\mu_e & \text{if } t \leq t_{\text{flat}} \\ \frac{T-t_{\text{flat}}}{T-1}\mu_i + \frac{t_{\text{flat}}-1}{T-1}\mu_e & \text{if } t > t_{\text{flat}} \end{cases},$$

for $t \in \llbracket T \rrbracket$ where $\mu_i, \mu_e \in [0, 1]$, $\mu_i \leq \mu_e$. After the *flattening time* $t_{\text{flat}} \in \llbracket T \rrbracket$, the expected rewards of the arms stop increasing. The expected reward curves of the simulated instance are reported in Figure 4a. The algorithms are evaluated on $T = 5 \cdot 10^6$ rounds. The standard deviation of the noise is $\sigma = 0.1$. The empirical cumulative regret suffered by the algorithms is shown in Figure 4b. We can observe that RC-BE($\alpha$) is the algorithm that achieves the lowest regret at the horizon. The behavior of all other algorithms is explained by the same observations stated for the exponential instance in Section 5. Conversely to what happens in the exponential instance, in this case, UCB1 shows a better performance than R-less-UCB. This is due to the fact that the change of the optimal arm happens later in time and the distance between the expected rewards of the first and last optimal arms is less w.r.t. the exponential instance presented in Section 5.

**Constant Instance.** In this simulation, the expected rewards of the arms do not change with time (i.e., stationary MABs). The expected reward curves of the simulated instance are reported in Figure 5a. The algorithms are evaluated on $T = 10^6$ rounds. The standard deviation of the noise is $\sigma = 0.01$. The empirical cumulative regret suffered by the algorithms is shown in Figure 5b. UCB1 is the algorithm that achieves the lowest regret. This is consistent with the fact that the instance is stationary. RC-BE($\alpha$) has the second-best performance. The reduction of the standard deviation of the noise leads to smaller confidence bounds and, thus, a better performance, for R-less-UCB. Conversely, Rexp3 is not able to exploit this fact, being based on the Exp3 algorithm which is designed for the adversarial setting.

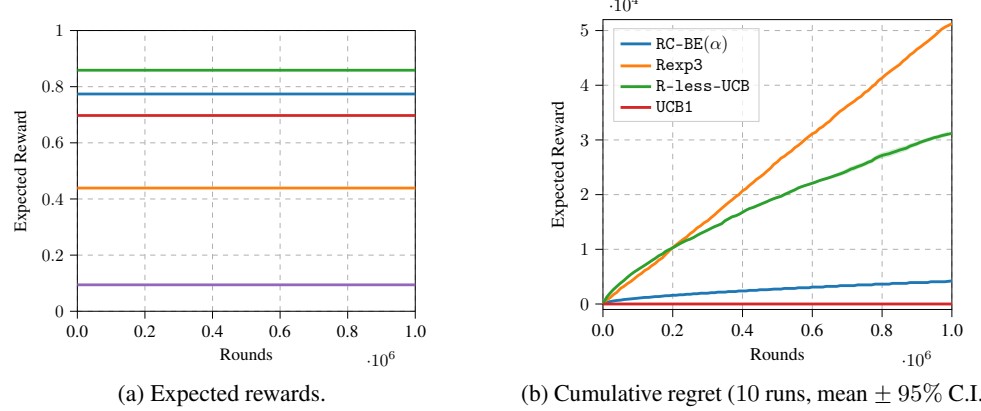

(a) Expected rewards.  (b) Cumulative regret (10 runs, mean $\pm$ 95% C.I.).

Figure 5: Constant instance.

## E.2 Compute Resources

The simulations were run on a single CPU core with a clock frequency of $2.60\,\mathrm{GHz}$. The system has a $8.0\,\mathrm{GiB}$ RAM. For each algorithm, we report the approximate time required to simulate a single run on the exponential instance with $5 \cdot 10^6$ rounds:

- `Rexp3`: $5\,\mathrm{min}\,50\mathrm{s}$;
- `R-less-UCB`: $8\,\mathrm{min}$;
- `UCB1`: $3\,\mathrm{min}\,30\mathrm{s}$;
- `RC-BE`$(\alpha)$: $1\,\mathrm{min}\,50\mathrm{s}$.

## F  Flaw in the Original Analysis of $K$-armed `Budgeted Exploration`

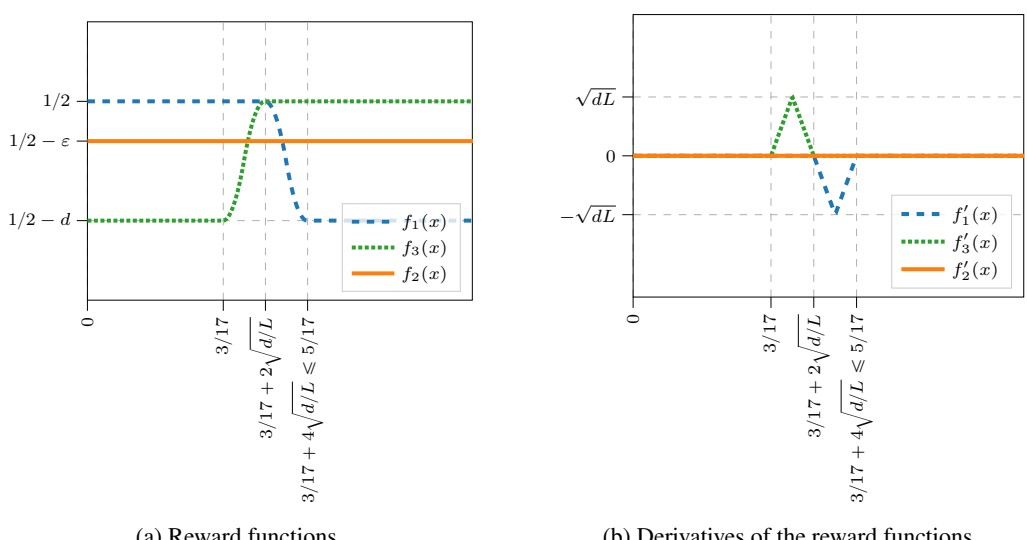

(a) Reward functions.  (b) Derivatives of the reward functions.

Figure 6: Example instance.

In this appendix, we highlight a flaw in the original analysis of the extension of `Budgeted Exploration` in the $K$-armed setting, which is presented in the unpublished preprint (Jia et al., 2024). For notation and definitions, refer to the original paper. The analysis relies on the following proposition, stated in Lemma I.7: "First, we observe that on the clean event $C$, any arm in $A^*$ can never be eliminated for "losing" to an arm in $(A^*)^c$". It is possible to construct a counterexample which satisfies the hypotheses of the lemma and violates the previous proposition. We now show

how. We work with 3 arms. We describe the evolution of the expected reward of the arms only in a certain window. This is sufficient for the construction of the counterexample since the lemma regards the behavior of the algorithm in a single window. The window is composed of $17W$ rounds, with $W \in \mathbb{N}_{\geqslant 2}$ to be chosen later. The expected rewards of the arms are defined as follows:

$$r_a(t) = f_a\left(\frac{t}{17W}\right) \text{ for } t \in [\![17W]\!]$$

where $f_a : [0,1] \to [-1,1]$ is a 2-Hölder function with Lipschitz constant $L > 0$ for $a \in [\![3]\!]$. Such functions and their derivatives are depicted in Figure 6a and Figure 6b, respectively. More specifically, we choose:

- The function in which the expected rewards of the first arm are embedded as:

$$f_1(x) = \frac{1}{2} - \int_0^x \begin{cases} 0 & \text{if } t \in \left[0, \frac{3}{17} + 2\sqrt{\frac{d}{L}}\right] \\ L\left(t - \frac{3}{17} - 2\sqrt{\frac{d}{L}}\right) & \text{if } t \in \left(\frac{3}{17} + 2\sqrt{\frac{d}{L}}, \frac{3}{17} + 3\sqrt{\frac{d}{L}}\right] \\ \sqrt{dL} - L\left(t - \frac{3}{17} - 3\sqrt{\frac{d}{L}}\right) & \text{if } t \in \left(\frac{3}{17} + 3\sqrt{\frac{d}{L}}, \frac{3}{17} + 4\sqrt{\frac{d}{L}}\right] \\ 0 & \text{if } t \in \left(\frac{3}{17} + 4\sqrt{\frac{d}{L}}, 1\right] \end{cases} dt.$$

- The function in which the expected rewards of the second arm are embedded as:

$$f_2(x) = \frac{1}{2} - \varepsilon.$$

- The function in which the expected rewards of the third arm are embedded as:

$$f_3(x) = \frac{1}{2} - d + \int_0^x \begin{cases} 0 & \text{if } t \in \left[0, \frac{3}{17}\right] \\ L\left(t - \frac{3}{17}\right) & \text{if } t \in \left(\frac{3}{17}, \frac{3}{17} + \sqrt{\frac{d}{L}}\right] \\ \sqrt{dL} - L\left(t - \frac{3}{17} - \sqrt{\frac{d}{L}}\right) & \text{if } t \in \left(\frac{3}{17} + \sqrt{\frac{d}{L}}, \frac{3}{17} + 2\sqrt{\frac{d}{L}}\right] \\ 0 & \text{if } t \in \left(\frac{3}{17} + 2\sqrt{\frac{d}{L}}, 1\right] \end{cases} dt.$$

The definitions rely on the constants $d, \varepsilon > 0$, $\varepsilon < d \leqslant 1/2$, which we choose later. To guarantee that the functions are well-defined, we impose:

$$4\sqrt{\frac{d}{L}} \leqslant \frac{2}{17} \quad \text{iff} \quad L \geqslant 34^2 d. \tag{35}$$

We work with deterministic rewards, which can be regarded as a special realization under the clean event $C$. Let $Z_a^{\text{total},t}$ be the cumulative reward of arm $a \in [\![K]\!]$ observed up to round $t \in [\![17W]\!]$, included. Assuming there is no elimination before round $3W$ (we choose $d$ and $\varepsilon$ in such a way that this is true), we have that:

$$Z_1^{\text{total},3W} = \frac{1}{2}W, \quad Z_2^{\text{total},3W} = \left(\frac{1}{2} - \varepsilon\right)W, \quad Z_3^{\text{total},3W} = \left(\frac{1}{2} - d\right)W.$$

Then:

$$Z_1^{\text{total},3W} - Z_2^{\text{total},3W} = \varepsilon W, \quad Z_1^{\text{total},3W} - Z_3^{\text{total},3W} = dW.$$

Let:

$$d := \frac{B}{W-1}, \quad \varepsilon := \frac{B}{2W}$$

where $B$ is the budget of the algorithm. These choices are such that we eliminate arm 3 at the end of round $3W$ (and not before), losing to arm 1. Arm 2, instead, stays alive. To satisfy $d \leqslant 1/2$, it is sufficient to require $W \geqslant 3B$. After round $3W$, the algorithm pulls only arms 1 and 2. When $r_1(t) \geqslant r_2(t)$, their difference is at most $\varepsilon$. Thus:

$$Z_1^{\text{total},5W} - Z_2^{\text{total},5W} \leqslant 2\varepsilon W = B.$$

Hence, arm 2 is not eliminated before round $5W$ (included). By the choice of the instance, in virtue of Equation (35), after round $5W$, we have $r_1(t) = 1/2 - d$. Thus, after each round robin cycle, which takes 2 rounds, $Z_2^{\text{total},t} - Z_1^{\text{total},t}$ increases by $d - \varepsilon$. Then:

$$Z_2^{\text{total},17W} - Z_1^{\text{total},17W} = 6(d-\varepsilon)W - (Z_1^{\text{total},5W} - Z_2^{\text{total},5W}) \geqslant 3B - B = 2B.$$

This means that, at some point after round $5W$, arm $1$ will be eliminated, losing to arm $2$. But it is evident that $1 \in A^*$ and $2 \in (A^*)^c$. However, it is important to notice that $2 \in \mathcal{I}_w^{\times}$, consistent with our analysis. It remains to show that there are choices of $B$, $W$, $T$, and $L$ which satisfy the hypotheses of the lemma and the additional requirements we imposed. In particular, they need to satisfy:

$$\begin{cases} \sqrt{\frac{17W \ln(3) \ln(T)}{3}} \leqslant B \leqslant \frac{W}{3} \\ 34^2 \frac{B}{W-1} \leqslant L \\ 17W \leqslant T \\ 2 \leqslant W \end{cases}.$$

It is clear that such an assignment exists. Furthermore, we can find such an assignment even when we restrict the budget to the natural choice which has order $W^{1/2}$.

