# OpenReview forum: "Tightening Regret Lower and Upper Bounds in Restless Rising Bandits"
_NeurIPS.cc/2025/Conference — NeurIPS 2025 poster_

### Official Review · Reviewer_w3Bt · 2025-06-10

**Clarity:** 3
**Significance:** 4
**Originality:** 3
**Rating:** 5
**Confidence:** 3

**Summary:**

The rising and rising concave restless MAB setting is considered. The previous best lower bound for both settings was $\Omega(T^{1/2})$. This work improves the lower bound to $\Omega(T^{3/5})$ for the rising concave setting and to $\Omega(T^{2/3})$ for the rising setting. The previous best upper bound for the rising setting was $O(T^{2/3})$, which matches the newly shown lower bound, while the best upper bound for the rising concave setting was $O(T^{2/3})$. This work improves the upper bound in the rising concave setting to $\tilde{O}(T^{7/11})$, which nearly matches the new lower bound. Experiments are provided that support the theoretical results.

**Questions:**

Please see the Weaknesses above.

**Ethical Concerns:**

["NO or VERY MINOR ethics concerns only"]

**Final Justification:**

The authors have addressed my concerns. I think this paper is a significant contribution as it almost gets the optimal rate in a setting that is of interest by many other researchers.

**Limitations:**

Yes

**Paper Formatting Concerns:**

No concerns.

**Quality:**

3

**Strengths And Weaknesses:**

**Quality**

*Strengths*
1. The definitions and results in the body are all well defined. I have not checked the appendix; however, it does seem that the analysis follows common strategies used in previous work, which makes it likely to be correct.

**Clarity**

*Strengths*
1. Setting is clearly defined and results are easy to follow, especially due to the nice table.

*Weaknesses*
1. After Theorem 3.2 it is mentioned that the lower bound matches the upper bound when $V_T \le 1$ since it is assumed that $\Gamma(1, T) \le 1$ in the lower bound. Does this imply that if $V_T > 1$ then the lower bound does not hold and the previous lower bound of $\Omega(T^{1/2})$ would have to be used in that case? Is this assumption critical for your lower bound construction to go through, and if so can it elaborated on why you believe it is just an artifact of the analysis?

**Significance**

*Strengths*
1. The paper provides improved lower bounds for the rising and rising concave restless MAB problems. It also provides an improved upper bound for the rising concave setting. The lower and upper bounds match in the rising setting and they almost match in the rising concave setting, which is a significant contribution.

**Originality**

*Weaknesses*
1. It would be helpful if the lower bound construction used to arrive at Lemma 3.1 could be compared to previous construction of Besbes et al. 2014. In particular what was the insight or change in the technique that allowed for the improved bounds. Currently it is unclear if the entire construction is novel or if only a specific modification was made that was very useful.

---

> ### Author Rebuttal · Authors · 2025-07-31
>
> We thank the Reviewer for the time spent reviewing our work and for having appreciated it. Below, our answers to the Reviewer's questions and comments.
>
> > After Theorem 3.2 it is mentioned that the lower bound matches the upper bound when $V_T \leq 1$ since it is assumed that $\Upsilon_\nu(1, T) \leq 1$ in the lower bound. Does this imply that if $V_T > 1$ then the lower bound does not hold and the previous lower bound of $\Omega(T^{1/2})$ would have to be used in that case? Is this assumption critical for your lower bound construction to go through, and if so can it elaborated on why you believe it is just an artifact of the analysis?
>
> When $V_T > 1$, $\min\\{ V_T, 1 \\}$ evaluates to $1$ and the lower bounds become $\Omega(T^{2/3} K^{1/3})$ and $\Omega(T^{3/5} K^{2/5})$ for the rising and rising concave settings, respectively. Observe that, in the rising (and thus also in the rising concave) setting $V_T \leq K$, thus, even when $V_T > 1$, the dependency on $T$ in the lower bound is not affected. In order to build the lower bounds, we generalized the approach used in (Besbes et al., 2014), where the time horizon is split in windows, in each window, one arm, chosen at random, is promoted to optimal, while all the suboptimal ones share the same expected rewards. An instance with this structure which further satisfies the non-decreasing expected rewards assumption and has expected rewards bounded in $[0, 1]$, has necessarily the cumulative increment $\Upsilon_\nu(1, T) \leq 1$ (see Figure 1). Intuitively, in order to get a certain value of variation budget $V_T$ in the lower bound, you need to construct a "difficult" instance which "consumes all the budget", i.e., $\Upsilon_\nu(1, T) = V_T$. Since $\Upsilon_\nu(1, T) \leq 1$, this is not possible when $V_T > 1$. This is why we believe that it is an artifact of the analysis.
>
> > It would be helpful if the lower bound construction used to arrive at Lemma 3.1 could be compared to previous construction of (Besbes et al., 2014). In particular what was the insight or change in the technique that allowed for the improved bounds. Currently it is unclear if the entire construction is novel or if only a specific modification was made that was very useful.
>
> Lemma 3.1 is a generalization of the construction used in Theorem 1 (Besbes et al., 2014) which carries a certain amount of novelty and is of independent interest (outside of the context of restless rising bandits) since it allows to construct lower bounds for general subclasses of restless MABs.
> In particular, as it has been partially anticipated in the previous answer, the challenging instance used in (Besbes et al., 2014) to derive the lower bound has the following structure. The time horizon is split in windows; in each window, one arm, chosen at random, is promoted to optimal and has expected reward equal to $\frac{1}{2} + \varepsilon$, while all the suboptimal ones share expected reward equal to $\frac{1}{2}$. This construction produces a class of instances with no particular structure since the expected reward of each arm can either increase, decrease, or stay constant between one window and the next, depending on the sequence of optimal arms. Thus, it is difficult to inject further assumptions on the evolution of the expected rewards (like non-decreasing or concave).
> Lemma 3.1 generalizes the approach introducing in each window two functions: the base trend $\overline{\mu}_w$, which specifies the evolution of the expected rewards of suboptimal arms, and the modified trend $\widetilde{\mu}_w$, which specifies the evolution of the expected reward of the optimal arm (which is still chosen at random, in each window). By carefully choosing the base and modified trends in each window, it is possible to construct a class of instances which satisfies the desired additional assumptions (e.g., see Figures 1, 2).
> At this point, Theorem 1 (Besbes et al., 2014) becomes a special case of  Lemma 3.1 that allows to derive the exact same lower bound by choosing base trend $\overline{\mu}_w(t) = \frac{1}{2}$ and modified trend $\widetilde{\mu}_w(t) = \frac{1}{2} + \varepsilon$ in each window.

---

> > ### Comment · Reviewer_w3Bt · 2025-08-02
> >
> > Thank you for clarifying my confusions. I am happy to keep my current rating of the work.
> > It would be nice to see a brief version of your explanation added to the camera ready paper regarding the relationship of your Lemma 3.1 and Theorem 1 from Besbes et al., 2014.

---

> > > ### Author Response · Authors · 2025-08-05
> > > **Re: Official Comment by Reviewer w3Bt**
> > >
> > > Thank you! We commit to adding this explanation in the camera-ready version.

---

### Official Review · Reviewer_W5ZX · 2025-06-29

**Clarity:** 3
**Significance:** 2
**Originality:** 2
**Rating:** 4
**Confidence:** 3

**Summary:**

This paper considers an instance of non-stationary multi-armed bandits given by the following framework: There are $K$ arms; each arm $i \in [K]$ has a time-varying mean $\mu_i(t)$ in $[0,1]$. The samples observed by the learner are drawn from a $\sigma$-subGaussian distribution. The objective is to minimize the expected regret with respect to a strategy that picks the arm with the minimal mean in each round $t$.

The authors assume a structure for the means. More precisely, they consider two cases:
(i) the case where the mean of each arm is increasing over the rounds, and
(ii) the case where the means are increasing while satisfying a concavity-like assumption.

The main contributions are threefold:
1. The authors derive a lower bound for the increasing means scenario.
2. A lower bound for the increasing concave means scenario.
3. An algorithm with upper bound guarantees for the increasing+concave means scenario.

The upper bound for the increasing+concave means case, in its dependence on the horizon $T$, is smaller than the lower bound for the increasing-only scenario, which shows that the concavity assumption impacts the optimal complexity of the problem.

**Questions:**

- Could you address the concern regarding the dependence on $V_T$? Removing the min would in my opinion improve the strength of the results.
- Since the authors assume that the arms' means are in $[0,1]$ and the distributions are $\sigma$-subGaussian, is it possible to make the dependence on $\sigma$ explicit in the lower bound by considering, for example, normal distributions instead of Bernoulli?
- Is there a way to bypass the need for prior knowledge of $V_T$ in the last bound of Theorem 4.4?

**Ethical Concerns:**

["NO or VERY MINOR ethics concerns only"]

**Final Justification:**

The main concerns about this paper were addressed in the rebuttal.

**Limitations:**

A discussion on the following points would benefit the reader: The threshold condition on $T$ for the last guarantee of Theorem 4.4 and the dependence on the number of arms of the guarantees of the same theorem.

**Paper Formatting Concerns:**

No concerns raised.

**Quality:**

3

**Strengths And Weaknesses:**

**Strengths:**
The paper gives a refinement of the framework of non-stationary bandits. The constructions of the distributions in the lower bounds especially for the increasing concave case are interesting and allow the authors to distinguish between the settings considered. The authors further adapt some existing ideas for algorithms and upper bounds by leveraging the assumed structure followed by the arms' means to derive bounds that are sharper in their dependence on the time horizon $T$.

**Weaknesses:**
- For the presented lower bounds, the main weakness is the dependence of the lower bound on the total variation $V_T$. Prior works, such as Besbes *et al.*, have a lower bound of order $V_T^{1/3}$, while the authors derive a lower bound of order $\min(1, V_T)^{1/3}$ for the rising case and $\min(1, V_T)^{1/5}$ for the rising and concave case. This difference is important, as it can create a gap when $V_T$ grows like $T^\alpha$.
- The dependence of the upper bounds on the number of arms $K$ is worse than known results for standard and non-stationary MAB problems.
- On the terminology used: to the best of my knowledge, the term *rising bandits* is used in the literature for a different type of framework, where the change in the means of the arms happens when the arms are pulled (which corresponds to the rested bandits setting). In contrast, the authors consider a *restless bandits* setting, where the arms' means change regardless of whether they are queried. If I am correct, I suggest the authors address this point, as it may be confusing to readers.
- Line 123: The comparison is not fair because in Besbes *et al.* no prior knowledge is required.
- Please modify the related work section in the appendix, as it is very similar to an existing one.

**Minor remark:**
Line 84: The considered definition corresponds to $\sigma$-subGaussian rather than $\sigma^2$-subGaussian.

---

> ### Author Rebuttal · Authors · 2025-07-31
>
> We thank the Reviewer for the time spent reviewing our work. Below, our answers to the Reviewer's questions and comments.
>
> > For the presented lower bounds, the main weakness is the dependence of the lower bound on the total variation $V_T$. Prior works, such as (Besbes et al., 2014), have a lower bound of order $V_T^{1/3}$, while the authors derive a lower bound of order $\min(1, V_T)^{1/3}$ for the rising case and $\min(1, V_T)^{1/5}$ for the rising concave case. This difference is important, as it can create a gap when $V_T$ grows like $T^\alpha$.
>
> > Could you address the concern regarding the dependence on $V_T$? Removing the $\min(\cdot)$ would in my opinion improve the strength of the results.
>
> In the rising setting (and thus also in the rising concave setting), the cumulative increment $\Upsilon_\nu(1, T)$ is always smaller or equal to $K$ (see Instances Characterization in Section 2). Hence, the variation budget $V_T$ is always chosen smaller than or equal to $K$. This implies that it is not possible for $V_T$ to grow like $T^\alpha$ (for sufficiently large $T$), making the dependence on $T$ in the lower bounds unaffected by the $\min(\cdot)$.
>
> > The dependence of the upper bounds on the number of arms $K$ is worse than known results for standard and non-stationary MAB problems.
>
> We agree with the Reviewer on the fact that the algorithm has a worse dependence in $K$.
> We think that this is partly due to the analysis, which, however, doesn't show significant margin of improvment.
> In particular:
> - The regret suffered by `RC-BE`($\alpha$) in a window is upper bounded by $3 K B\_w\^{(\alpha)} + \Delta_w\^{(\alpha)} d\_w^\*$ (see Lemma 4.1). The presence of $K$ in the first term is due to the fact that each arm can consume its exploration budget. The second term also increases with $K$, indeed, the diameter $d_w^*$ (see lines 221, 222) scales with the size of $\mathcal{I}_w^\times$ (line 221), which is $K$ in the worst case.
> - The analysis relies on an upper bound to the number of times two non-decreasing concave curves with values in $[0, 1]$ can intersect, diverge by a quantity $G > 0$, and then intersect again. When the curves are more than two, the maximum number of intersections of this kind between any pair of curves increases with the number of curves $K$.
>
> It is likley, that the dependence is also partly due to the algorithm itself, because of the extensive round-robin phases.
> However, we believe that the improvement in the dependence on $T$, under the customary implicit assumption $K \ll T$, gives us a better overall result compared to the state of the art. We will add a discussion on this in the updated version of the paper and the improvement of the dependence on $K$ among the future works.
>
> > On the terminology used: to the best of my knowledge, the term rising bandits is used in the literature for a different type of framework, where the change in the means of the arms happens when the arms are pulled (which corresponds to the rested bandits setting). In contrast, the authors consider a restless bandits setting, where the arms' means change regardless of whether they are queried. If I am correct, I suggest the authors address this point, as it may be confusing to readers.
>
> The term "rested" is used for bandits in which the expected reward changes only when the arm is pulled, while the term "restless" is used to consider the case in which the arm expected rewards evolve as a function of time (a.k.a. non-stationary setting). The term "rising" (Metelli et al., 2022) is used for all kinds of bandits in which the expected rewards are non-decreasing, being they restless or rested. In this work, we focus on restless rising bandits (as in the title), and use the term rising bandits, without the adjective restless, only after the setting has been carefully defined (see line 90).
>
> > Line 123: the comparison is not fair because in (Besbes et al., 2014) no prior knowledge is required.
>
> The reference to line 123 is likely incorrect, since in such line there is no comparison with (Besbes et al., 2014). We believe that the Reviewer wants to refer to line 183 (please correct us if this is not the case). However, we want to highlight two facts:
> - `Rexp3` (Besbes et al., 2014) does require prior knowledge of $V_T$ to obtain the $O(T^{2/3} V_T^{1/3})$ rate (see Theorem 2 of (Besbes et al., 2014), where hyperparameter $\Delta_T$ and by consequence $\gamma$ are chosen as a function of $V_T$).
> - Our algorithm allows to obtain a better regret without the knowledge of $V_T$ by choosing hyperparameter $\alpha$ in two different ways: either $\alpha = 8/3$ or $\alpha = \alpha''$ with $V_T = K$ (see Theorem 4.4), since, as remarked in one of the previous points, $K$ is always a valid upper bound to the cumulative increment.
>
> > Please modify the related work section in the appendix, as it is very similar to an existing one.
>
> We thank the Reviewer for the comment. We updated the manuscript by rephrasing parts of the related works section.
>
> > Line 84: the considered definition corresponds to $\sigma$-subGaussian rather than $\sigma^2$-subGaussian.
>
> We thank the Reviewer for the comment. We updated the manuscript.
>
> > Since the authors assume that the arms' means are in $[0, 1]$ and the distributions are $\sigma$-subGaussian, is it possible to make the dependence on $\sigma$ explicit in the lower bound by considering, for example, normal distributions instead of Bernoulli?
>
> Yes, it would be possible to use Gaussian distributions, but it would require significantly modifying the proof.
> Indeed, the analysis relies on the fact that the Bernoulli distribution has a discrete support. This allows to define some probability density functions (see Appendix B, p. 13) by directly evaluating the probability measures of interest at specific points. With continuous supports instead, Radon-Nikodym derivatives are needed. A smarter approach to make $\sigma$ appear in the lower bound, would involve considering another distribution with discrete support, with, for instance, the following structure:
>
> $$
> f(\\mu, \\sigma) := \begin{cases}
> \frac{3}{2} \sigma & \text{w.p.} & \frac{1}{4} + \frac{\mu}{2 \sigma} \\\\
> -\frac{1}{2} \sigma & \text{w.p.} & \frac{3}{4} - \frac{\mu}{2 \sigma}
> \end{cases}
> $$
>
> where $\mu \in [0, 1]$ is the **expected value** and $\sigma \geq 1$ the **subgaussianity constant** (in virtue of Hoeffding's lemma). Observe that $\mu \in [0, 1]$ and $\sigma \geq 1$ guarantee that the above distribution is well defined.
> In this way, all the arguments of the proof still hold (since they depend on the fact that the support is discrete and not on the actual values in it), furthermore, by means of Lemma D.1 (which again holds also for $f(\mu, \sigma)$ since it depends only on the fact that the cardinality of the support is $2$), it is possible to obtain an upper bound to the KL divergence of these new distributions which has the same structure of the KL divergence of two Gaussians with standard deviation $\sigma$, allowing to obtain the desired dependence on $\sigma$ in the lower bound.
> We thank the Reviewer for the constructive comment, as it allows to improve our work. We will add a remark on this in the final version.
>
> > Is there a way to bypass the need for prior knowledge of $V_T$ in the last bound of Theorem 4.4?
>
> The goal of the second bound in Theorem 4.4 is exactly to show that a better rate can be obtained when the knowledge of a tight upper bound for $\Upsilon_\nu(1, T)$ in the form of a variation budget $V_T$ is available. Indeed, the exponent to the term $\Upsilon_\nu(1, T)$ in the first bound is $1$. Under the assumption $\Upsilon_\nu(1, T) \leq V_T$, $\Upsilon_\nu(1, T) \approx V_T$, we get a better rate in the second bound. When the knowledge of a tight upper bound $V_T$ is not available, we can resort to the choices $\alpha = 8/3$ or $\alpha = \alpha''$ with $V_T = K$ (see Theorem 4.4), which allow to obtain still the same rate in $T$.
>
> > A discussion on the following points would benefit the reader: the threshold condition on $T$ for the last guarantee of Theorem 4.4 and the dependence on the number of arms of the guarantees of the same theorem.
>
> The additional requirement on $T$: $T \geq \max\\{ K^{-8/3} V_T^{-4/3} + 1, K^{16/5} V_T^{8/5} \\}$ present in the second bound of Theorem 4.4 guarantees $\alpha'' \geq 1$ (as specified in the proof of Theorem 4.4 in Appendix C) so that it is possible to apply the first part of the theorem. We will add a footnote in the main paper to highlight this fact.
> The fact that the second guarantee of the theorem requires a threshold on $T$ which depends on $K$ doesn't undermine the validity of the result, since the aim is to study the asymptotic behavior of the regret as $T$ grows, while other quantities (like $K$) are treated as constants.
> Furthermore, we are able to obtain the same rate of $\widetilde{O}(T^{7/11})$ in $T$ with the first bound which doesn't have such requirement.

---

> > ### Comment · Reviewer_W5ZX · 2025-08-04
> >
> > Thank you for your detailed response.

---

### Official Review · Reviewer_yLk5 · 2025-06-30

**Clarity:** 2
**Significance:** 3
**Originality:** 2
**Rating:** 4
**Confidence:** 3

**Summary:**

This paper proposes a general framework on the lower bound for the restless rising bandits and an algorithm that achieves a regret bound on restless rising concave bandits. The lower bound for the restless rising bandits matches the upper bound proven in previous works and the lower bound for the restless rising concave bandits discovers the degree of hardness of the problem.

**Questions:**

1. Could author provide a real-world example that motivates the model for the restless rising concave MAB bandits?

2. Which techniques or derivations in the proof are newly developed in this work?

3. What is the challening part of deriving a novel regret bound? What is the limitations of the previous works that could not improve the regret bound on the rising restless concave bandits?

4. In eq (3), what is the definition of $w(T)$?


**Minors**

In line 39, 'expected' is duplicated.

**Ethical Concerns:**

["NO or VERY MINOR ethics concerns only"]

**Final Justification:**

The authors response helped me understand the practicality of the model and the novelty of the technical derivations.

**Limitations:**

Yes.

**Paper Formatting Concerns:**

I found no formatting concerns in this paper.

**Quality:**

3

**Strengths And Weaknesses:**

**Strengths**

1. The general lower bound (Lemma 3.1) covers a wide range of the restless rising MAB bandits.

2.The upper bound shows an improvement over the previous work on the restless rising concave bandits and shows the interplay between the rising budget and the cumulative regret.

**Weakness**

1. It is not sure whether there is a novel techniques that overcomes the challenges of the previous works.

2. It would be better to include a specific example for the restless rising bandits for concrete motivation.

---

> ### Author Rebuttal · Authors · 2025-07-31
>
> We thank the Reviewer for the time spent reviewing our work and for the positive feedback. Below, our answers to the Reviewers questions and concerns.
>
> > Could author provide a real-world example that motivates the model for the restless rising concave MAB bandits?
>
> **Example 1: Athlethes training**: Consider the scenario in which a coach must choose an athlete to call up for the next competition from those they train. Ideally, the coach would like to select the athlete with the highest expected performance in the group. However, this expected performance must be estimated from the history of each athlete’s past call-ups, by observing the actual performances which are affected by noise, as they depend on the athlete’s condition on the particular day. Finally, the expected performance of each athlete varies over time. Through training, it is reasonable to assume that it increases over the course of their career, though with a gradually diminishing rate, in line with the assumptions of non-decreasing and concave growth. This problem can be modeled as a restless rising concave MAB, where each athlete corresponds to an action and each competition to a round.
>
> **Example 2: Online Model Selection with Paraller Training**: In many applications, an AI-powered service (such as an LLM) is offered to a customer base. Every time a user interacts with the service, the interaction is logged and used to improve the service. In modern applications, it is common to train multiple models (using the same log of interactions) and to use, at inference time, the one that promises the better customer experience. While every model tends to improve with an increasing dataset, it is only possible to evaluate its current quality by letting it face an actual interaction with a user, that can then leave a feedback. In this scenario, every model represents an arm, and in every round a user arrives, interacts the selected model for that round, and leaves a feedback. This feedback allows all of the models to re-train and improve (restless rising dynamic), as they can learn from a new observation, but can only quantify the quality of the model that is currently running (bandit feedback).
>
> > It is not sure whether there is a novel techniques that overcomes the challenges of the previous works.
>
> > Which techniques or derivations in the proof are newly developed in this work?
>
> **Lower Bounds**: Lemma 3.1 provides a novel generalization of the construction used in Theorem 1 (Besbes et al., 2014) to derive the lower bound for the general restless setting. The scope of this generalization can go beyond this work and allows to construct lower bounds (Theorems 3.2 and 3.3) for generic subclasses of restless bandits. The instances used to derive the lower bounds for the rising and rising concave setting are novel (see Figures 1, 2). In particular, the instance used to construct the lower bound for the rising setting (Figure 1), modifies the abrupt changes in the optimal arm exploited by Besbes et al. (2014) to further satisfy the non-decreasing expected rewards assumption.
> Instead, the instance used to construct the lower bound for the rising concave setting (Figure 2) smooths the changes of the optimal arm, being in this way compliant with the additional assumption on the concavity of the expected rewards, while trying to keep significant suboptimality gaps.
>
> **Upper Bound**: The analysis of the proposed algorithm is novel and changes significantly from the analysis of the algorithm we choose as starting point. In particular, the use of equivalence relations obtained by taking the transitive closure of "crossing" relations is novel (see Regret Analysis in Section 4) and solves issues present in previous approaches (see Appendix F). The improvement in the upper bound rate for rising concave MABs is due to a novel approach which limits the number of times two non-decreasing concave curves with values in $[0, 1]$ can intersect, diverge by a quantity greater than or equal to $G > 0$, and then intersect again (see Lemmas C.8, C.9 and C.10 in Appendix C).
>
> > What is the challening part of deriving a novel regret bound?
>
> The key insight, which allowed to derive a novel regret bound, is that the rising concave setting is easier w.r.t. the general restless setting since, under the non-decreasing assumption plus the concavity, it gets increasingly difficult for the optimal arm to change and diverge significantly. Our work exploited this observation, developing a bound to the number of times this can happen in the rising concave setting (see Lemma 4.3).
> This bound has been obtained by studying an optimization problem (as anticipated in the answer to the previous question) which considers the number of times two non-decreasing concave curves with values in $[0, 1]$ can intersect, diverge by a quantity greater than or equal to $G > 0$, and then intersect again.
> The main difficulty lies in reframing the regret bound as the aforementioned optimization problem. This process involves, among the other things, the transitive closure of "crossing" relations (cited in the previous answer), the definition of novel quantities like the diameter $d_w(i)$ (line 221), and the regret decomposition due to Lemma 4.1.
>
> > What is the limitations of the previous works that could not improve the regret bound on the rising restless concave bandits?
>
> Previous works focused on devising ad-hoc optimistic estimators which exploit the concavity of the instance. Unfortunately, this approach has been unsuccessful since it is difficult to design estimators with a satisfactory bias-variance trade-off.
> In particular, the estimator designed in (Metelli et al., 2022) tries to predict the evolution of the expected rewards curves with a linear affine projection built through an optimistic estimate of the values and the increments of the expected reward of an arm, in the rounds in which the arm has been pulled. This estimator doesn't exhibit desirable concentration properties as the projection is very sensitive to the noise in the increments estimates.
>
> > In eq (3), what is the definition of $w(T)$?
>
> The function $w(t)$ is defined at line 123 as $w(t) : t \mapsto \min\\{ w \in \mathbb{N}_{\geq 1} \ \text{s.t.} \ e_w \geq t \\}$ and represents the index of the window to which round $t$ belongs. Thus, $w(T)$ is the index of the last window in the considered time horizon $T$.
>
> > Typos
>
> Thank you, we will fix them.

---

> > ### Comment · Reviewer_yLk5 · 2025-08-07
> > **Rebuttal Acknowledgment**
> >
> > I appreciate the authors detailed response.
> > My questions are resolved and I will keep my score as is with higher confidence.

---

### Official Review · Reviewer_jBg5 · 2025-07-02

**Clarity:** 4
**Significance:** 3
**Originality:** 3
**Rating:** 5
**Confidence:** 4

**Summary:**

This paper studies the statistical complexity of two restless multi-armed bandits (MABs) models: (1) restless rising bandits and (2) restless rising concave bandits. A rising bandits is a bandits that has non-decreasing rewards over time, while a rising concave bandits is the one that has non-decreasing and concave rewards over time. In this paper, the authors propose a generic lower-bound instance constructing method for restless MAB, by dividing  the time horizon into windows with base and modified trends that are hard to distinguish.

Given this, the authors prove a new lower bound at $\Omega(T^{2/3})$ for restless rising bandits and $\Omega(T^{3/5})$ for the concave one. Meanwhile, they introduce an RC-BE algorithm that achieves a O(T^{7/11}) upper bound for the concave bandit. Finally, numerical experiments are conducted for RC-BE against Rexp3, R-less-UCB and the classical UCB1 algorithms as baselines, where the former outperforms all of the latters.

**Questions:**

### Questions:

1, Is the $K^{27/11}$ dependence on the upper bound relatively tight? I.e., is it a matter of algorithmic design or a matter of analysis?

2, How will the performance change if the concavity is enhanced? I.e. will it be beneficial if the bandits reward is $\alpha$-strongly concave (and as $\alpha$ increases)?

3, Is Lemma 3.1 able to be generalized to more bandits settings (for instance, non-stationary bandits as dynamic regret, or adversarial bandits)?

### Suggestions:

The authors are not required to make any changes to their original submission or deliver more results during the rebuttal period, from my point of view. They are encouraged to revise according to the suggestions before submitting their camera-ready or resubmission versions.

1, It would be better to plot the regret comparison diagrams on a log-log scale, where the slope represents the polynomial order dependence.

2, Some typos to fix: e.g. Page 2 Line 39, “expected expected”. Also, the reference of Jia et al (2024) is missing (while an Arxiv reference of it exists at least).

**Ethical Concerns:**

["NO or VERY MINOR ethics concerns only"]

**Final Justification:**

The rebuttal clarifies most of my concerns. I am supporting to this paper as it extends the scope of multi-armed bandits with non-trivial and rigorous techniques.

**Limitations:**

yes

**Quality:**

4

**Strengths And Weaknesses:**

### Strengths:

1, This paper proposes novel methods to prove information-theoretic lower bounds on restless bandits. This is beneficial for the whole community of online learning, reinforcement learning, and ML theory in general.

2, The authors propose an any-time practical algorithm that beats baselines that are commonly adopted. More importantly, it gets rid of pre-knowledge assumptions (w.r.t. $T$ and $V_T$) comparing to existing works.

3, The paper is clearly written, with results rigorously delivered and well justified.

### Weakness:

The main concern from my perspective is the regret dependences on $K$: On the one hand, the theoretical guarantee is $O(K^{27/11})$ w.r.t. $K$, which is far from the lower bound and intuitively far from optimal. On the other hand, the numerical experiments are only conducted on $K=5$, which is relatively small and also implies an undermine on the potential application to massive arms.

Besides, the tuning of $\gamma$ either requires pre-knowledge on $V_T$ or leads to a worse dependency (please correct me if I am wrong).

**Note**: Although the regret gap is not closed on $T$, I do not consider it as a significant weakness since they have already improving the existing results substantially, and I agree to leave it for future study.

---

> ### Author Rebuttal · Authors · 2025-07-31
>
> We thank the Reviewer for the time spent reviewing our work and for having appreciated it. Below, our answers to the Reviewer's questions and comments.
>
> > Is the $O(K^{27/11})$ dependence on the upper bound relatively tight? I.e., is it a matter of algorithmic design or a matter of analysis?
>
> We think that it is likely that, by employing a different analysis path, it would be possible to obtain a better dependence on $K$ without modifying the algorithm, but the current analysis strategy doesn't show significant room for improvement.
> In particular:
> - The regret suffered by `RC-BE`($\alpha$) in a window is upper bounded by $3 K B\_w\^{(\alpha)} + \Delta\_w\^{(\alpha)} d\_w^\*$ (see Lemma 4.1). The presence of $K$ in the first term is due to the fact that each arm can consume its exploration budget. The second term also increases with $K$, indeed, the diameter $d_w^*$ (see lines 221, 222) scales with the size of $\mathcal{I}_w^\times$ (line 221), which is $K$ in the worst case.
> - The analysis relies on an upper bound to the number of times two non-decreasing concave curves with values in $[0, 1]$ can intersect, diverge by a quantity $G > 0$, and then intersect again. When the curves are more than two, the maximum number of intersections of this kind between any pair of curves increases with the number of curves $K$.
>
> Finally, we suspect that the dependence on $K$ is due in part to the algorithm itself because of its extensive round-robin phases.
>
> > The tuning of $\gamma$ either requires pre-knowledge on $V_T$ or leads to a worse dependency.
>
> We assume that $\gamma$ is a typo, and that the Reviewer wants to refer to the hyperparameter $\alpha$ of the `RC-BE`($\alpha$) algorithm (please, correct us if this is not the case).
> Under the choice $\alpha = 8/3$, which doesn't require pre-knowledge of $V_T$, the dependence on $K$ is indeed worse ($O(K^3)$). Anyway, in a rising MAB (and thus also in a rising concave MAB), $V_T = K$ is always a valid upper bound to the cumulative increment $\Upsilon_\nu(1, T)$ and, under the choice $\alpha = \alpha''$ with $V_T = K$ (see Theorem 4.4) we get a better rate in $K$: $O(K^{24/11})$. Choosing $\alpha = 8/3$ versus $\alpha = \alpha''$ with $V_T = K$ really depends on the relative scale of the quantities involved in the upper bound.
>
> > How will the performance change if the concavity is enhanced? I.e. will it be beneficial if the bandits reward is $\alpha$-strongly concave (and as $\alpha$ increases)?
>
> Instead of $\alpha$-strong concavity, we considered other forms of enhanced concavity, although we did not include these results in the paper.
> Indeed, $\alpha$-strong concavity conflicts with the non-decreasing assumption as $\alpha$-strong concave curves need to be strictly decreasing for $t \geq t_0$ for some $t_0 \in \mathbb{R}$.
> Instead, we studied a subclass of rising concave MABs in which the increments are further constrained to satisfy $\gamma_i(t) \leq t^{-a}$ for some $a \geq 1$ (observe that every rising concave MAB with expected rewards bounded in $[0, 1]$ must satisfy the additional assumption with $a = 1$ since $\gamma(t) \leq (\mu(t) - \mu(1))/t \leq 1/t$). In this setting, assuming the knowledge of $a$, it is possible to devise an algorithm with upper bound rate $\widetilde{O}(T^{\max\{ 1/2, 1-a/3 \}})$ which matches the lower bound for stationary MABs for $a \geq \frac{3}{2}$. However, we believe that the additional assumption is very restrictive. Furthermore, the algorithm shows the same regret upper bound ($\widetilde{O}(T^{2/3})$) which holds for general restless MABs when $a = 1$ (which, as remarked before, is the standard rising concave MAB setting).
> We're going to add these results in the final version.
>
> > Is Lemma 3.1 able to be generalized to more bandits settings (for instance, non-stationary bandits as dynamic regret, or adversarial bandits)?
>
> Lemma 3.1 already holds for the whole class of non-stationary bandits with dynamic regret in the current form. It doesn't hold in the adversarial setting due to the fact that the regret is there defined differently by comparing the played action with the best fixed action in hindsight.
>
> > Suggestions and typos
>
> Thank you, we will fix them.

---

> > ### Comment · Reviewer_jBg5 · 2025-08-04
> >
> > Thanks the authors for their reply! The rebuttal clarifies many of my confusions. I will keep supporting this paper as most reviewers do.

---

### Official Review · Reviewer_9yKg · 2025-07-02

**Clarity:** 3
**Significance:** 3
**Originality:** 3
**Rating:** 4
**Confidence:** 2

**Summary:**

This paper investigates restless MABs whose expected rewards evolve over time.
They consider two different settings where the expected rewards are either non-decreasing (rising) and non-decreasing concave (rising concave).
They derive regret lower bounds of $\Omega(T^{2/3})$ for rising MABs and $\Omega(T^{3/5})$ for rising concave MABs, showing that the rising assumption alone does not simplify learning.
For the rising concave case, they propose the RC-BE($\alpha$) algorithm achieving a regret upper bound of $\tilde{O}(T^{7/11})$.

**Questions:**

Please comment point 1,2 in the above weakness part.

There is a typo in line 209: $\Delta_w^{(\alpha)} \rightarrow \Delta_l^{(\alpha)}$

**Ethical Concerns:**

["NO or VERY MINOR ethics concerns only"]

**Final Justification:**

I value this work for proposing the Rising MAB and Rising Concave MAB frameworks, which capture non-stationary rewards of environment. The theoretical analysis is concrete and contributes meaningful insights to the literature. Additionally, the authors have provided helpful intuitions in response to my questions about the theorem in the initial review. The authors have provided a clear rebuttal that addresses my questions. I maintain my positive rating.

**Limitations:**

Yes.

**Paper Formatting Concerns:**

No major formatting issues in this paper.

**Quality:**

3

**Strengths And Weaknesses:**

**Strengths**
1. The overall writing and presentation in the paper pretty good.
2. Table 1 was very helpful. It clearly sums up past work on restless MABs and demonstrates what this paper contributes.
3. Figure 1 was also helpful for understanding the rising and rising-concave setting.
4. Repeated actions may yield steadily improving rewards. Their restless rising (concave) bandits model this effect better and match reality than stationary models.

**Weaknesses**

I have no major issues but a few minor questions and would appreciate clarification to confirm my understanding.

1. Beyond the formal lower bound in Theorem 3.2, could you give an intuitive explanation for why the additional rising structure does not make the problem easier?
2. Theorem 3.2 shows a lower bound of $\Omega(T^{2/3})$ for rising MABs. Theorem 3.3 shows $\Omega(T^{3/5})$ for rising concave MABs. Does this imply that the rising MAB is a harder problem than the rising concave MAB? Could you briefly compare the two settings?

---

> ### Author Rebuttal · Authors · 2025-07-31
>
> We thank the Reviewer for the time spent reviewing our work and for having appreciated it. Below, our answers to the Reviewer's questions and comments.
>
> > Beyond the formal lower bound in Theorem 3.2, could you give an intuitive explanation for why the additional rising structure does not make the problem easier?
>
> The complexity of the general restless setting is due to the fact that the optimal arm can **change abruptly** and very often over the time horizon. This can be seen by looking at the lower bound instance used in the proof of Theorem 1 (Besbes et al., 2014), where the time horizon is split in windows: in each window all arms have the same expected reward, equal to $\frac{1}{2}$, except for one, chosen at random, which has expected reward equal to $\frac{1}{2} + \varepsilon$, being thus optimal. In the rising scenario, despite the additional constraint, we can **still exploit abrupt changes** (see Figure 1) to construct instances which are sufficiently challenging, but we have to be more careful in the variation $\varepsilon$ due to the fact that we cannot allow arms to "come back" to expected value $\frac{1}{2}$ or "go beyond" expected value $1$. We conjecture that what makes identificability challenging is the chance to have very rapid shifts in expected values.
>
> > Theorem 3.2 shows a lower bound of $\Omega (T^{2/3})$ for rising MABs. Theorem 3.3 shows $\Omega (T^{3/5})$ for rising concave MABs. Does this imply that the rising MAB is a harder problem than the rising concave MAB? Could you briefly compare the two settings?
>
> Yes, it is correct. For rising MABs, we have a lower bound in the order of $\Omega(T^{2/3})$, while we have an algorithm for rising concave MABs whose regret is upper bounded by $\widetilde{O}(T^{7/11})$. Being $2/3 > 7/11$, this allows us to conclude that rising concave MABs constitute an easier setting. In light of what has been remarked in the answer to the previous question, this is due to the fact that, under the non-decreasing assumption in conjuction with the concavity, it gets increasingly harder for the optimal arm to change and diverge significantly from suboptimal ones.
>
> > Suggestions and typos
>
> Thank you, we will fix them.

---

> > ### Comment · Reviewer_9yKg · 2025-08-07
> >
> > Thank you for the thoughtful response. Your clarifications addressed my concerns. I will maintain my positive rating.

---

### Decision · Program_Chairs · 2025-09-17

**Decision:**

Accept (poster)

**Comment:**

The paper studies the problem of restless rising bandits, in which all arms improve over time (that is, their average return increases). They first provide a lower bound of $\Omega(T^{2/3})$, matching existing upper bounds for this problem. They then analyze the problem when arm means are concave (reward improvement decreases over time) and prove lower bounds of $\Omega(T^{3/5})$. They complement this bound with an upper bound of $\tilde{O}(T^{7/11})$, showing that the concave case is separated from both the stationary setting (where an $O(\sqrt{T})$ regret is achievable) and the general rising setting.

The reviewers agreed that the problem is well-motivated, the paper is well-written and that all the aforementioned contribution are significant and might be generalized to other similar settings. The main weakness raised by the reviewers is the sub-optimal dependence of the bounds on the number of arms $K$, but this is only a minor weakness.